# Social trauma engages lateral septum circuitry to occlude social reward

Long Li[1,2], Romain Durand-de Cuttoli[1,2], Antonio V. Aubry[1,2], C. Joseph Burnett[1,2], Flurin Cathomas[1,2], Lyonna F. Parise[1,2], Kenny L. Chan[1,2], Carole Morel[2,3], Chongzhen Yuan[2,4,5], Yusuke Shimo[1,2], Hsiao-yun Lin[2,4,5], Jun Wang[2,4,5] & Scott J. Russo[1,2✉]

In humans, traumatic social experiences can contribute to psychiatric disorders[1]. It is suggested that social trauma impairs brain reward function such that social behaviour is no longer rewarding, leading to severe social avoidance[2,3]. In rodents, the chronic social defeat stress (CSDS) model has been used to understand the neurobiology underlying stress susceptibility versus resilience following social trauma, yet little is known regarding its impact on social reward[4,5]. Here we show that, following CSDS, a subset of male and female mice, termed susceptible (SUS), avoid social interaction with non-aggressive, same-sex juvenile C57BL/6J mice and do not develop context-dependent social reward following encounters with them. Non-social stressors have no effect on social reward in either sex. Next, using whole-brain Fos mapping, in vivo Ca²⁺ imaging and whole-cell recordings, we identified a population of stress/threat-responsive lateral septum neurotensin (NT[LS]) neurons that are activated by juvenile social interactions only in SUS mice, but not in resilient or unstressed control mice. Optogenetic or chemogenetic manipulation of NT[LS] neurons and their downstream connections modulates social interaction and social reward. Together, these data suggest that previously rewarding social targets are possibly perceived as social threats in SUS mice, resulting from hyperactive NT[LS] neurons that occlude social reward processing.

Social avoidance manifests across a host of psychiatric illnesses, with causes ranging from disinterest in social contact[6] to negative emotional states evoked by social encounters[7]. While the causes of social avoidance are diverse[8], past social trauma can result in severe social avoidance thought to reflect reduced social reward[2,9]. Despite a deep clinical understanding of social trauma and its resultant effects on social behaviour, we know very little about the underlying neural circuitry involved. Preclinical social trauma models, such as chronic social defeat stress (CSDS), have been used to better understand neural circuit mechanisms that control emotional behaviour[4,5,10]. CSDS reduces exploratory behaviours and preference for natural rewards like sucrose, and results in severe social avoidance interpreted as social anhedonia[5,10]. However, past CSDS studies assessed social interaction with an adult CD-1 mouse, similar to those used as aggressors to induce the social trauma. Social avoidance under these circumstances probably reflects fear or submissive behaviour rather than impaired social reward.

To better understand whether social reward deficits are induced by CSDS, we assessed social behaviour by testing social interaction and social conditioned place preference (sCPP) with a non-threatening, same-sex juvenile C57BL/6J mouse that, under control conditions, is rewarding. CSDS—but not non-social chronic stressors like chronic variable stress (CVS)—blocks social reward in a subset of mice termed susceptible (SUS). We next employed a circuit-based approach to better understand the mechanisms by which previous traumatic social experience with an adult male aggressor affects subsequent social reward processing. Following CSDS, SUS mice exhibit heightened activity within lateral septum neurotensin (NT[LS]) neural circuitry, which occludes social reward and promotes sustained social avoidance behaviour even when presented with a non-threatening social situation.

## SUS mice exhibit social reward deficits

To investigate how CSDS affects social interaction and social reward, 7–8-week-old wild-type (WT) male and female mice underwent standard CSDS followed by social interaction testing with a CD-1 or ERα-Cre F1 mouse[10,11]. As previously described, mice were classified as either resilient (RES) or SUS based on their social interaction behaviour (that is, social interaction (SI) ratio) (Fig. 1a,b,g and Extended Data Fig. 1a,b,d,e). This was followed by a resident intruder (RI) test and sCPP with 4–6-week-old, same-sex juvenile C57BL/6J mice. During the RI test, control (CTRL) and RES mice exhibited similar social behaviours towards the juvenile, including the amount of active interaction (that is, approach, close following and sniffing). Mice in these groups rarely withdrew when the juvenile approached and investigated, which we define as passive social investigation. Conversely, SUS mice

[1]Nash Family Department of Neuroscience, Icahn School of Medicine at Mount Sinai, New York, NY, USA. [2]Friedman Brain Institute, Icahn School of Medicine at Mount Sinai, New York, NY, USA. [3]Department of Pharmacological Sciences, Icahn School of Medicine at Mount Sinai, New York, NY, USA. [4]Department of Neurology, Icahn School of Medicine at Mount Sinai, New York, NY, USA. [5]James J Peters VA Medical Center, Research & Development, New York, NY, USA. ✉e-mail: scott.russo@mssm.edu

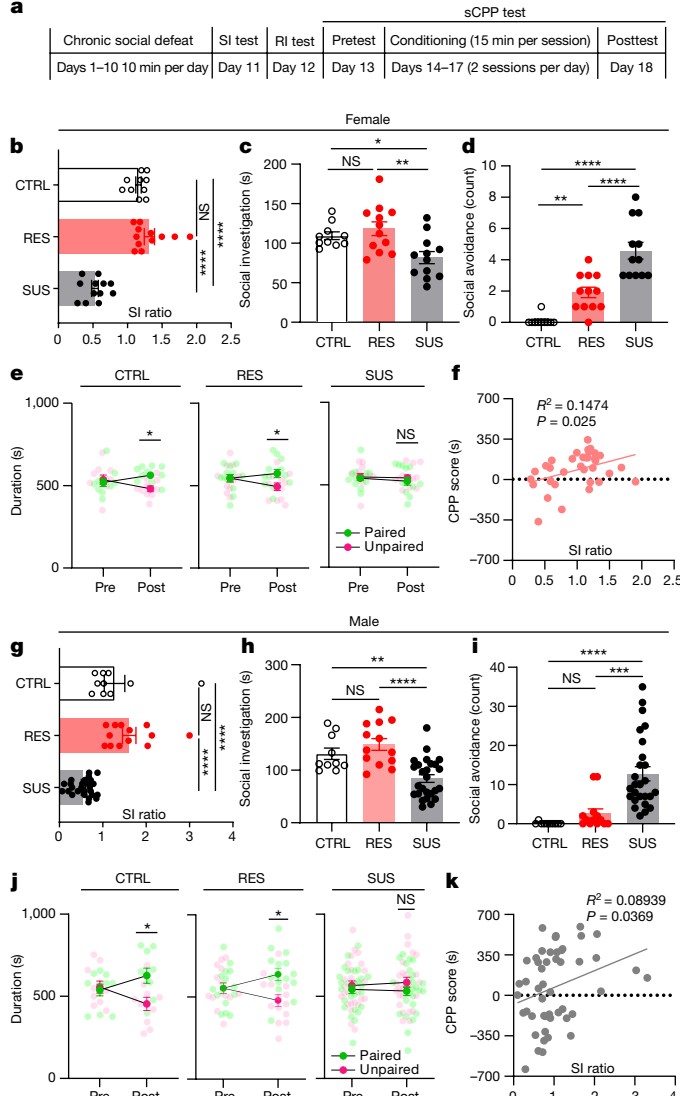

**Fig. 1 | Susceptible mice show social reward impairment after CSDS in both males and females. a**, Experimental timeline for social behaviour tests following chronic social defeat. **b,g**, SI ratios of females (one-way analysis of variance (ANOVA) with Tukey's multiple-comparisons test, $F(2, 31) = 53.96$, $P < 0.0001$, $n = 10$ (CTRL), 12 (RES), 12 (SUS) (**b**); and of males (one-way ANOVA, $F(2, 46) = 24.36$, $P < 0.0001$, $n = 10$ (CTRL), 13 (RES), 26 (SUS) (**g**). **c,h**, Social investigation time from RI test of females (one-way ANOVA, $F(2, 31) = 6.755$, $P = 0.0037$ (**c**); and of males ($F(2, 46) = 14.82$, $P < 0.0001$) (**h**). **d,i**, Social avoidance of females ($F(2, 31) = 33.13$, $P < 0.0001$) (**d**) and males ($F(2, 46) = 15.37$, $P < 0.0001$) (**i**). **e**, Time spent in paired and unpaired chambers during sCPP test (female CTRL (two-way repeated-measures ANOVA followed by Šídák's multiple-comparisons test, $F(1, 18) = 7.023$, $P = 0.0163$, $n = 10$); RES mice ($F(1, 22) = 4.598$, $P = 0.0433$, $n = 12$); and SUS mice ($F(1, 22) = 0.08155$, $P = 0.7779$, $n = 12$)). **f**, Correlation between SI ratio and sCPP in females ($R^2 = 0.1474$, $P = 0.025$). **j**, Time spent in paired and unpaired chambers during sCPP test (two-way repeated-measures ANOVA: male CTRL ($F(1, 18) = 6.074$, $P = 0.0240$, $n = 10$); RES mice ($F(1, 26) = 7.499$, $P = 0.0110$, $n = 13$); and SUS mice ($F(1, 50) = 0.4818$, $P = 0.4908$, $n = 26$)). **k**, Correlation between SI ratio and sCPP in males ($R^2 = 0.08939$, $P = 0.0369$). NS, not significant. *$P < 0.05$, **$P < 0.01$, ***$P < 0.001$, ****$P < 0.0001$. All data expressed as mean ± s.e.m.

exhibited much less active social investigation, longer latency to the first social bout and significantly more social avoidance during passive social investigation with a juvenile (Fig. 1c,d,h,i and Extended Data Fig. 1c,f). Social investigation time, social avoidance and latency to

investigate correlated with SI ratio during testing with a CD-1 (Extended Data Fig. 1g–l). These results show that SUS mice exhibit avoidance not only toward aggressive adult CD-1 male mice, but also toward non-threatening, same-sex C57BL/6J juvenile mice. We next used the sCPP test to assess social preference; CTRL and RES, but not SUS, mice formed social preference to the intruder-paired context (Fig. 1e,j), suggesting that juvenile interaction is not rewarding to SUS mice. sCPP score correlated with SI ratio (Fig. 1f,k) as well as social investigation time, social avoidance counts and latency to the first social bout during the RI test (Extended Data Fig. 1m–r). The female oestrous cycle was not associated with any differences in sCPP formation (Extended Data Fig. 1s). Interestingly, we found that female mice formed a significant sCPP only when the juvenile mice were confined to a wire-mesh cup during conditioning (Extended Data Fig. 1t), so we used this design for all studies in females. All behavioural parameters were normally distributed except for social avoidance (Extended Data Fig. 1u). Given that sCPP is dependent on intact learning and memory processes, we performed novel-object recognition and novel-location tests and found no evidence of learning and memory deficits in SUS or RES mice compared with CTRL mice (Extended Data Fig. 2a–c). To test whether the order in which behavioural tests were performed affected aspects of social behaviour, we reversed the order of testing (sCPP–RI–SI) in WT mice following CSDS and found similar effects (Extended Data Fig. 2d,e). Next, we grouped mice first by sCPP scores (social preference) and found a similar positive correlation with social investigation in the RI test along with a trend for SI ratio (Extended Data Fig. 2f,g), which again suggests that these different social behaviours largely correlate with one another. Together, these data support the idea that CSDS-induced social avoidance results from disruptions in social reward processing, which led us to consider that SUS mice may in fact perceive juvenile social targets as threatening or stressful.

## NT^LS neurons are hyperactivated in SUS mice

To investigate the circuit mechanisms mediating social reward deficits in SUS mice, we conducted a cleared whole-brain Fos mapping procedure using the iDISCO+ method[12] to examine differentially modulated brain regions following CSDS when mice were exposed to juvenile intruders (Fig. 2a and Extended Data Fig. 3a–h). Cleared brains (Fig. 2b) were imaged on a lightsheet microscope (Fig. 2c), followed by registration and annotation (Extended Data Fig. 3i) using ClearMap[12]. To first screen potentially relevant brain regions, Fos⁺ cells were compared among CTRL, SUS and RES mice to identify differentially regulated brain regions in both sexes (Fig. 2d, Extended Data Tables 1–5 and Supplementary Table 1). Interestingly, we found dramatic sex-based differences in Fos activity when comparing RES and SUS mice, despite both sexes exhibiting similar social deficits. Compared with RES males, SUS males showed a significant increase in Fos⁺ cells in 47 regions whereas SUS females showed significant increases in only 22 regions. Notably, the lateral septum (LS) was one of the most highly activated brain regions in both SUS males and females compared with RES mice, so we selected it for further investigation. To confirm that Fos activation in SUS mice was due to the presence of a social target, we performed an additional RI test following CSDS with a novel object versus a juvenile intruder. Under these conditions, we found that Fos activity was significantly higher in SUS mice following juvenile interaction compared with novel-object interaction. Although we observed a slight increase in Fos activity following both novel-object and juvenile interaction in RES mice, there were no significant differences in time spent between them (Extended Data Fig. 3j,k).

Due to its dense reciprocal interconnections throughout the brain's primary reward centres, the LS is often thought of as a nexus for mood[13–16], motivation[17,18] and spatial information processing[19]. Interestingly, in SUS mice following juvenile RI we found that most Fos-expressing neurons were located specifically in the lateral-ventral

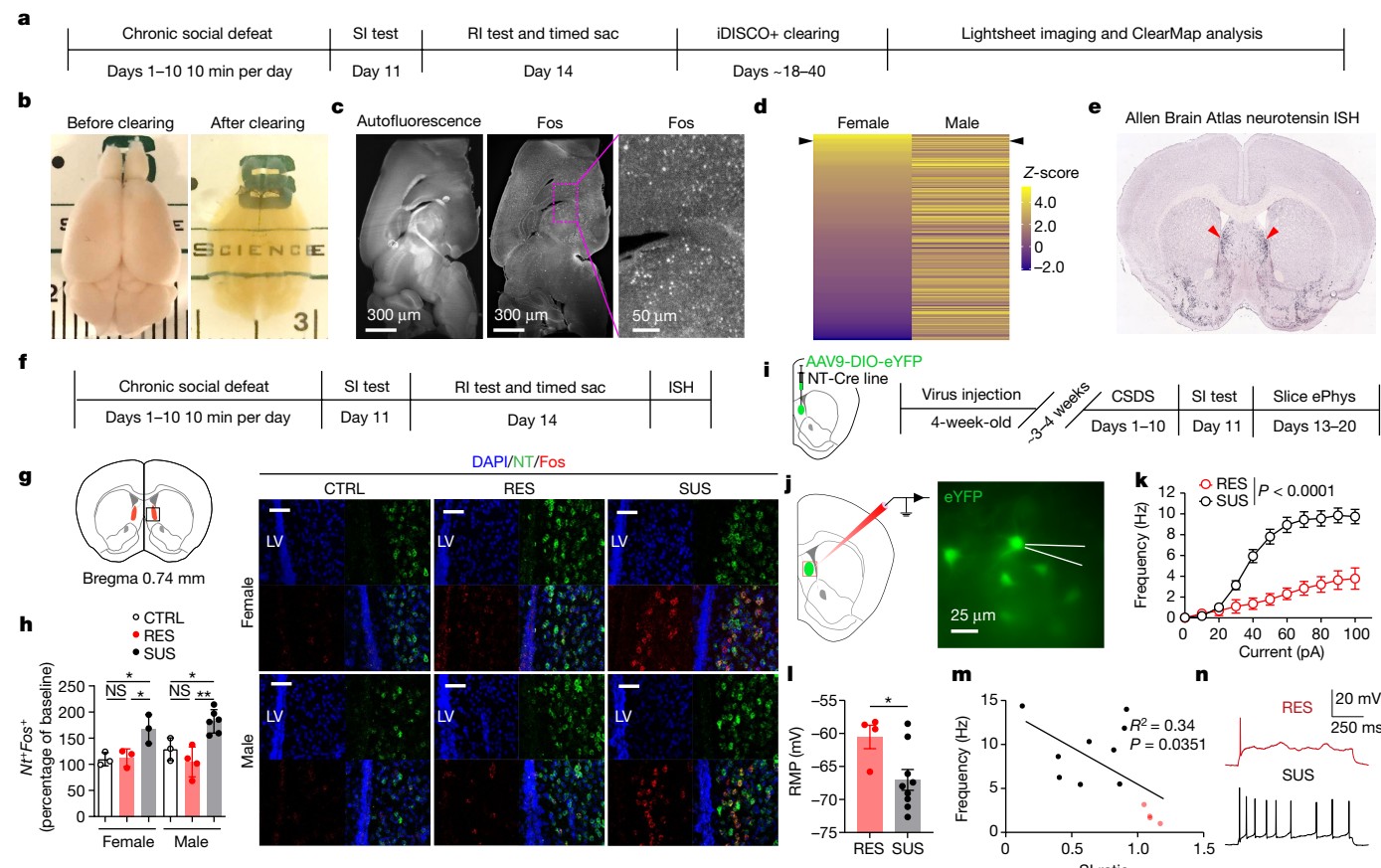

**Fig. 2 | NT^LS neuronal activity is associated with altered social behaviours in susceptible mice. a**, Timeline of iDISCO+ Fos analysis from the RI test after CSDS. Timed sac, mice were perfused 90 min after RI test. **b**, Mouse brain before and after iDISCO+ clearing. **c**, Autofluorescence and Fos signal from lightsheet imaging. **d**, ClearMap analysis showing differentially activated brain regions from RES versus SUS mice. Arrowheads indicate LS. **e**, Neurotensin expression from Allen Brain Atlas ISH data. **f**, Timeline of ISH. **g,h**, Multiplex ISH (**g**) showing Fos expression (**h**) in NT neurons in females (one-way ANOVA, $F(2, 6) = 7.887$, $P = 0.0209$, $n = 3$ mice per group, three slices per mouse) and males ($F(2, 10) = 13.13$, $P = 0.0016$, $n = 3$ (CTRL), 4 (RES), 6 (SUS), three slices per mouse); scale bars, 50 µm. LV, lateral ventricle. **i**, Timeline of slice electrophysiology (ePhys)

following CSDS. **j**, eYFP+ NT^LS neurons patched in whole-cell configuration. **k**, Current–frequency curve showing counts of action potentials evoked by incremental steps of injected current. NT^LS neurons from SUS mice ($n = 55$ neurons) compared with RES mice (two-way ANOVA, $P < 0.0001$, $n = 19$). **l**, Resting membrane potential (RMP) for SUS and RES mice (two-tailed Mann–Whitney test, $P = 0.0336$, $n = 4$ (RES), 9 (SUS)). **m**, Correlation between SI ratio and firing rate evoked by a 100 pA step current (Pearson's correlation, $R^2 = 0.34$, $P = 0.0351$). Each dot represents the mean value per mouse for RES (red, $n = 4$) and SUS (black, $n = 9$) mice. **n**, Sample traces of excitability for RES (red) and SUS (black) mice following 100 pA current injection. $*P < 0.05$, $**P < 0.01$. All data expressed as mean ± s.e.m.

portion of the LS. Using the Allen Brain Atlas in situ hybridization (ISH) database[20], we found several genes expressed specifically in the lateral-ventral portion of the LS, including neurotensin (*Nt*) (Fig. 2e) and corticotrophin-releasing hormone receptor 2 (*Crhr2*). Oxytocin receptor (*Oxtr*) was expressed specifically in the lateral portion, somatostatin (*Sst*) was expressed mainly in the dorsal-lateral portion and dopamine receptor D3 (*Drd3*) was expressed only in the medial portion of the LS. Several recent studies have examined the role of these molecularly defined cell types in regulation of behaviour, including *Sst+* neurons in fear conditioning[21], *vGAT+* and *Nt+* neurons in stress-suppressed feeding[22,23], *Crhr2+* neurons in anxiety-like behaviour[24], as well as *Oxtr+* and *Drd3+* neurons in social fear[25] and social dysfunction[26]. To determine which cell type was activated by juvenile social interaction in SUS mice, we performed multiplex ISH on brain slices from CSDS mice following juvenile RI (Fig. 2f). We found over 94% colocalization between *Nt* and *Fos* in SUS mice, with very limited expression of *Fos* in *Nt−* cells (Fig. 2g,h and Extended Data Fig. 4a). Around 100% of all *Nt+* cells were GABAergic (Extended Data Fig. 4b,c). Interestingly, *Nt* and *Crhr2* messenger RNA were largely colocalized in the anterior part but not in the posterior part of the LS, where we found significant increases in Fos levels following juvenile RI in SUS mice (Extended Data Fig. 4d,e).

*Nt+* neurons had an overlap of about 5% with *Drd3+* and of about 20% with *Oxtr+* neurons (Extended Data Fig. 4f–i). Interestingly, we also found an increase in *Sst+ Fos+* neurons in SUS mice following juvenile social interaction relative to CTRL mice, but not between RES and SUS mice (Extended Data Fig. 4j,k). Last, we found no differences in *Nt− Fos+* neurons between CTRL, RES and SUS mice (Extended Data Fig. 4l). Together these data highlight a potentially strong involvement of NT^LS neurons in social reward deficits in SUS mice.

To confirm that NT^LS neurons were indeed hyperactivated in SUS mice following interaction with a juvenile, we used a whole-cell slice electrophysiological approach to record NT^LS neurons in defeated male mice following a juvenile RI test (Fig. 2i,j). We found that NT^LS neurons from SUS mice showed increased excitability (Fig. 2k,n), as well as decreased resting membrane potential (Fig. 2l and Extended Data Fig. 5a), when compared with RES mice. Interestingly, we also found that the excitability of these cells was negatively correlated with the SI ratio observed following CSDS (Fig. 2m). These changes in intrinsic properties of NT^LS neurons suggest that CSDS induces lasting adaptations in these cells, which mediate social dysfunction. Interestingly, we found no differences in other properties of these cells (action potential threshold, amplitude, half-width or fast hyperpolarization;

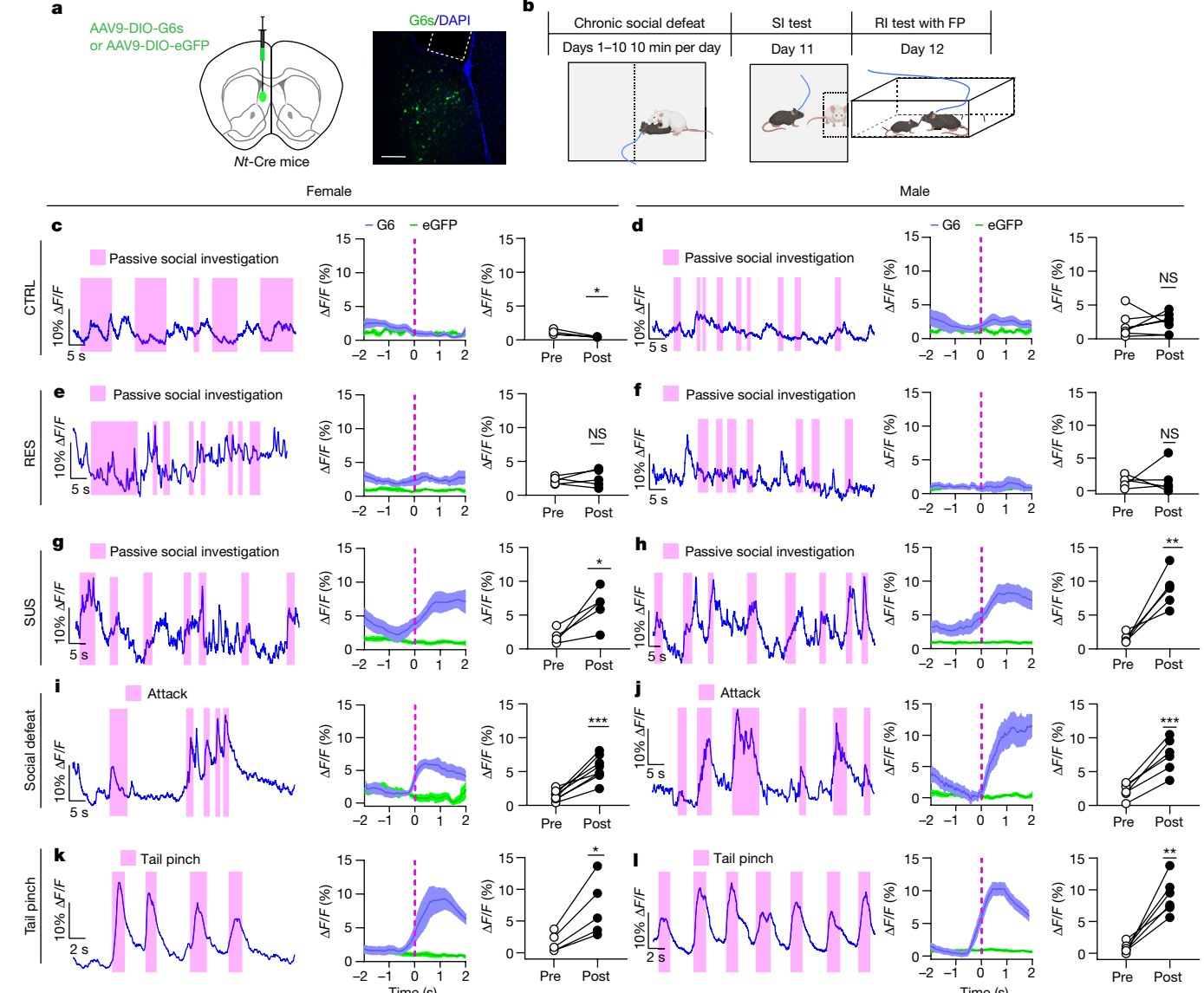

**Fig. 3 | In vivo NT$^{LS}$ activity in different social and stress contexts.**
**a**, AAV-DIO-GCaMP6s injection and expression in LS. **b**, Timeline of FP experiments. **c**–**h**, Left, representative Ca$^{2+}$ trace of NT$^{LS}$ neurons during resident intruder test (pink strips indicate passive social investigation); middle, peri-event plot of NT$^{LS}$ neuron activity 2 s before and after intruder approach; right, statistics for neuron activity 2 s before and 2 s after social events in CTRL females (paired two-tailed $t$-test, $t6 = 3.379$, $P = 0.0149$, $n = 7$) (**c**) and males ($t6 = 0.5081$, $P = 0.6295$, $n = 7$) (**d**); in RES females ($t4 = 0.6528$, $P = 0.5495$, $n = 5$) (**e**) and males ($t4 = 0.2939$, $P = 0.7834$, $n = 5$) (**f**); and in SUS females ($t4 = 3.772$, $P = 0.0196$, $n = 5$) (**g**) and males ($t4 = 4.844$, $P = 0.0084$, $n = 5$) (**h**).

**i**–**l**, Left, representative Ca$^{2+}$ trace of NT$^{LS}$ neurons in CTRL mice during social defeat and tail pinches; middle, peri-event plot of NT$^{LS}$ neuron activity 2 s before and 2 s after attack/tail pinch; right, statistics of neuron activity 2 s before and 2 s after event in female defeat ($t7 = 6.852$, $P = 0.0002$, $n = 8$) (**i**), male defeat ($t6 = 6.973$, $P = 0.0010$, $n = 7$) (**j**), female tail pinch ($t4 = 3.988$, $P = 0.0163$, $n = 5$) (**k**) and male tail pinch ($t5 = 6.137$, $P = 0.0017$, $n = 6$) (**l**). All data were analysed using paired two-tailed $t$-test. *$P < 0.05$, **$P < 0.01$, ***$P < 0.001$. All data expressed as mean ± s.e.m. Scale bar, 100 μm. The illustrations in **b** were created with BioRender (https://biorender.com).

Extended Data Fig. 5a), confirming that CSDS specifically increases the excitability of these cells in SUS mice.

To further investigate NT$^{LS}$ neuron activity in vivo during social encounters with juveniles, we injected Cre-dependent adeno-associated virus (AAV)-DIO-GCaMP6s into the LS of *Nt*-Cre transgenic mice to label NT$^{LS}$ neurons (Fig. 3a). We then measured fluorescent Ca$^{2+}$ activity by fibre photometry (FP) in CTRL, RES and SUS mice during juvenile RI (Fig. 3b and Extended Data Fig. 5b). We found no increase in NT$^{LS}$ neuron activity in CTRL (Fig. 3c,d) and RES (Fig. 3e,f) mice in response to juvenile approach, but SUS mice exhibited significantly higher activity (Fig. 3g,h). Surprisingly, the magnitude (approximately 5–10% change in fluorescence ($\Delta F/F$)) of increase in NT$^{LS}$ neuron activity during juvenile

approach was similar to that observed when unstressed CTRL mice encountered an aversive experience, such as coming under attack by an aggressive CD-1 mouse (Fig. 3i,j) or experiencing a painful investigator-administered tail pinch (Fig. 3k,l). Moreover, NT$^{LS}$ neuron activity showed no change following palatable food consumption (Extended Data Fig. 5c). These findings are consistent with the idea that NT$^{LS}$ neurons respond to aversive, but not to rewarding, stimuli. We further tested NT$^{LS}$ neuron activity during sCPP conditioning and found that NT$^{LS}$ neurons in SUS mice showed higher activity during the juvenile-paired conditioning session, with no changes observed in CTRL or RES mice (Extended Data Fig. 5d). On the basis of these data, we suggest that, following CSDS, SUS mice may overgeneralize social threat

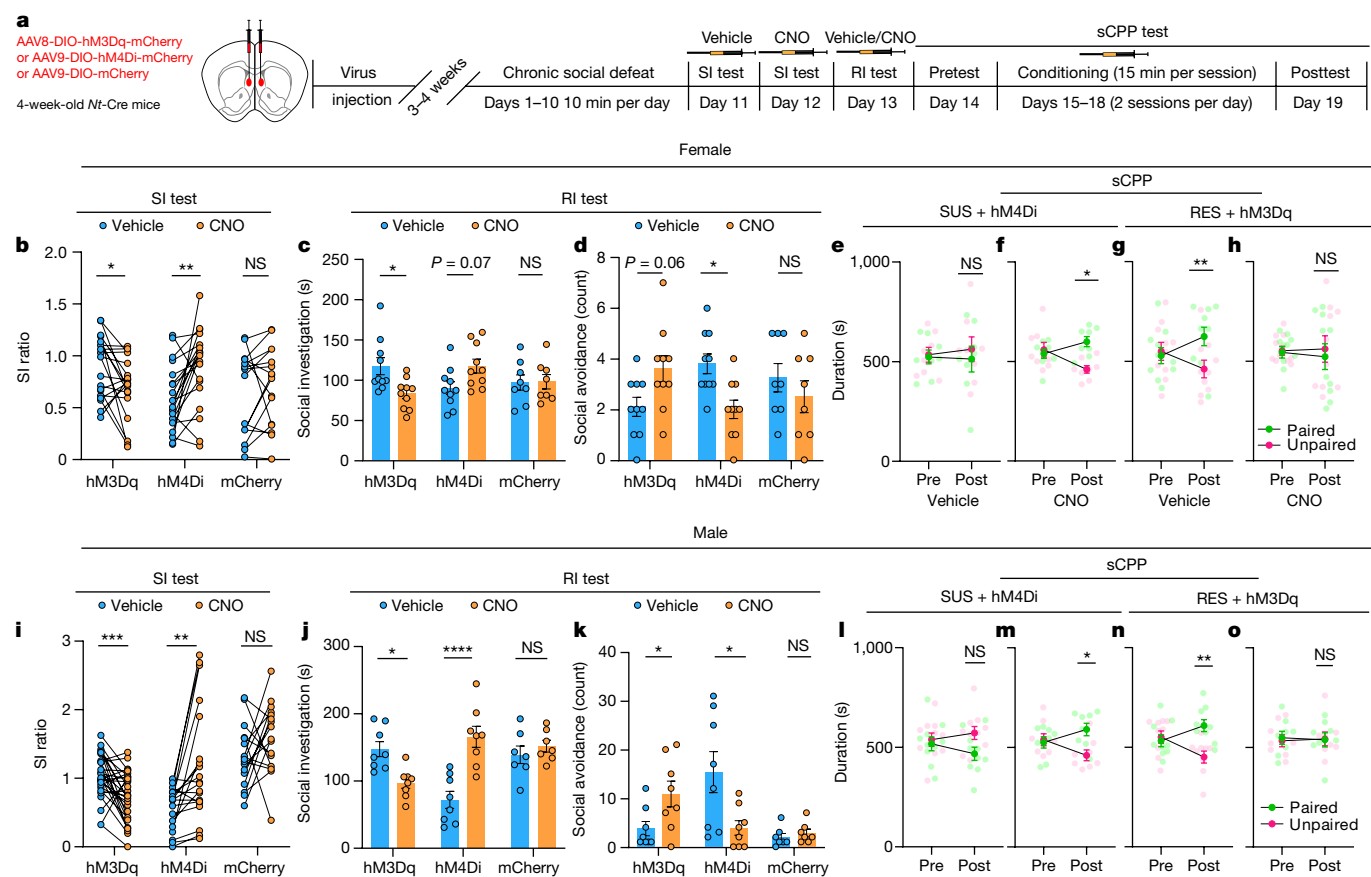

**Fig. 4 | Chemogenetic manipulations of NT$^{LS}$ neurons alter social behaviour following CSDS. a**, Experimental timeline for DREADD experiments. **b–d,i–k**, SI ratio, social investigation and social avoidance following chemogenetic activation (RES mice) or inhibition (SUS mice) of NT$^{LS}$ neurons during social test following CSDS (two-way repeated-measures ANOVA, female: $F_{(2, 53)} = 9.785$, $P = 0.0002$, $n = 18$ (hM3Dq), 20 (hM4Di), 16 (mCherry) (**b**), $F_{(2, 25)} = 5.807$, $P = 0.0085$ (**c**), $F_{(2, 25)} = 5.906$, $P = 0.0079$, $n = 9$ (hM3Dq), 10 (hM4Di), 8 (mCherry) (**d**); male: $F_{(2, 64)} = 12.96$, $P < 0.0001$, $n = 30$ (hM3Dq), 20 (hM4Di), 17 (mCherry) (**i**), $F_{(2, 20)} = 19.46$, $P < 0.0001$ (**j**), $F_{(2, 20)} = 10.12$, $P = 0.0009$, $n = 8$ (hM3Dq), 8 (hM4Di), 7 (mCherry) (**k**). **e–h,l–o**, Social

preference rescued by inhibition of NT$^{LS}$ neurons in female SUS mice (two-way repeated-measures ANOVA, CNO (**e**), $F_{(1, 14)} = 7.272$, $P = 0.0174$, $n = 8$; vehicle (**f**), $F_{(1, 14)} = 0.3070$, $P = 0.5883$, $n = 8$); and in male SUS mice (CNO (**l**), $F_{(1, 14)} = 4.710$, $P = 0.0477$, $n = 8$; vehicle (**m**), $F_{(1, 18)} = 1.627$, $P = 0.2183$, $n = 10$). Activation of NT$^{LS}$ populations in RES females (two-way repeated-measures ANOVA, CNO (**g**), $F_{(1, 18)} = 0.1653$, $P = 0.6891$, $n = 10$; vehicle (**h**), $F_{(1, 18)} = 8.490$, $P = 0.0093$, $n = 10$); and in RES males (CNO (**n**), $F_{(1, 14)} = 0.2221$, $P = 0.6447$, $n = 8$; vehicle (**o**), $F_{(1, 16)} = 9.283$, $P = 0.0077$, $n = 9$) blocked social CPP formation. *$P < 0.05$, **$P < 0.01$, ***$P < 0.001$, ****$P < 0.0001$. All data expressed as mean ± s.e.m.

cues and perceive juveniles as social threats, similar to that observed when being attacked by a highly aggressive CD-1 mouse.

## NT$^{LS}$ neurons modulate social behaviour

To assess whether NT$^{LS}$ neurons regulate social behaviours following CSDS, we utilized viral vectors expressing designer receptors exclusively activated by designer drugs (DREADDs) to bidirectionally manipulate the activity of NT$^{LS}$ neurons during SI with a CD-1 mouse, and also during juvenile RI and sCPP. About 3–4 weeks before CSDS, we injected AAV-DIO-hM3Dq, AAV-DIO-hM4Di or AAV-DIO-mCherry viruses into the LS of 4-week-old *Nt*-Cre mice (Fig. 4a and Extended Data Fig. 6a). Mice were randomly assigned to CTRL or CSDS conditions. For inhibition studies with hM4Di to show necessity, we used only SUS mice whereas for activation studies aimed at showing sufficiency we used only RES mice (note: different baseline SI for vehicle-treated SUS hM4Di versus RES hM3Dq mice in Fig. 4). Testing was performed using a within-subjects design in which mice were first tested for SI 30 min after vehicle injection; then, 30 min before the second SI, mice were injected intraperitoneally with clozapine N-oxide (CNO). We found bidirectional effects of NT$^{LS}$ neuron modulation on SI in both females and males, with increased activity reducing SI in RES mice

and decreased activity enhancing SI in SUS mice (Fig. 4b,i). Mice were then split into two groups for RI, ensuring that the SI ratio was balanced across groups; mice received either vehicle or CNO during the RI test. Inhibition of NT$^{LS}$ neurons increased social investigation time and normalized avoidance behaviour in both sexes (Fig. 4c,d,j,k). For sCPP, we treated hM4Di-injected SUS mice and hM3Dq-injected RES mice with vehicle or CNO during social conditioning sessions. We found that, by inhibiting NT$^{LS}$ neurons in SUS mice, we could normalize preference for the social conditioned compartment to CTRL or RES levels in both sexes (Fig. 4e,f,l,m). Conversely, by activating NT$^{LS}$ neurons in RES mice, we were able to reduce social investigation and social preference compared with their vehicle-treated controls in both sexes (Fig. 4g,h,n,o). Thus, we find that activation of NT$^{LS}$ neurons resulting from social trauma is both necessary and sufficient to elicit social behaviour deficits. Interestingly, activation of NT$^{LS}$ neurons in stress-naïve mice affected neither SI ratio nor sCPP (Extended Data Fig. 6b–d), which suggests that a history of social trauma is critical. In line with this, we find no effect of non-social stressors, such as CVS, on social reward (Extended Data Fig. 6e,f), despite the fact that both CSDS and CVS similarly reduce preferences for natural rewards like sucrose[27]. Consistent with this, a recent study showed that ventral CA3 neurons projecting to the LS play a role in acute social stress-induced

avoidance[28], but not in unstressed mice[29]. To test whether NT[LS] neurons can more generally regulate reward or aversion behaviour, we utilized a real-time place preference assay (RTPP) in stress-naïve mice and found no effect of optogenetic stimulation of NT[LS] neurons on preference (Extended Data Fig. 6g,h). Taken together, these data support the idea that NT[LS] circuits modulate social behaviours in a context-dependent fashion.

To determine whether NT[LS] neurons encode context-specific information related to non-social stressors, we exposed WT mice to chronic restraint stress (CRS) and then performed an interaction test with a new restraint tube. We found that mice exposed to CRS had a longer latency to approach the tube and reduced time spent investigating the tube (Extended Data Fig. 7a,b). We then silenced NT[LS] neurons with an inhibitory DREADD and found that this partially rescued tube avoidance (Extended Data Fig. 7c,d). In a separate group, we paired WT mice with juvenile bedding/odour during CRS (CRSO) and found no differences in the juvenile RI test, suggesting that mice do not generalize avoidance to a juvenile social target based on exposure to these olfactory cues (Extended Data Fig. 7e,f). Overall, these data suggest that NT[LS] neurons are involved in more general computations that use past information from stressful or threatening situations to guide future behaviours towards cues associated with the same or similarly threatening/stressful situations. Last, we found a role for NT[LS] neurons in mediating anxiety-related behaviours, such as the elevated plus maze (EPM), marble burying test and open field test (OFT) (Extended Data Fig. 7g–j). Together these results highlight the LS as a critical node in the regulation of emotional behaviours, particularly in response to aversive/stressful experiences.

## NT[LS] circuitry regulates social behaviour

The LS contains long-range GABAergic projection neurons[30] and has topographically distributed, wide-range input–output projections[18,31,32]. To determine the output patterns of NT[LS] neurons, we applied multiple viral-mediated anterograde tracing tools. First, we injected AAV-DIO-enhanced yellow fluorescent protein (eYFP) into the LS of *Nt*-Cre mice and imaged eYFP[+] axon terminals throughout the brain (Fig. 5a). We then used HSV-1 (H129ΔTK-TT) for anterograde trans-synaptic tracing[33] to verify which regions form monosynaptic connections with NT[LS] neurons (Fig. 5b and Extended Data Fig. 8a). Interestingly, many of the downstream regions identified, such as the medial-lateral preoptic area (LPO/MPO), nucleus accumbens (NAc), anterior hypothalamic nucleus (AHN), lateral hypothalamus (LH), periaqueductal grey (PAG), medial amygdala (MEA) and supramammillary nucleus (SuM), are all involved in various aspects of social behaviour[34] or conditioned reward[35]. Among these regions, the NAc is involved in social reward[36,37] and stress susceptibility[35,38], and the PAG in social aggression[39], as well as in defensive and escape behaviours[40,41]. Although the AHN plays a role in defensive behaviour[42] and parental behaviour[43], its role in social reward remains unknown. We wanted first to determine whether the same or different NT[LS] neurons project to each of these sites. We injected a Cre-dependent retrograde AAV (rgAAV-DIO) expressing tdTomato into the AHN, NAc or PAG of *Nt*-Cre mice (Extended Data Fig. 8b). In the same mice, rgAAV-DIO-eYFP was injected into the alternate regions and we visualized overlap between tdTomato and eYFP in NT[LS] neurons. We also injected cholera toxin subunit B (CTB) into the NAc (CTB488), AHN (CTB555) and PAG (CTB647) (Extended Data Fig. 8d) and found similar results: AHN/NAc-, AHN/PAG- and NAc/PAG-projecting LS neurons showed little overlap (Extended Data Fig. 8c,e), further confirming that LS neurons projecting to these regions represent mostly separate subpopulations. To investigate the function of these NT[LS] circuits, we injected AAV-DIO-ChR2(H134R) into the LS of 5-week-old *Nt*-Cre mice and implanted ferrules in the NAc, AHN or PAG. Three weeks later, mice underwent a subthreshold CSDS (stCSDS) and social behaviour was tested during a 2-day, 5 min juvenile RI test in which laser on/off order

was counterbalanced (Fig. 5c). Activation of NT[LS]→AHN or NT[LS]→NAc circuits decreased active social investigation time without affecting social avoidance behaviour during passive social bouts initiated by the juvenile (Fig. 5d–i). However, activation of NT[LS]→PAG circuit had no effect on either social investigation time or social avoidance (Fig. 5j–l). To further validate whether manipulation of NT[LS]→AHN or NT[LS]→NAc circuits can bidirectionally modulate social interaction, we injected AAV-DIO-eNpHR3.0 into the LS and then performed CSDS (Extended Data Fig. 9a). We found that inhibition of the NT[LS]→AHN or NT[LS]→NAc circuits increased social investigation and partially decreased social avoidance during the RI test (Extended Data Fig. 9b–e). To test whether these pathways bidirectionally control social preference, we injected either AAV-DIO-ChR2 or AAV-DIO-eNpHR3.0, exposed mice to social defeat stress and then performed optical stimulation of NT[LS]→AHN or NT[LS]→NAc circuits during the social conditioning session. As expected, we found that bidirectional regulation of both pathways affected sCPP (Extended Data Fig. 9f–m). The eNpHR3.0 manipulation seemed to have more subtle effects in general, possibly due to its poor efficacy in presynaptic inhibition. Recently developed G-protein-coupled optogenetic tools[44,45] may provide a more convincing method for long-range presynaptic inhibition in future studies.

To confirm that these projections were monosynaptic and inhibitory, we injected AAV-DIO-ChR2 into the LS of *Nt*-Cre mice and performed ex vivo whole-cell electrophysiology with ChR2-assisted circuit mapping of NT[LS]→NAc and NT[LS]→AHN pathways. Our data show both pathways to be monosynaptic (with TTX), inhibitory (Cs-based internal, clamped at 0 mV) and GABAa-dependent (SR-95531, Gabazine) (Extended Data Fig. 10a,b). We also validated that 15 Hz of blue-light stimulation can reliably evoke NT[LS] neurons (Extended Data Fig. 10c). Because it has been reported that other cell types in the LS can modulate stress behaviours under different conditions, we tested whether non-NT neurons in the LS also play a role in social trauma-induced social deficits by injecting AAV-Flpo and AAV-CreOff-FlpOn-ChR2 viruses into the LS of *Nt*-Cre mice to label non-NT neurons with ChR2 (Extended Data Fig. 10d). We first validated the specificity of this approach using Multiplex ISH (Extended Data Fig. 10e,f), and found very little overlap between ChR2 and NT. We next validated stimulation parameters for ChR2 using slice electrophysiology and confirmed that 15 Hz reliably activated non-NT neurons in the LS (Extended Data Fig. 10g). We then stimulated non-NT neurons in the LS in vivo at 15 Hz during the RI test following CSDS and found no effect on social interaction (Extended Data Fig. 10h). Taken together, these results suggest that the activation of inhibitory NT[LS] projections to the AHN and NAc is both necessary and sufficient to alter social investigation and social preference of mice following traumatic social experience.

## Discussion

Many components of social behaviour, including its rewarding properties, are evolutionarily conserved between humans and rodents[46,47]. Although it is well established that social stress leads to the development of depression, anxiety[48] and post-traumatic stress disorder[38], the neural circuits that mediate the negative consequences of social stress—particularly with regard to social reward—are not well defined. We view preclinical social stress models as imperative to this early-phase work so that we can define potential circuit mechanisms of trauma-impaired social reward to inform future studies in humans.

Utilizing an unbiased approach, we identified the LS as one of the most highly regulated regions activated in both male and female SUS mice in response to a normally rewarding social target, suggesting that it might be a particularly important region in regard to explaining the common social deficits exhibited by both sexes. Detailed analysis of the LS identified a specific population of GABAergic projection neurons expressing the neuropeptide neurotensin. In unstressed mice we found these cells to be responsive during situations of threat, including in

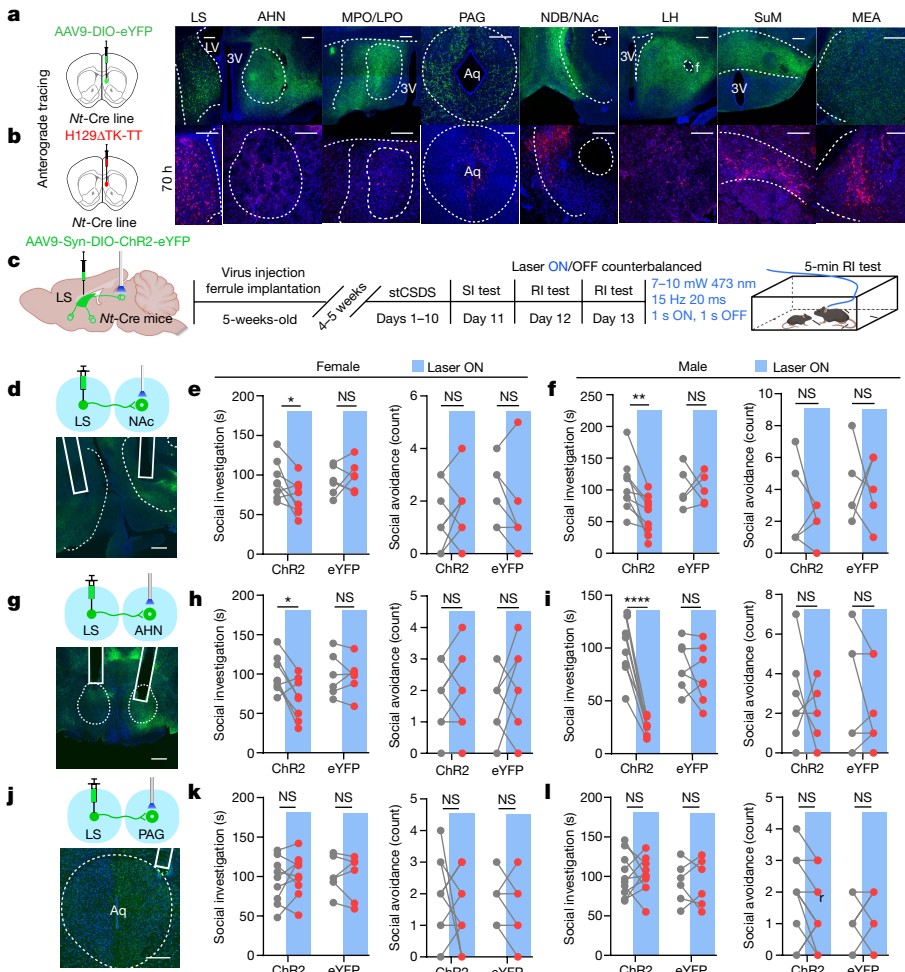

**Fig. 5 | Optogenetic manipulation of NT^LS downstream regions alters social behaviour in CSDS mice. a**, Anterograde AAV-DIO-eYFP tracing from NT^LS neurons. **b**, Anterograde HSV-1 (H129ΔTK-TT) trans-synaptic tracing (70 h post injection) verifies monosynaptic connections between NT^LS neurons and regions shown in a. **c**, AAV-DIO-ChR2 injection and timeline for resident intruder optogenetics experiments. **d–l**, ChR2 axon terminal activation in NAc (**d**), AHN (**g**) and PAG (**j**) during RI test in females (NAc (**e**), social investigation, $F_{(1,12)} = 4.836$, $P = 0.0482$; social avoidance, $F_{(1,12)} = 2.935$, $P = 0.1123$, $n = 8$ (ChR2), 6 (eYFP); AHN (**h**), social investigation, $F_{(1,12)} = 4.947$, $P = 0.0461$, social avoidance, $F_{(1,12)} = 0.8571$, $P = 0.3728$, $n = 8$ (ChR2), 7 (eYFP)); PAG (**k**), social

investigation, $F_{(1,13)} = 0.6986$, $P = 0.4183$; social avoidance, $F_{(1,13)} = 0.07324$, $P = 0.7909$, $n = 8$ (ChR2), 6 (eYFP); and in males (social investigation, NAc (**f**), $F_{(1,13)} = 4.540$, $P = 0.0528$; social avoidance, $F_{(1,13)} = 0.2848$, $P = 0.6026$, $n = 9$ (ChR2), 5 (eYFP); AHN (**i**), social investigation, $F_{(1,13)} = 28.94$, $P = 0.0001$, social avoidance, $F_{(1,13)} = 0.06521$, $P = 0.8024$, $n = 8$ (ChR2), 7 (eYFP); PAG (**l**), social investigation, $F_{(1,14)} = 0.002038$, $P = 0.9646$; social avoidance, $F_{(1,14)} = 1.750$, $P = 0.2071$, $n = 9$ (ChR2), 6 (eYFP) (**f**)). Two-way repeated-measures ANOVA was performed for all comparisons. $*P < 0.05$, $**P < 0.01$, $****P < 0.0001$. All data expressed as mean ± s.e.m. Scale bars, 200 μm. The illustrations in **c** were created with BioRender (https://biorender.com).

response to aggressive attack behaviour. However, following chronic social trauma in SUS mice we found that these neurons generalize their responses to non-threatening social situations, including during interactions with non-aggressive juvenile mice. Notably, NT^LS and *Drd3*^+ neurons exert opposing functions to control social behaviour following stress (Fig. 4 and ref. [26]). *Drd3*^+ and *Nt*^+ neurons are topographically distinct in the LS and it is likely that they have different input/output projection patterns, and possibly even form distinct synaptic connections within the LS. Thus, we hypothesized that NT^LS neurons might play a unique role in regulation of social reward by inhibiting downstream reward centres. Indeed, anterograde tracing studies identified known reward centres—including the NAc and AHN—as receiving moderate/dense innervation from NT^LS neurons, and activation of these inputs reduced social interaction and context-dependent social reward with a juvenile.

Because anxiety is well known to inhibit adaptive social behaviours; one critical question is whether NT^LS neurons are encoding social aversion or whether they simply encode a generalized state of anxiety that impairs social behaviours. According to our data, generalized anxiety

states measured by EPM/OFT are separable from social behaviour deficits: (1) when we stimulate NT^LS neurons in social stress-naïve mice, we are able to produce a generalized exploratory deficit in the EPM/OFT (Extended Data Fig. 7); however, such stimulation does not induce avoidance of a social target (Extended Data Fig. 6b). (2) Both RES and SUS mice in the CSDS model exhibited anxiety-like behaviours in the OFT and EPM, yet only SUS mice exhibited social avoidance and reduced social reward. (3) Although CVS produces an increase in generalized anxiety-like behaviour, it has no effect on social interaction or social reward (Extended Data Fig. 6e,f). Thus, in addition to regulation of generalized anxiety states, NT^LS neurons encode contextual information about stressful/traumatic past experiences to guide future behavioural responses.

Overall our findings demonstrate that, in both male and female SUS mice, rewarding social targets are perceived as stressful or threatening, which engages NT^LS circuitry and impairs social reward processing in a context-dependent manner. Interestingly, in studies of patients with depression and comorbid social anxiety disorder, it was shown that social trauma abnormally increases the representation

of social threat[49]. Furthermore, children who have experienced trauma are reported to exhibit heightened perceptual threat sensitivity, negative and neutral emotion misclassification and attention biases towards threat-related cues[50]. Our research thus provides an important foundation for understanding the neural mechanisms underlying post-trauma social reward processing. Future studies in humans to test the relevance of LS circuitry in mediation of social threat perception and reward sensitivity in victims of trauma will be highly informative.

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

## Methods

### Animals

Wild-type C57BL/6J mice, 7–8 weeks old (males, 22–26 g; females, 18–22 g; Jackson Laboratory) were used as experimental mice in CSDS studies; 4–6-week-old C57BL/6J mice (Jackson Laboratory) were used as new intruders in both the RI test and sCPP test; 16–24-month-old male CD-1 (ICR) mice (sexually experienced retired breeders; Charles River Laboratories) were used as aggressors for male CSDS. $ER\alpha$-Cre mice (017911, B6N.129S6(Cg)-Esr1[tm1.1(cre)And]/J; Jackson laboratory) were crossed with CD-1 females to obtain F1 males, which were used as aggressors for female CSDS. $Nt$-Cre (01752, B6;129-Nts[tm1(cre) Mgmj]/J; Jackson Laboratory) homozygous mice were crossed with WT C57BL/6J mice, and the F1 generation was used as experimental mice in the CSDS studies. Littermates were randomly assigned to experimental groups. All mice were allowed 1 week of acclimation to the housing facilities before the start of experiments. WT CD-1 and F1 $ER\alpha$-Cre mice were single housed, $Nt$-Cre and WT C57BL/6J mice were housed in groups of between three and five. All mice were maintained on a 12/12-h light/dark cycle (07:00–19:00) with ad libitum access to food and water. Housing and experimental rooms were maintained at 20–22 °C and 40–60% humidity. Experiments were conducted during the light phase. Procedures were performed in accordance with the National Institutes of Health Guide for Care and approved by the Use of Laboratory Animals and the Icahn School of Medicine at Mount Sinai Institutional Animal Care and Use Committee. Additional information about mice used in this study can be found in the Life Sciences Reporting Summary.

### Aggressor screening, CSDS and stCSDS

Female[11] and male[10] aggressor screening for CSDS and SI tests was performed as previously described. Experimental males were single housed after CSDS, and females were group housed during CSDS but single housed after defeat. Defeat was halted if an intruder showed any signs of injury. An all-male CSDS lasted 10 min per day for 10 days an all-female CSDS lasted 5 min per day for 10 days; stCSDS lasted for 5 and 2 min per day for males and females, respectively, for 10 days.

### Chronic variable stress

CVS was modified from our previous work[51]. Male and female mice were randomly assigned to CTRL and CVS groups. CVS groups underwent 28 days of stress with one stressor per day, the stressors consisting of 1 h foot shock (random shock 60 times in 1 h), 1 h tail suspension and 1 h restraint.

### Chronic restraint stress

Male mice were randomly assigned to CTRL and CRS groups. The CRS group underwent 28 days of 1 h restraint stress each day. For the juvenile odour-paired CRS, mice were restrained in a 50 mL restrainer and put in a new cage with bedding from a same-sex C57BL/6J juvenile mouse.

### Social interaction test

SI tests were performed 24 h after the last defeat, as described previously[10]. Mice were habituated in the testing rooms for 1 h before testing and all testing was performed under red-light conditions. SI tests were performed with mice freely exploring in a target-free arena (44 cm (w) × 44 cm (d) × 38 cm (h)) for 2.5 min, followed by another 2.5 min target-present (CD-1 or $ER\alpha$-Cre mice) session during which target mice were confined in a wire-mesh enclosure (10 cm (w) × 6.5 cm (d) × 38 cm (h)). The 'interaction zone' of the test arena encompassed a 14 cm × 24 cm rectangular area projecting 8 cm around the wire-mesh enclosure. The 'corner zones' encompassed a 9 cm × 9 cm area projecting from both corner joints opposing the wire-mesh enclosure. We calculated SI ratio as the ratio of time spent in the interaction zone with a CD-1 or F1 $ER\alpha$-Cre mouse present over time spent with the target absent. All mice with a SI ratio over 1 were classified as RES mice and all

with a ratio below 1 as SUS mice. Corner ratio was calculated as the ratio of time spent in the corner zone with an adult CD-1 or F1 $ER\alpha$-Cre target mouse present over time spent when the target mouse was absent.

### Resident intruder test

The RI test was modified from a previously described protocol[52]. After defeat and SI, mice were habituated in the testing rooms for 1 h before testing, and all testing was performed under dim light conditions. Experimental mice were kept in their home cage, placed under an Ethovision camera, habituated for 2–3 min with their wired cage top removed and then intruders (mice or objects) were introduced into their home cage and allowed to interact freely for 5 min (RI test for iDISCO+ cohort lasted for 10 min, to maximally stimulate Fos expression). Social investigation included the amount of active interaction including approach, close following and sniffing. Social avoidance was defined as the escaping from a juvenile mouse of the experimental mouse when approached and investigated by the former. A speed of more than 20 cm s$^{-1}$ was considered an escape. SUS mice typically escaped at speeds of 20–65 cm s$^{-1}$ immediately to avoid social encounters following juveniles' approach/investigation. All experimental mice showing aggressive behaviours towards juveniles (around 1%) were excluded from analyses. All RI behaviours were scored blindly by experimenters.

### Social conditioning place preference

The sCPP protocol, as published previously[53], consisted of three phases: pretest, social conditioning and posttest. Mice were habituated in the testing rooms for 1 h before conditioning or testing. All phases were conducted under red-light and sound-attenuated conditions. The CPP apparatus (Med Associates) has a neutral middle zone that allows for unbiased entry and two conditioning chambers with different walls and floors. On the pretest day, mice were introduced into the middle chamber and allowed to explore freely in all three chambers of the CPP box for 20 min. No group differences in bias for either chamber were found, and conditioning groups were balanced in an unbiased fashion to account for start side preference, as described previously[54]. The conditioning phase consisted of four consecutive days with two conditioning sessions each day: during the morning paired sessions (08:00–12:00), experimental mice were confined to the assigned chamber for 15 min with a new same-sex juvenile C57BL/6J intruder; during the afternoon unpaired session (13:00–17:00) mice were put into the opposite chamber without a social target for 15 min. For female sCPP, during conditioning the juvenile mice were confined in a wire-mesh cup, which we found was necessary for females to form CPP, whereas males formed a preference only when they were able to freely interact with the juvenile outside the cup. All groups were counterbalanced for conditioning chamber. On posttest day, experimental mice were placed in the middle chamber of the CPP apparatus and allowed to freely explore all chambers for 20 min. Duration spent within either context was used to measure CPP. For chemogenetics experiments, CNO was administered during the full conditioning sessions. Behavioural analysis of sCPP data was performed by assessing (1) subtracted CPP (posttest phase duration spent in the intruder-paired chamber minus test phase duration spent in the intruder-unpaired chamber, accounting for test session behaviour only); and (2) group and individual durations in both pretest and posttest sessions.

### Novel-object recognition and object location test

Novel-object recognition (NOR) and object location tests were performed as previously described[55]. Male mice were habituated in the testing room for 1 h before testing and then placed in the middle of an empty plexiglass open field (45 cm (w) × 45 cm (d) × 38 cm (h)) under dim light for 10 min (habituation phase). Twenty minutes after the habituation phase, mice were placed in the same open field with two objects (A and B) and allowed to explore for 10 min. Mice were then

placed back in their home cage for 20 min before being put back into the open field with object B replaced by a new object, C. Mice were allowed to explore for 10 min. Following the NOR test, mice were transferred back to their home cage for 20 min before being returned to the open field, in which the location of object A was changed and the time spent interacting was recorded. Time spent with the new versus familiar object or location was recorded and scored by Ethovision software.

## Elevated plus maze

The EPM was performed as previously described[11]. Male mice were habituated in the testing room for 1 h before testing and then placed in the middle of the plexiglass EPM under red light for 5 min. Each arm of the maze measured $12 \times 50$ cm$^2$. Behaviour was tracked using Noldus Ethovision (Noldus Interactive technologies). Total time spent in the open and closed arms was measured.

## Open field test and locomotor measures

Open field test was performed as previously described[11]. Male mice were habituated in the testing room for 1 h before testing and then placed in the middle of the plexiglass arenas ($44 \times 44 \times 35$ cm$^3$) under red light for 10 min. Behaviour was tracked using Noldus Ethovision (Noldus Interactive technologies) to record the total distance moved, as well as the time spent in the centre ($22 \times 22$ cm$^2$) versus outer zones.

## Marble burying test

The marble-burying test was performed as previously described[56]. Male mice were habituated in the testing room for 1 h before testing. Fresh, unscented mouse corncob bedding (depth 5 cm) was put in standard rat cages (26 cm (w) $\times$ 48 cm (d) $\times$ 20 cm (h)) with filter-top covers, and another cage was inserted onto the surface of the bedding to create parallel lines on the bedding surface that could be used for marble placement. Standard glass toy marbles (1.6 cm diameter) were placed gently on the surface of the bedding in five rows of four. Marbles were washed in 70% ethanol, rinsed in distilled water and dried before each use. Mice were introduced into the corner of the cage to explore for 30 min with the filter-top covered on the cage. A marble was scored as buried if two-thirds of its surface area was covered by bedding. A 2-day, DREADD-manipulated marble-burying test was performed using a within-subjects design; mice were given either CNO or vehicle in a counterbalanced way, and thus they received CNO or vehicle on the first day and the alternative on the second day.

## Real-time place preference

The RTPP experiments was performed as previously described[54]: mice were placed in the centre of an arena (44 cm (w) $\times$ 44 cm (d) $\times$ 35 cm (h)) with a central divider and allowed to explore freely for 20 min. The time spent on each side was recorded using Noldus Ethovision (Noldus Interactive technologies). For the first 10 min of the test, one side of the open field was paired with 20 ms pulses of 15 Hz blue-light stimulation (473 nm, 7–10 mW, 1 s on, 1 s off). For the second 10 min of the test, laser stimulation was paired with the opposite side of the arena; this was done to minimize inherent bias toward one side of the arena. There was a 1-min interval between the two phases. Total time spent in the unstimulated and stimulated sides was calculated and analysed.

## Perfusion and brain tissue processing

For immunohistochemistry and iDISCO+, mice were euthanized by injection of 10% chloral hydrate and perfused transcardially with ice-cold 1× PBS (pH 7.4), followed by fixation with cold 4% paraformaldehyde in 1× PBS. Brains were postfixed for 12 h in the same fixative at 4 °C. For immunohistochemistry, coronal sections were prepared on a vibratome (Leica) at 50 μm to assess viral placement and for immunohistochemistry. For ISH, mouse brains were rapidly removed and flash-frozen in −30 °C isopentane for 5–10 s then kept at −80 °C until sectioning at 15 μm using a cryostat (Leica). Animals injected with AAV

viruses were perfused at least 4 weeks after injection; animals injected with H129ΔTK-TT were perfused 48 and 70 h after injection.

## IHC, ISH and confocal microscopy

For Fos IHC, slices were incubated for 2 h in blocking solution (3% normal donkey serum, 0.3% Triton X-100 in PBS) then incubated overnight in primary antibody (mouse anti-Fos, 1:1,000 (Santa Cruz Biotechnology, C-10)) at 4 °C. Slices were then washed in PBS for 3 × 20 min and incubated in secondary antibody (Cy2 (no. 711-225-152), Cy3 (no. 711-165-152), Cy5 (no. 711-175-152), AffiniPure Donkey Anti-Rabbit IgG (H+L), 1:1,000 (Jackson ImmunoResearch)) for 2 h at room temperature, then washed in PBS for 3 × 20 min before staining with DAPI (1 μg mL$^{-1}$, Sigma) for 20 min. Sections were then mounted with Eco-Mount (Life sciences) and coverslipped (Fisher). For Fos analysis, magnification of ×20 and tile-scan function were used to acquire the entire region of interest. Analysis of Fos-positive cells was performed using Fiji (NIH)[57]. For representative images of viral infection, images were acquired at ×10 magnification using the tile-scan function. For ISH, RNAscope Multiplex Fluorescent Kits (Advanced Cell Diagnostics) were used according to the manufacturer's instructions. Briefly, fresh-frozen brains were slide mounted at 15 μm thickness, fixed for 15 min in cold 4% PFA and dehydrated serially with 50, 70 and 100% EtOH/H$_2$O for 2 min each, followed by 20 min Protease IV (RNAscope) treatment. Proprietary probes (Advanced Cell Diagnostics) for Fos (316921, accession no. NM_010234.2); *Sst* (404631-C2, accession no. NM_009215.1), *Gad67* (400951-C2, accession no. NM_008077.4), *Oxtr* (412171-C2, accession no. NM_001081147.1), *Drd3* (447721-C, accession no. NM_007877.1) or *Crhr2* (413201-C2, accession no. NM_009953.3); *Nt* (420441-C3, accession no. NM_024435.2) were hybridized at 40 °C for 2 h then subjected to a series of amplification steps at 40 °C (1-FL, 30 min; 2-FL, 15 min; 3-FL, 30 min; 4-FL, 15 min). Reagent Alt-A was used for the fourth amplification step, with Channel 1 at 488 nm, Channel 2 at 550 nm and Channel 3 at 647 nm. Finally, slides were treated for 1 min with DAPI and immediately coverslipped with Eco-Mount. All slices were imaged using a Zeiss LSM 780 confocal microscope. Cells and Fos from all ISH and IHC images were counted blindly across groups.

## iDISCO+ staining, imaging and ClearMap analysis

The iDISCO+ staining protocol was modified from http://www.idisco.info. Fixed whole brains were incubated with the primary Fos antibody (no. 226003, 1:1,000, Synaptic Systems) and secondary donkey anti-rabbit IgG (H+L) Highly Cross-Adsorbed Secondary Antibody, Alexa Fluor 647 (no. A-31573, 1:1,000, Thermo Fisher Scientific) for 7 days each. A LaVision lightsheet microscope with zoom body was used for half-brain sagittal scanning, with dynamic focus and a step size of 4 um. Cleared brains were processed as previously described using ClearMap[12]. Fos$^+$ cells were quantified using the cell detection module, with cell detection parameters optimized and validated based on the intensity and shape parameters of the signal. The autofluorescence channel was aligned to the Allen Institute's Common Coordinate Framework using the Elastix toolbox. Brain areas were collapsed into their parent region (for example, the rostroventral, caudodorsal and ventral lateral septum were combined into the 'lateral septal nucleus'). These decisions were made before analysis. To compare cell counts in RES and SUS animals, a negative binomial regression was applied using the glm.nb function from the MASS package in R. Group classifications were dummy coded (0 for the SUS group and 1 for the RES group). The maximum-likelihood coefficients $\alpha$ and $\beta$ were determined through iterative reweighted least squares. A significant $\beta$ means that group status is related to cell count number at the specified region of interest. The z-values in Extended Data Fig. 2i correspond to this $\beta$ coefficient, normalized by its sample standard deviation. $P$ values were corrected for multiple comparisons using the Benjamini–Hochberg procedure to decrease false discovery rate. $Q$-values below 0.05 were considered significant.

## Stereotaxic surgery and viral gene transfer

*Nt*-Cre mice (4–5 weeks old) were anaesthetized by intraperitoneal injection with a mixture of ketamine HCl (100 mg kg$^{-1}$) and xylazine (10 mg kg$^{-1}$) and positioned on a stereotaxic instrument (David Kopf Instruments). In the LS (from bregma: AP +0.7 mm; ML ±0.4 mm; DV –3.0 mm), 0.5 μL of virus was bilaterally infused using 33-gauge Hamilton needles over 5 min, with needles left in place for 5 min after injection. For DREADD virus delivery, 0.5 μl of AAV8-hSyn-DIO-hM3D-mCherry (2.0 × 10$^{12}$ vg mL$^{-1}$, no. 44361-AAV8, Addgene), AAV9-hSyn-DIO-hM4D-mCherry (2.0 × 10$^{12}$ vg mL$^{-1}$, no. 44362-AAV9, Addgene) or AAV9-hSyn-DIO-mCherry (2.0 × 10$^{12}$ vg mL$^{-1}$, no. 50459-AAV9, Addgene) was injected into the LS. For anterograde tracing, 0.5 μL of AAV9-hSyn-DIO-EYFP (2.0 × 10$^{12}$ vg mL$^{-1}$, no. 50457-AAV9, Addgene) or 0.15 μl of H129ΔTK-TT (4.0 × 10$^9$ vg mL$^{-1}$, Center for Neuroanatomy with Neurotropic Viruses) was injected unilaterally into the LS. For retrograde tracing of LS downstream regions, 0.5 μL of retrograde AAV-DIO-EGFP/tdTomato (2.0 × 10$^{12}$ vg mL$^{-1}$, nos. 50457-AAVrg and 28306-AAVrg, Addgene) was injected into the medial part of the NAc (from bregma: AP +1.5 mm; ML ±0.5 mm; DV −4.4 mm), AHN (from bregma: AP −0.7 mm; ML ±0.5 mm; DV −5.0 mm) or PAG (from bregma: AP −4.2 mm; ML ±0.2 mm; DV −2.5 mm). For CTB injection, 0.5 μL of Alexa Fluor 488-conjugated Cholera Toxin Subunit B (1.0 mg mL$^{-1}$, no. C-34775, Thermo Fisher) was injected into the NAc (from bregma: AP +1.5 mm; ML ±0.5 mm; DV −4.4 mm), 0.5 μL of Alexa Fluor 555-conjugated Cholera Toxin Subunit B (1.0 mg mL$^{-1}$, no. C-34776, Thermo Fisher) was injected into the AHN (from bregma: AP −0.7 mm; ML ±0.5 mm; DV −5.0 mm) and 0.5 μL of Alexa Fluor 647-conjugated Cholera Toxin Subunit B (1.0 mg mL$^{-1}$, no. C-34778, Thermo Fisher) was injected into the PAG (from bregma: AP −4.2 mm; ML ±0.2 mm; DV −2.5 mm). For optogenetics, 0.5 μL of AAV9-EF1a-DIO-EYFP (3.0 × 10$^{12}$ vg mL$^{-1}$, no. 27056-AAV9, Addgene), AAV9-Ef1a-DIO eNpHR3.0-EYFP (3.0 × 10$^{12}$ vg mL$^{-1}$, no. 26966-AAV9, Addgene) or AAV9-EF1a-DIO-ChR2-EYFP (3.0 × 10$^{12}$ vg mL$^{-1}$, no. 20298-AAV9, Addgene) was injected into either the LS (cell body stimulation) or downstream regions (terminal stimulation). For Cre-Off virus injection, AAV-EF1a-Flpo (2.0 × 10$^{12}$ vg mL$^{-1}$, no. 55637-AAV1, Addgene) and AAV-nEF-Coff/Fon-ChR2(ET/TC)-EYFP (2.0 × 10$^{12}$ vg mL$^{-1}$, no. 137141-AAV8, Addgene) were mixed 1:1 and injected into the LS. All AAV injections were performed 3 weeks before perfusion or behavioural experiments. For aggressors used in female CSDS, we targeted the VMHvl of *ERα*-Cre F1 mice as described previously[11,58]. For FP, 0.5 μL of AAV9-CAG-FLEX-G6s/EGFP virus (2.0 × 10$^{12}$ vg mL$^{-1}$, no. 100842-AAV9, 51502-AAV9 Addgene) was injected unilaterally into the LS. For optogenetic (ChR2) and FP experiments, cannulae (ChR2: MFC_200/240-0.22_3mm_MF1.25_FLT; FP: MFC_200/250-0.57_3mm_MF1.25_FLT) were implanted at the same time as viral delivery (for LS local, fibres were implanted 0.2 mm above the injection site). For optogenetic (ChR2 and eNpHR3.0) experiments on NT$^{LS}$ terminal stimulation, cannulae (MFC_200/240-0.22_MF1.25_FLT, 5 mm for NAc/AHN, 3 mm for PAG) were implanted into the NAc (from bregma: AP +1.5 mm; ML ±1.5 mm; DV −4.4 mm, 15° angle), the AHN (from bregma: AP −0.7 mm; ML ±1.5 mm; DV −4.8 mm, 10° angle) or PAG (from bregma: AP −4.2 mm; ML ±0.2 mm; DV −2.3 mm). For secure fixture of the optic fibre, dental cement (Grip cement; Dentsply) was added to the skull and around the fibres.

## DREADD manipulation

For *ERα*-Cre mice (used for female CSDS), CNO (1 mg kg$^{-1}$, Tocris) was given intraperitoneally 30 min before CSDS[11]. For OFT, EPM and the marble-burying, SI and RI tests, CNO was given 30 min before the test; for sCPP, CNO was given 30 min before each conditioning session.

## Optogenetics manipulation

For blue (ChR2) and orange (eNpHR3.0) light stimulation, optical fibres (BFP(2)_200/220/900-0.22_4m_FCM-2xMF1.25, Doric Lenses) were connected to either a 473 nm blue laser diode (no. BCL-473-050-M, Crystal Laser) or a 589 nm orange laser diode (no. MGL-III-589-50mW, Opto Engine LLC) using a patch cord with a FC/PC adaptor (no. MFP_200/240/900-0.22_4m_FC-MF1.25, Doric Lenses). A function generator (no. 33220A, Agilent Technologies) was used to generate 20 ms blue-light pulses at 15 Hz, 1 s on/1 s off for all ChR2 experiments. Constant orange light was used for eNpHR3.0 experiments during the 5 min resident intruder test. For sCPP studies, orange light was delivered in a 4 min on/1 min off pattern. The intensity of light delivered to the brain was 7–10 mW. These parameters are consistent with previously validated and published protocols[24]. For all optogenetics tests, experimental mice were habituated to patch cords for 2 days before testing in RI. For RI experiments, mice were tested over 2 days, counterbalanced under laser-on and -off conditions. For social CPP tests, light was provided during the social conditioning session. For the RTPP test, blue-light delivery was controlled by TTL from Noldus Ethovision (Noldus Interactive technologies).

## Ex vivo electrophysiology

AAV9-hSyn-DIO-EYFP (0.5 ul, 2.0 × 10$^{12}$ vg mL$^{-1}$, Addgene) was injected bilaterally into the LS of 4-week-old male *Nt*-Cre mice. Two to three weeks after injection, the mice underwent CSDS. Before slice preparation, all mice were exposed to a 4–6-week-old, same-sex juvenile intruder for 5 min. About 20 min after the RI test, mice were anaesthetized using isoflurane. The brain was rapidly extracted and coronal sections (250 μm) sliced using a Compresstome (no. VF-210-0Z, Precisionary Instruments) in cold (0–4 °C) sucrose-based artificial cerebrospinal fluid (SB-aCSF) containing (in mM): 87 NaCl, 2.5 KCl, 1.25 NaH$_2$PO$_4$, 4 MgCl$_2$, 23 NaHCO$_3$, 75 sucrose and 25 glucose. After 60 min at 32 °C for recovery, slices were maintained in oxygenated (95% CO$_2$/5% O$_2$) aCSF containing (in mM): 130 NaCl, 2.5 KCl, 1.2 NaH$_2$PO$_4$, 2.4 CaCl$_2$, 1.2 MgCl$_2$, 23 NaHCO$_3$ and 11 glucose at room temperature for the remainder of the day, and transferred to a recording chamber continuously perfused at 2–3 mL min$^{-1}$ with oxygenated aCSF. Patch pipettes (4–7 MΩ) were pulled from thin-walled borosilicate glass using a micropipette puller (no. P-97, Sutter Instruments) and filled with a K gluconate (KGlu)-based intrapipette solution containing (in mM): 116 KGlu, 20 HEPES, 0.5 EGTA, 6 KCl, 2 NaCl, 4 ATP and 0.3 GTP and 2 mg mL$^{-1}$ biocytin (pH adjusted to 7.2). Cells were visualized using an upright microscope with an IR-DIC lens and illuminated with a white light source (Scientifica). A 470 nm LED (no. pE-300$^{ultra}$, Cooled) illumination through the microscope objective was used for visualization of eYFP$^+$ cells (using a bandpass filter cube, Olympus). Excitability was measured in current-clamp mode by injection of incremental steps of current (0–100 pA, +10 pA at each step). For recording of optically evoked inhibitory postsynaptic currents (oIPSCs), AAV9-EF1a-DIO-ChR2-eYFP (0.5 μL, 3.0 × 10$^{12}$ vg mL$^{-1}$, Addgene) was injected bilaterally into the LS of 4-week-old male *Nt*-Cre mice. At 5–8 weeks post injection, coronal brain slices of NAc/AHN were prepared as described above and NAc/AHN neurons were recorded in voltage-clamp mode using an internal solution containing (in mM): 120 Cs-methanesulfonate, 10 HEPES, 10 Na-phosphocreatine, 8 NaCl, 5 TEA-Cl, 4 Mg-ATP, 1 QX-314, 0.5 EGTA and 0.4 Na-GTP. NT$^{LS}$ terminals were stimulated through the microscope x40 objective (15 Hz, 5 ms per pulse, 470 nm; no. pE-300$^{ultra}$, CoolLed). oIPSCs were recorded at 0 mV in the presence of tetrodotoxin (TTX, 1 μM, Tocris) to probe monosynaptic effects. oIPSCs were blocked by bath application of gabazine (no. SR-95531, 10 μM, Tocris) confirming the GABAergic nature of the synaptic contact. Whole-cell recordings were performed using a patch-clamp amplifier (Axoclamp 200B, Molecular Devices) connected to a Digidata 1550 LowNoise acquisition system (Molecular Devices). Signals were low-pass filtered (Bessel, 2 kHz) and collected at 10 kHz using the data acquisition software pClamp 11 (Molecular Devices). Electrophysiological recordings were extracted and analysed using Clampfit (Molecular Devices). All groups were counterbalanced by

days after defeat. All recordings were performed blind to experimental condition.

## Fibre photometry

Fibre photometry was performed according to the Neurophotometrics manual and published protocols[59]. A fibre-optic patch cord (no. MFP_200/240/900-0.48_3m_FC-MF1.25, Doric Lenses) was attached to the implanted cannula with cubic zirconia sleeves covered with dark-coloured, shrinkable tubing. The other end of the fibre-optic cable was coupled to a Neurophotometrics LED port. The open-source Bonsai programme was used to control the system; 470 and 415 nm LED lights were used for GCaMP6s signal and autofluorescence measurement. Light at the fibre tip ranged from 40 to 80 μW and was constant across trials over testing days. Simultaneous recording of 40 fps from both 470 and 415 nm channels was achieved phase to phase and visualized via Bonsai. Three weeks after virus injection and ferrule implantation, when mice were around 8 weeks old, they underwent CSDS and SI; they were then habituated to the patch cord for 2 days and Ca²⁺ fluorescence was recorded during the RI test, social CPP conditioning session, stress and food reward tests. Once connected to the apparatus, mice were allowed to rest and habituate for 3–5 min before starting. For the RI test, we recorded Ca²⁺ fluorescence during 2 min of baseline activity without an intruder, followed by 5 min of intruder exposure. The food reward was performed in an open field, and peanut butter cups were placed in the arena close to the corners. All food-biting events were scored manually. MATLAB custom-coding was used for analysis of signal. The 415 nm channel served as the control channel and was subtracted from the GCaMP6s channel to eliminate autofluorescence, bleaching and motion effects. Change in fluorescence ($\Delta F/F$) was calculated as the percentage of mean fluorescence signal of GCaMP6s signal. In general, these motion artefacts had very little effect on overall GCaMP6s signal. Behavioural data were aligned with fluorescence recording data by dividing behavioural video frames with GCaMP6s signal frames. For analysis of LS GCaMP6s activity during discrete behaviours in the RI test, average $\Delta F/F$ (%) in the 2 s before and after a discrete event (passive social investigation) were compared. A passive social investigation was determined to occur at the moment of the intruder-initiated passive social approach.

## Statistical analysis

All statistical details can be found in the figure legends, including type of statistical analysis used, $P$ values, $n$, what $n$ represents, degrees of freedom and $t$ or $F$ values. All $t$-tests, one-way ANOVA and repeated two-way ANOVA were performed using GraphPad Prism software (GraphPad Software Inc.). One-way ANOVA analysis was followed by Tukey's multiple-comparisons test, and two-way repeated-measures ANOVA analysis was followed by Šídák's multiple-comparisons test. Statistically significant differences are indicated in each figure (*$P < 0.05$, **$P < 0.01$, ***$P < 0.001$, ****$P < 0.0001$). For detailed $P$ values please see Source data. Analyses of Fos staining, ISH data and behavioural videos during the RI test were performed blinded to experimental conditions. Sample sizes were chosen according to previous experiments. For Extended Data tables and the Supplementary table, $P$ values were corrected for multiple comparisons using the Benjamini–Hochberg procedure to reduce false discovery rate. $Q$-values below 0.05 were considered significant for all Extended Data tables.

## Statistics and reproducibility

Figure 2c and Extended Data Fig. 3i were repeated in three separate cohorts per sex, with all showing similar results. Figure 5a,b (right) was repeated in three seperate male cohorts ($n = 6$) and in one female cohort ($n = 2$), with all showing similar results. Extended Data Fig. 3j was repeated twice in both sexes, with both showing similar results. Extended Data Fig. 6a was repeated in four separate cohorts in both sexes, with all showing similar results. Extended Data Figs. 8a,d (right) and 10e were repeated twice in males only, with both cohorts showing similar results. Extended Data Fig. 8b was repeated three times, with all showing similar results.

## Image visualization

Brain slice schematics in Figs. 2g,i,j, 3a, 4a and 5a,b and Extended Data Figs. 4d,f,g, 7c, 8a,b,d and 10d were adapted from the Allen Brain Atlas Reference using Adobe Illustrator.

## Reporting summary

Further information on research design is available in the Nature Portfolio Reporting Summary linked to this article.

## Data availability

All raw data for animal behaviours, ISH and IHC are available as Source data files. The Allen Brain Atlas ISH database was used to search for potential molecular markers in LS.

## Code availability

All MATLAB code for Ca²⁺ imaging analysis and Python code for iDISCO+ analysis can be obtained from github (https://github.com/nyclong/2021-07-11642-Nature.git).

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

**Acknowledgements** We thank A. Smith for help with the iDISCO+ protocol and ClearMap analysis, M. Flanigan for advice on validation of the sCPP test, N. Tzavaras, K. Cialowicz and G. Doherty from Mount Sinai Microscopy CoRE for help with lightsheet imaging and N. Rome and S. Gaydos for help with brain slicing. This research was supported by US National Institutes of Health grant nos. R01MH114882, R01MH127820, R01MH104559, R01MH120514 and R01MH120637 (S.J.R.), Brain & Behavior Research Foundation (no. 30233 to L.L., no. 29104 to F.C. and no. 29699 to C.M.), Canadian Institutes of Health Research (no. 201811MFE-414896-231226 to K.L.C.) and NIH Virus Center (grant no. P40 OD010996).

**Author contributions** L.L. and S.J.R. conceived the project. Surgeries were performed by L.L., L.F.P., C.M. and R.D.C. Tissue preparation was performed by L.L., C.J.B., F.C., L.F.P., K.L.C., Y.S. and H.-Y.L. IHC/FISH, microscopy imaging/analysis, fibre photometry data collection and analysis were performed by L.L. Behavioural experiments were performed by L.L. and C.Y. Electrophysiology experiments were performed by R.D.C. iDISCO+ procedures were performed by L.L. iDISCO+ data were analysed by A.V.A. J.W. provided intellectual input and edited the manuscript. Results were analysed and interpreted by L.L., R.D.C., A.V.A. and S.J.R. The manuscript was written by L.L., R.D.C., A.V.A. and S.J.R. and edited by all authors.

**Competing interests** The authors declare no competing interests.

**Additional information**
**Correspondence and requests for materials** should be addressed to Scott J. Russo.

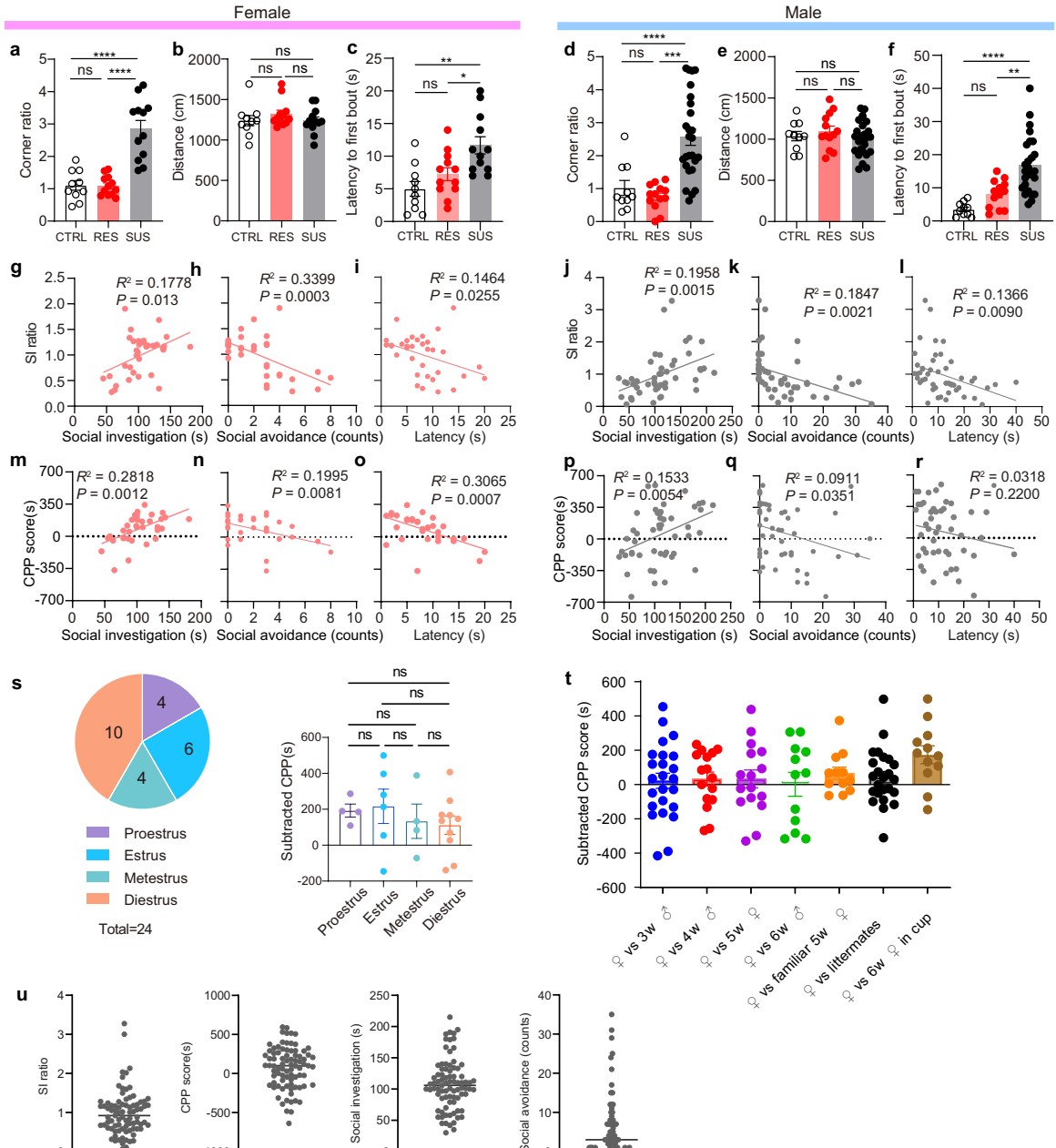

**Extended Data Fig. 1 | Detailed ethological analysis of social behaviour alterations in CSDS animals. a–f,** SUS mice show higher corner ratio (One-Way ANOVA, female, F (2, 31) = 29.62, $P < 0.0001$, $n = 10$ (CTRL), 12 (RES), 12 (SUS) (**a**), males, F (2, 46) = 17.58, $P < 0.0001$, $n = 10$ (CTRL), 13 (RES), 26 (SUS) (**d**)), while showing no locomotor activity deficits (One-Way ANOVA, female, F (2, 31) = 0.8416, $P = 0.4406$ (**b**), males, F (2, 46) = 0.8416, $P = 0.4376$, (**e**)) during SI test and showing longer latency to first social bout (One-Way ANOVA, female, F (2, 31) = 8.399, $P = 0.0012$, (**c**), males, F (2, 46) = 16.63, $P < 0.0001$, (**f**)) during RI test. **g–l,** Correlation between SI ratio and social investigation time (female, $R^2 = 0.1728$, $P = 0.0145$ (**g**), male, $R^2 = 0.1958$, $P = 0.0015$ (**j**)), social avoidance (female, $R^2 = 0.3399$, $P = 0.0003$ (**h**), male, $R^2 = 0.1847$, $P = 0.0021$ (**k**)) and latency to first social bout (female, $R^2 = 0.1464$, $P = 0.0255$ (**i**), male, $R^2 = 0.1366$,

$P = 0.0090$ (**l**)). **m–r,** Correlation between CPP score and social investigation time (female, $R^2 = 0.2818$, $P = 0.0012$ (**m**), male, $R^2 = 0.1533$, $P = 0.0054$ (**p**)), social avoidance (female, $R^2 = 0.1995$, $P = 0.0081$ (**n**), male, $R^2 = 0.0911$, $P = 0.0351$ (**q**)) and latency to first social bout (female, $R^2 = 0.3065$, $P = 0.0007$ (**o**), male, $R^2 = 0.0318$, $P = 0.2200$ (**r**)). **s,** Subtracted CPP score of different stages of female estrus cycle show stage of the cycle has no effect on social CPP formation (One-Way ANOVA, F (3, 20) = 0.5148, $P = 0.6768$, $n = 4$ (Proestrus), 6 (Estrus), 4 (Metestrus), 10 (Diestrus)). **t,** Subtracted CPP score of different social targets for female social CPP. **u,** Distribution of different behaviour parameters of CTRL, RES, SUS animals. ns, not significant. *$P < 0.05$, **$P < 0.01$, ***$P < 0.001$, ****$P < 0.0001$. All data are expressed as mean ± s.e.m.

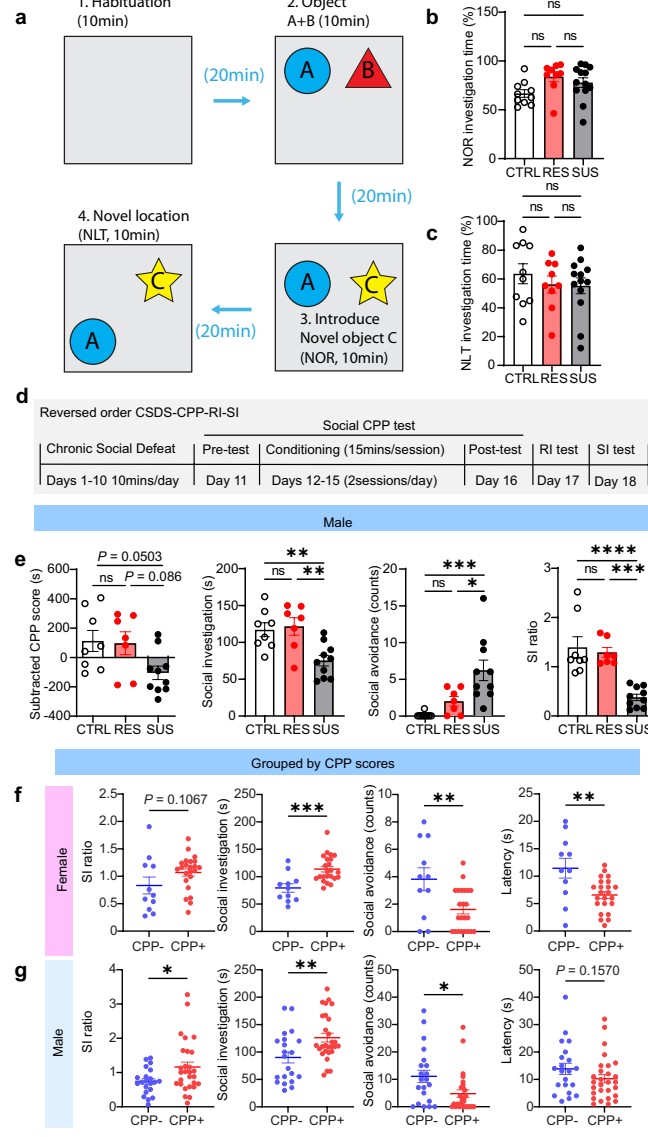

**Extended Data Fig. 2 | Memory tests in animals exposed to CSDS, effects of reversed test order on social behaviours, and behavioural parameters grouped by CPP score. a**, Schematic diagram of novel object recognition (NOR) test and novel location test (NLT). **b**, Novel object investigation time (One-Way ANOVA, F (2, 29) = 3.041, P = 0.0633, n = 10 (CTRL), 9 (RES), 13 (SUS)) and **c**, novel location investigation time (One-Way ANOVA F (2, 29) = 0.5601, P = 0.5772). **d**, Schematic diagram of reversed behaviour test order. **e**, Subtracted CPP score (One-Way ANOVA, F (2, 22) = 3.984, P = 0.0334), social investigation time (F (2, 22) = 8.267, P = 0.0021), social avoidance (F (2, 22) = 9.919, P = 0.0008) and SI ratio (F (2, 22) = 18.32, P < 0.0001, n = 8 (CTRL), 7 (RES), 9 (SUS). **f, g**, Social behavioural parameters after grouping by CPP score, (**f**) females, SI ratio (two-tailed t-test, t = 1.660, df = 32, P = 0.1067), social investigation time (t = 3.788, df = 32, P = 0.0006), social avoidance (t = 3.001, df = 32, P = 0.0052), latency (t = 3.204, df = 32, P = 0.0031), n = 11 (CPP-), 23 (CPP+). (**g**) males, SI ratio (two-tailed t-test, t = 2.298, df = 47, P = 0.0261), social investigation time (t=2.868, df = 47, P = 0.0062), social avoidance (t = 2.545, df = 47, P = 0.0143), latency (t = 1.438, df = 47, P = 0.1570), n = 21 (CPP-), 28 (CPP+). ns, not significant, *P < 0.05, **P < 0.01, ***P < 0.001, ****P < 0.0001. All data are expressed as mean ± s.e.m.

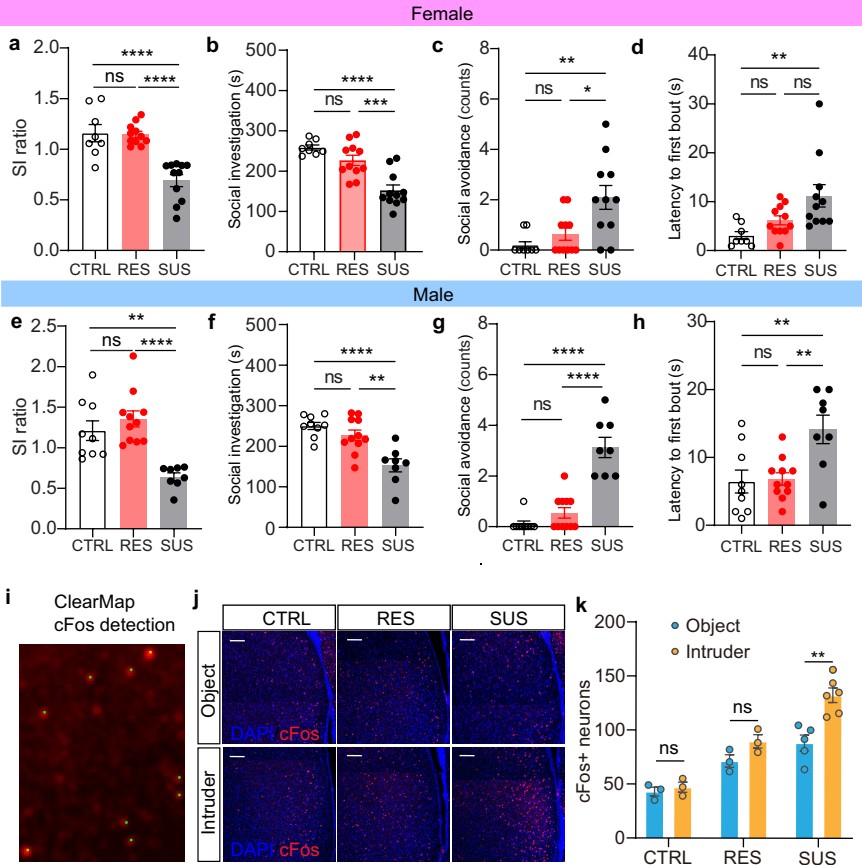

**Extended Data Fig. 3 | Cleared whole brain activity data. a−d**, Female iDISCO+ groups' SI ratio (One-Way ANOVA, F (2, 27) = 22.25, $P$ < 0.0001, $n$ = 8 (CTRL), 11 (RES), 11 (SUS)) (**a**), social investigation time (One-Way ANOVA, F (2, 27) = 20.24, $P$ < 0.0001) (**b**), social avoidance (One-Way ANOVA, F (2, 27) = 7.747, $P$ = 0.0022) (**c**) and latency to first social bout (One-Way ANOVA, F (2, 27) = 6.075, $P$ = 0.0066) (**d**). **e−h**, Male iDISCO+ groups' SI ratio (One-Way ANOVA, F (2, 25) = 13.85, $P$ < 0.0001, $n$ = 9 (CTRL), 11 (RES), 8 (SUS)) (**e**), social investigation time (One-Way ANOVA, F (2, 25) = 14.07, $P$ < 0.0001) (**f**), social avoidance (One-Way ANOVA, F (2, 25) = 38.76, $P$ < 0.0001) (**g**) and latency to first social bout (One-Way ANOVA,

F (2, 25) = 7.370, $P$ = 0.0030) (**h**). **i**, ClearMap cFos detection verification shows annotation (green dots) matching cFos signal (red nuclei). **j**, Brain slice cFos staining verification in males showing most of the cFos[+] cells are in the lateral part of the lateral septum. **k**, Statistics of LS cFos during RI test following CSDS in males (Unpaired two-tailed t-test, CTRL, t4 = 1.434, $P$ = 0.2248, $n$ = 3 per group, RES, t4 = 1.957, $P$ = 0.1219, $n$ = 3 per group, SUS, t9 = 4.429, $P$ = 0.0016, $n$ = 5 (object), 6 (intruder)). ns, not significant, * $P$ < 0.05, ** $P$ < 0.01, *** $P$ < 0.001, **** $P$ < 0.0001. All data are expressed as mean ± s.e.m. Scale bar, 100 μm.

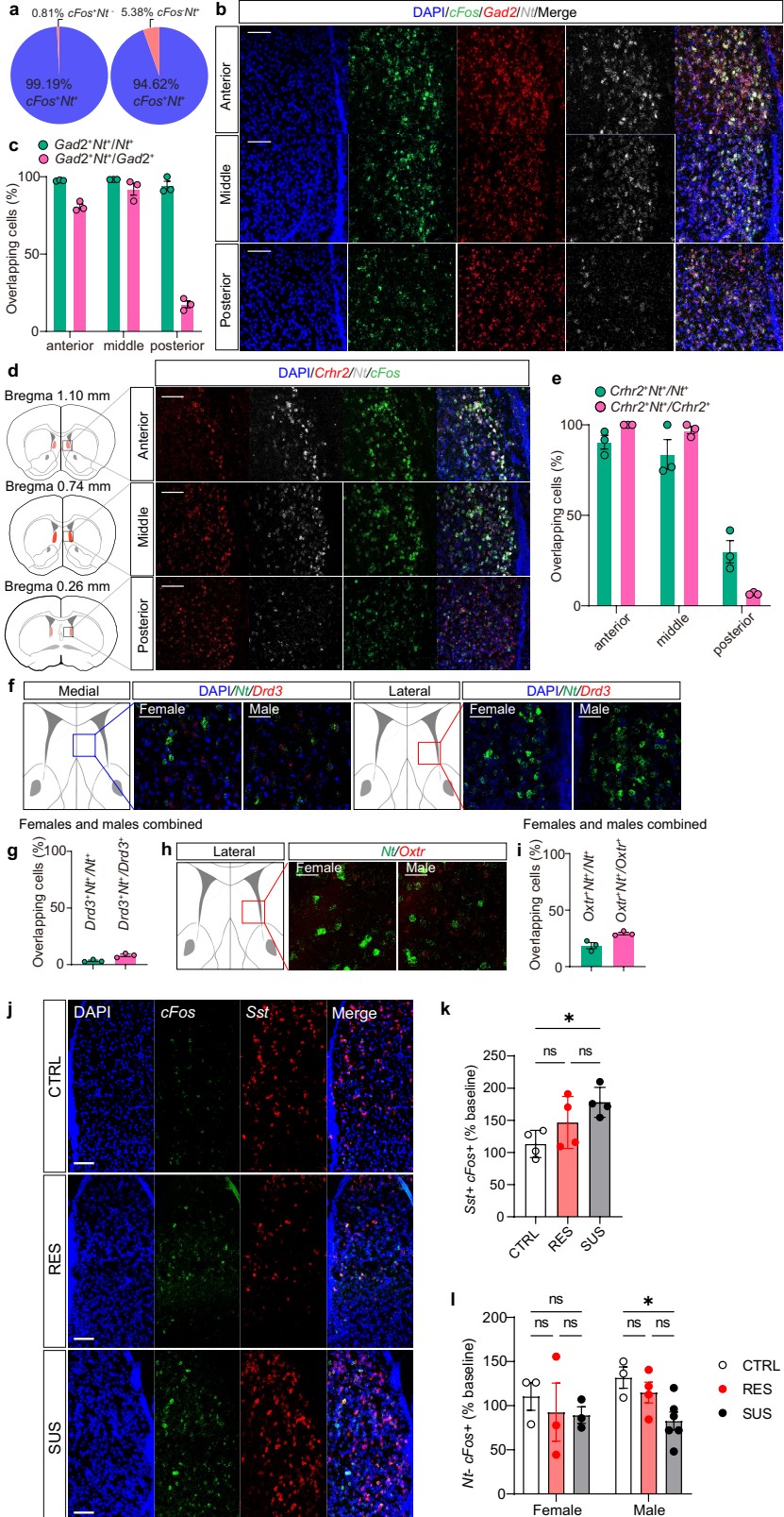

**Extended Data Fig. 4** | See next page for caption.

**Extended Data Fig. 4 | Characterization of co-expression of neurotensin and other genes in lateral septum. a**, Ratio of $Nt^+cFos^+/Nt^+$ (492 of 520 neurons) and $Nt^+cFos^+/cFos^+$ (492 of 496 neurons) in LS of SUS mice. **b**,**c**, Multiplex ISH shows $Nt$ and $Gad2$ colocalization in anterior, middle and posterior LS (**b**) shows most of the neurotensin neurons are GABAergic neurons (**c**). ($n$ = 3 slices per mouse, $n$ = 3 mice per group, scale bar, 100 μm). **d**,**e**, Multiplex ISH shows $Nt$ and $Crhr2$ colocalization in LS (**d**) shows $Nt$ neurons and $Crhr2$ neurons are largely overlapping in anterior and middle parts of the LS, but show very low colocalization in the posterior part of the LS (**e**). ($n$ = 3 slices per mouse, $n$ = 3 mice per group, scale bar, 100 μm). **f**, **g**, Multiplex ISH shows $Nt$ and $Drd3$ do not overlap in the LS ($n$ = 3 slices per mouse, $n$ = 3 mice per group, scale bar, 50 μm). **h**, **i**, Multiplex ISH shows $Nt$ and $Oxtr$ showing very low overlap in the LS ($n$ = 3 slices per mouse, $n$ = 3 mice per group, scale bar, 50 μm). **j**, Confocal images of $Sst$ and cFos expression in CTRL, RES and SUS mice after RI test. **k**, Comparison of Sst$^+$ cFos$^+$ neurons between the three groups (One-Way ANOVA, F (2, 9) = 4.780, $P$ = 0.0385, $n$ = 4 per group, Scale bar, 100 μm). **l**, Comparison of Nt$^-$ cFos$^+$ neurons between female CTRL, RES and SUS mice (One-Way ANOVA, F (2, 6) = 0.2755, $P$ = 0.7683, $n$ = 3 per group) and males (F (2, 10) = 4.980, $P$ = 0.0316, $n$ = 3–6). ns, not significant, $*P$ < 0.05. All data are expressed as mean ± s.e.m.

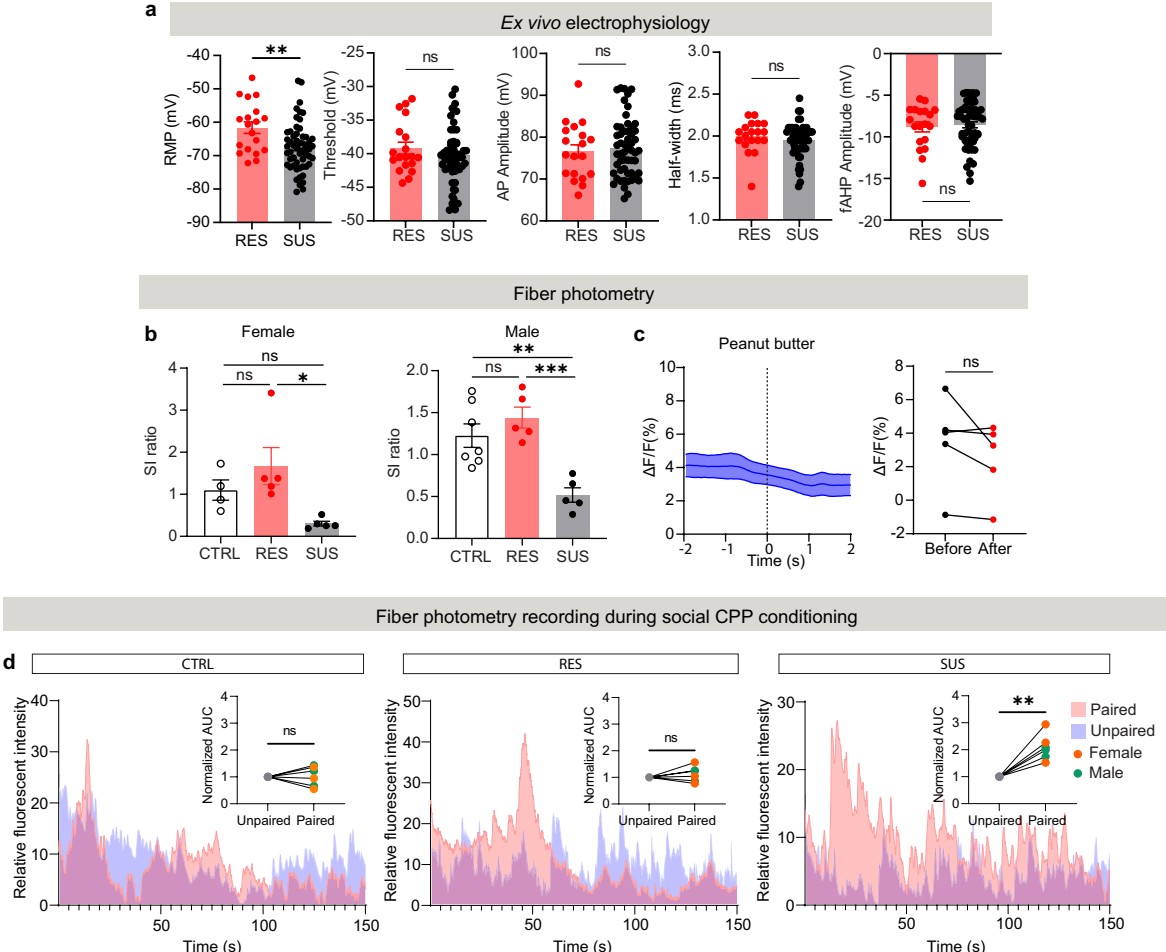

**Extended Data Fig. 5 | Slice electrophysiology following CSDS, measurement of Ca²⁺ activity in NT^LS neurons responses during food reward and social conditioning. a**, Characterization of *ex vivo* electrophysiology parameters measured in NT^LS neurons following CSDS in SUS and RES mice. From left to right: Bar graph of average resting membrane potential per neuron (mean SUS= −67.49 mV +/− 0.975 mV, *n* = 55 with 4–8 neurons/mouse; mean RES = −61.62 mV +/− 1.73 mV, *n* = 19 with 4–7 neurons/mouse; two-tailed Welch's t test, *P* = 0.0059). No difference in action potential threshold (two-tailed Welch's test, *P* = 0.3037), amplitude (two-tailed Welch's test, *P* = 0.6661), half-width duration (two-tailed Welch's test, *P* = 0.3757) or fast after hyperpolarization (fAHP) amplitude (two-tailed Welch's test, *P* = 0.7154). **b**, Social interaction ratio of fiber photometry cohort after CSDS in females

(One-Way ANOVA, F (2, 11) = 5.629, *P* = 0.0207, *n* = 4 (CTRL), 5 (RES), 5 (SUS)) and males (One-Way ANOVA, F (2, 14) = 12.93, P = 0.0007, *n* = 7 (CTRL), 5 (RES), 5 (SUS (5)). **c**, Ca²⁺ activity in NT^LS neurons during consumption of a peanut butter cup (left). Dashed line in **c** marks the beginning of biting. Comparison of Ca²⁺ ΔF/F change before and after biting (right, two-tailed paired t test, t = 1.577, df = 4, *P* = 0.1900, *n* = 5). **d**, Ca²⁺ traces in paired (pink) and unpaired (blue) conditioning chambers. The inset shows normalized area under curve (AUC) between paired and unpaired conditioning sessions (sexes combined, paired two-tailed t-test, CTRL, t = 0.1977, df = 5, *P* = 0.8511, RES, t = 1.049, df = 5, *P* = 0.3423, SUS, t = 5.453, df = 5, *P* = 0.0028), *n* = 6 per group). ns, not significant, *P* < 0.05, ** *P* < 0.01, *** *P* < 0.001. All data are expressed as mean ± s.e.m.

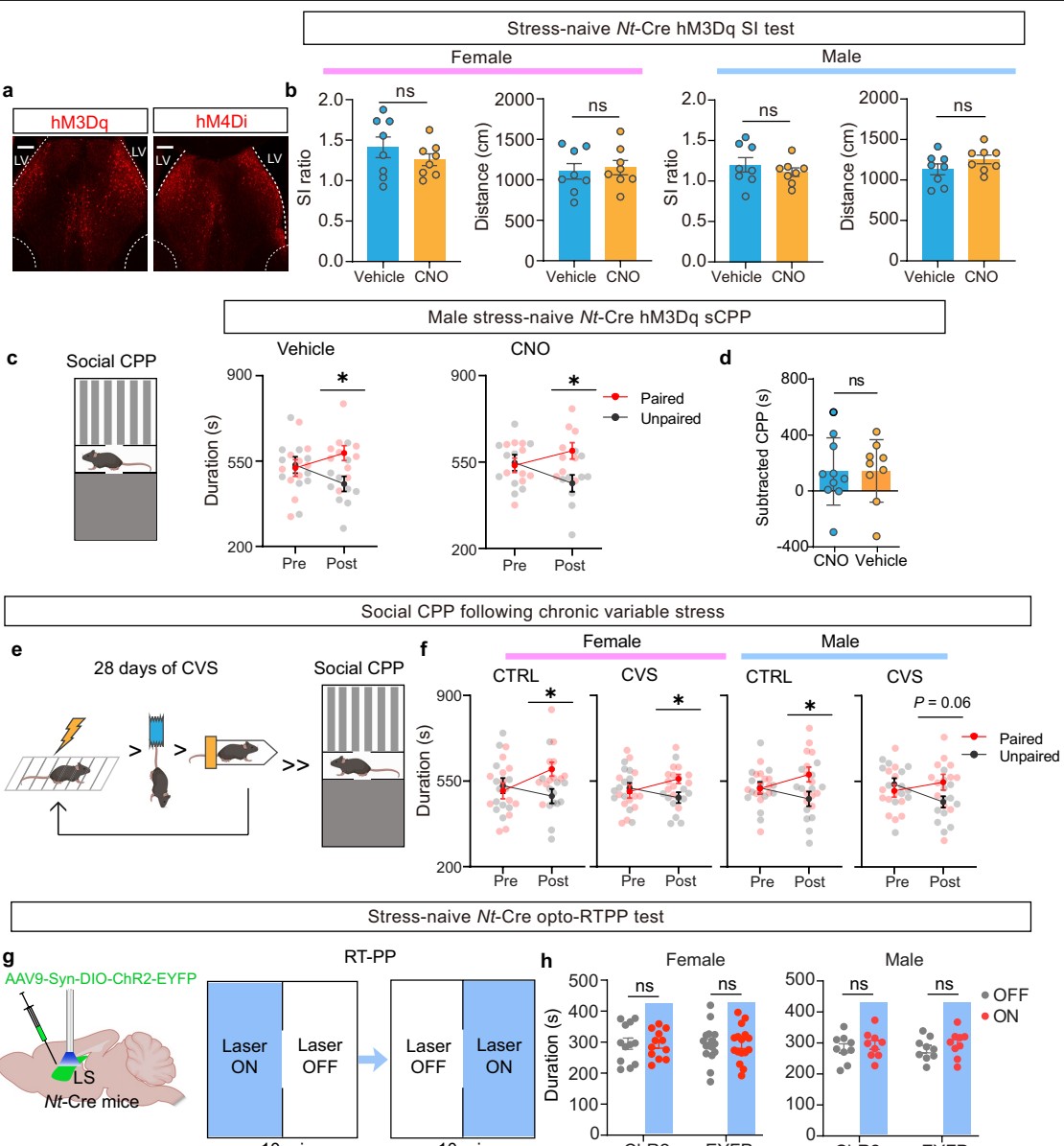

**Extended Data Fig. 6 | NT^LS neuron manipulation in stress-naïve mice does not change social interaction and CVS has no effect on sCPP. a,** Expression of AAV-DIO-DREADDs in NT^LS neurons. **b,** NT^LS activation in stress-naïve mice does not change social behaviour in females (unpaired two-tailed t-test, t14 = 1.044, P = 0.3141, n = 8 per group) or males (unpaired two-tailed t-test, t14 = 1.434, P = 0.3975, n = 8 per group). NT^LS activation in stress-naïve mice does not change locomotor activity in females (unpaired two-tailed t-test, t14 = 0.3473, P = 0.7335, n = 8 per group) or males (unpaired two-tailed t-test, t14 = 1.425, P = 0.1762, n = 8 per group). **c,** NT^LS chemogenetic activation in stress-naïve mice does not change social preference (Two-way repeated measures ANOVA, Vehicle, F = (1, 16) = 7.198, P = 0.0163, n = 9, CNO, F (1, 18) = 6.644, P = 0.0190, n = 10). **d,** Subtracted CPP score comparison between vehicle and CNO group

(unpaired two-tailed t-test, t = 0.03518, df = 17, P = 0.9723, n = 10 per group). **e,** Schematic diagram of CVS and CPP test. **f,** CVS effect on sCPP in either sex (Two-way repeated measures ANOVA, female, CTRL, F (1, 22) = 4.824, P = 0.0389, n = 12, CVS, F (1, 22) = 5.172, P = 0.0331, n = 12; male, CTRL, F (1, 22) = 5.042, P = 0.0351, n = 12; CVS, F (1, 22) = 3.900, P = 0.0610, n = 12). **g,** Schematic diagram of virus injection and optogenetic manipulation of NT^LS neurons during RTPP test. **h,** NT^LS neuron activation does not alter real time place preference in either stress-naïve females (Paired two-tailed t-test, ChR2, t11 = 0.06179, P = 0.9518, n = 12, EYFP, t15 = 0.01923, P = 0.9849, n = 16) or males (Paired two-tailed t-test, ChR2, t8 = 0.3855, P = 0.7099, n = 9; EYFP, t8 = 0.6333, P = 0.5442, n = 9). ns, not significant. * P < 0.05. All data are expressed as mean ± s.e.m. Scale bar, 100 μm. BioRender was used to generate schematic figures in **c,e,g**.

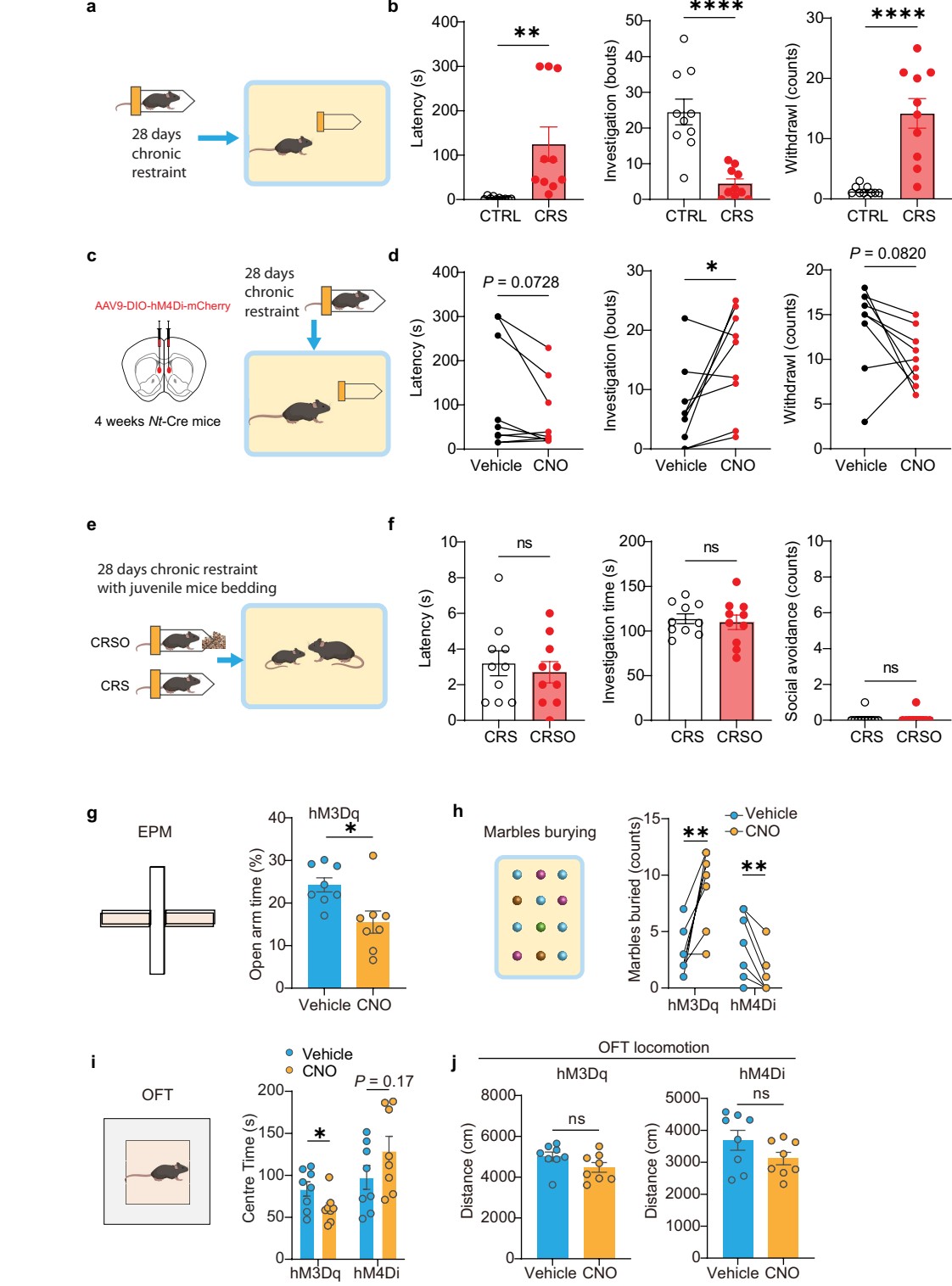

**Extended Data Fig. 7** | See next page for caption.

**Extended Data Fig. 7 | NT$^{LS}$ neurons modulate chronic restraint stress-induced object avoidance and anxiety-like behaviours. a**, Schematic diagram of chronic restraint stress and RI test. **b**, Comparison of latency to the first investigation (unpaired two-tailed t-test, t = 3.142, df = 18, P = 0.0056) and number of investigation bouts (t = 5.259, df = 18, P < 0.0001) with a novel restraint tube, or withdrawal from the tube (t = 5.229, df = 18, P < 0.0001), n = 10, between control (CTRL) and chronic restraint stressed (CRS) animals. **c**, Schematic diagram of CRS with DREADD manipulation during RI test. **d**, Latency (paired two-tailed t-test, t = 2.065, df = 8, P = 0.0728), and number of investigation bouts (t = 2.619, df = 8, P = 0.0307) with a novel restraint tube, or withdrawal from the tube (t = 1.988, df = 8, P = 0.0820), n = 9. **e**, Schematic diagram of chronic restraint stress with juvenile bedding/odour (CRSO) and RI test. **f**, Latency (unpaired two-tailed t-test, t = 0.5452, df = 18, P = 0.5923) and number of investigation bouts (t = 0.3971, df = 18, P = 0.6960) with a novel

restrainer tube, or withdrawal from the tube (t = 0.000, df = 18, P > 0.9999), n = 10. **g**, NT$^{LS}$ activation in stress-naïve male mice leads to higher anxiety-like behaviour in the elevated plus maze (unpaired two-tailed t-test, t14 = 2.824, P = 0.0135, n = 8 per group). **h**, NT$^{LS}$ activation or inhibition in stress-naïve male mice modulate marble burying behaviour (two-tailed paired t-test, hM3Dq, t7 = 4.631, P = 0.0024, n = 8; hM4Di, t7 = 4.020, P = 0.0051, n = 8). **i**, NT$^{LS}$ activation or inhibition in stress-naïve male mice leads to higher or lower anxiety levels, respectively, in the open field test (unpaired two-tailed t-test, hM3Dq, t14 = 2.189, P = 0.0461, n = 8 per group; hM4Di, t14 = 1.424, P = 0.1762, n = 8 per group). **j**, with no locomotor changes (unpaired two-tailed t-test, hM3Dq, t14 = 1.641, P = 0.1230; hM4Di, t14 = 1.566, P = 0.1398). ns, not significant. * P < 0.05, ** P < 0.01, **** P < 0.0001. All data are expressed as mean ± s.e.m. BioRender was used to generate schematic figures in **a**,**c**,**e**,**i**.

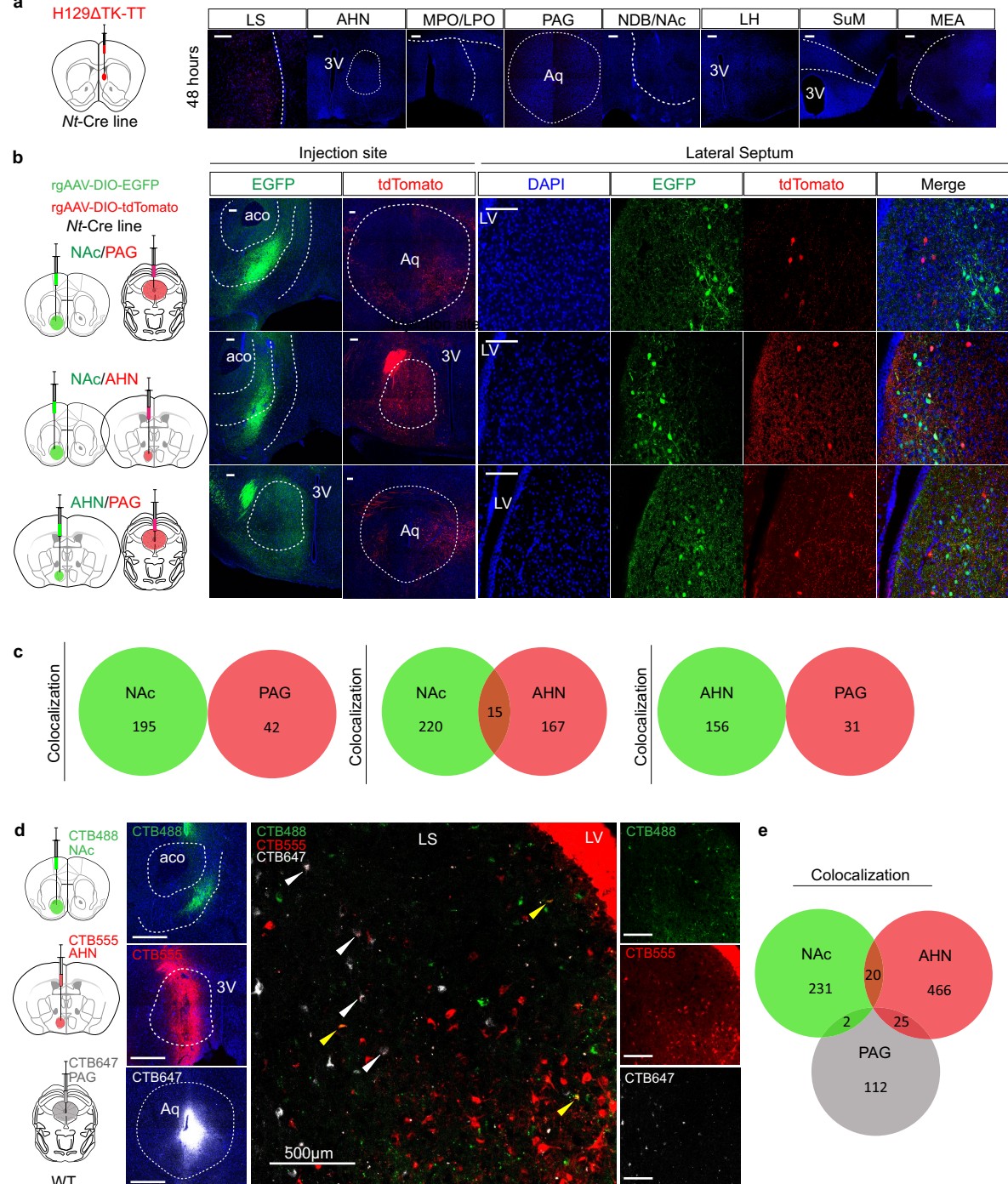

**Extended Data Fig. 8 | NT[LS] monosynaptic downstream region verification.**
**a**, Schematic diagram of H129ΔTK-TT virus injection. 48 h after injection, there were no tdTomato[+] neurons in downstream regions of NT[LS] neurons.
**b**, Retrograde AAV-DIO-EGFP/tdTomato tracing verification shows NT[LS]→NAc, NT[LS]→AHN, NT[LS]→PAG monosynaptic connections. **c**, Colocalization analysis for overlapping NT[LS] projection neurons to the NAc/AHN/PAG. **d**, Cholera toxin subunit b (CTB) tracing for NT[LS]→NAc, NT[LS]→AHN, and NT[LS]→PAG projections.

Representative image of LS (middle panel) with yellow arrowheads indicating neurons projecting to both NAc and AHN. White arrowheads indicate neurons projecting to both PAG and AHN. **e**, Number of projections showing colocalization of CTB tracers from three downstream regions (3 slices per brain region per mouse, *n* = 3 mice). All data are expressed as mean ± s.e.m. Scale bar, 100 μm.

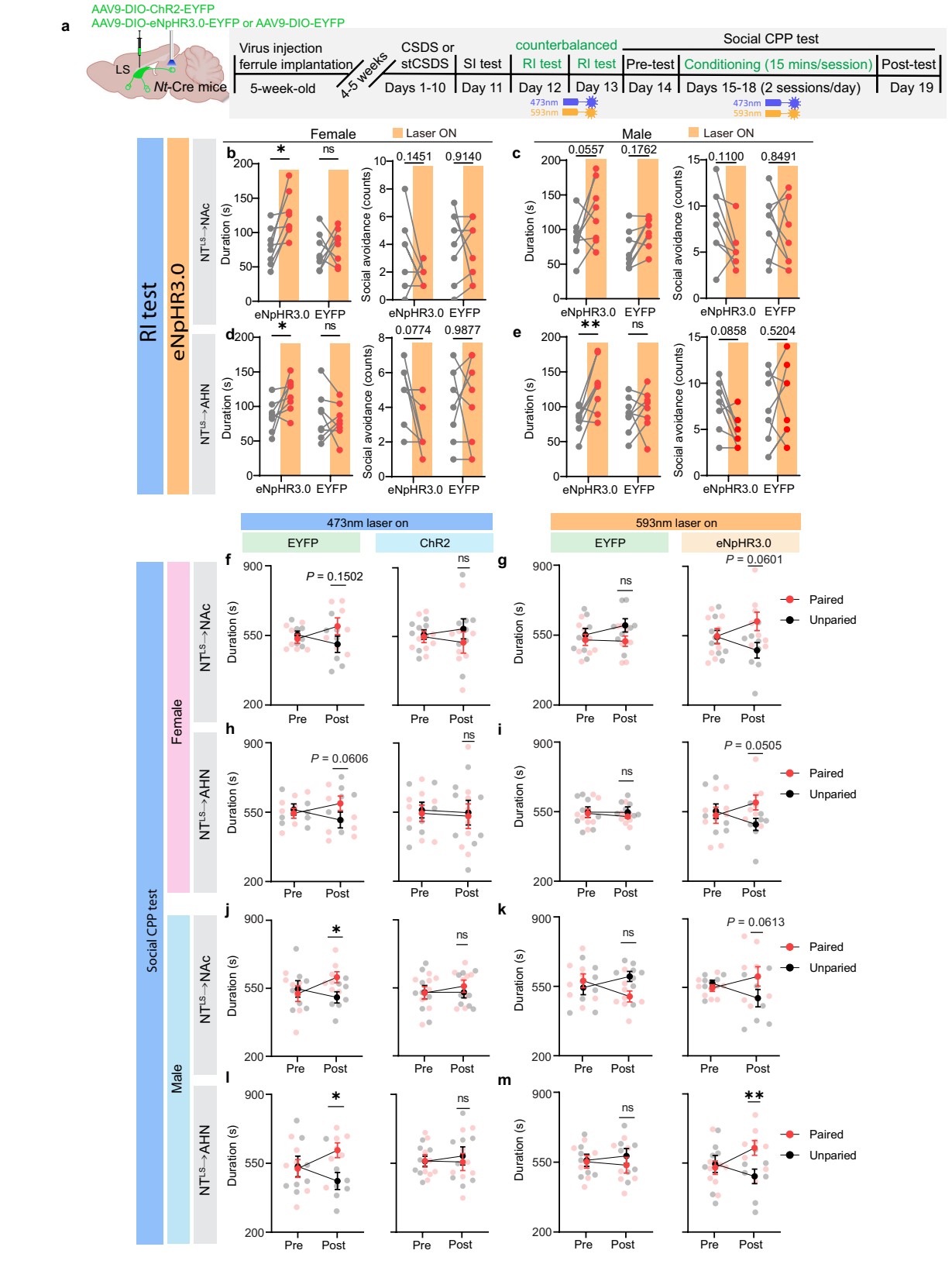

**Extended Data Fig. 9** | See next page for caption.

**Extended Data Fig. 9 | Role of NT$^{LS}$ inputs to NAc and AHN in regulating social behaviours. a**, Schematic of NT$^{LS}$ targeting with NpHR and manipulation during social behaviour tests. **b**–**e**, NpHR axon terminal inhibition in the NAc (**b**, **c**) and AHN (**d**, **e**) rescued social investigation time and partially rescued social avoidance in both females (**b**, social investigation, F (1, 14) = 3.484, $P = 0.0831$, $n = 8$ per group; social avoidance, F (1, 14) = 1.180, $P = 0.2956$, $n = 8$ per group, **d**, social investigation, F (1, 14) = 4.982, $P = 0.0425$, $n = 8$ per group; social avoidance, F (1, 14) = 2.266, $P = 0.1545$, $n = 8$ per group) and males (**c**, social investigation, F (1, 14) = 0.2046, $P = 0.6580$, $n = 8$ per group; social avoidance, F (1, 14) = 1.214, $P = 0.2891$, $n = 8$ per group, **e**, social investigation, F (1, 14) = 4.597, $P = 0.0501$, $n = 8$ per group; social avoidance, F (1, 14) = 5.359, $P = 0.0363$, $n = 8$ per group). **f-m**, Optogenetic manipulation of NT$^{LS}$→NAc and NT$^{LS}$→AHN circuits during social CPP conditioning in both sexes. **f**, Activation of NT$^{LS}$→NAc with ChR2 blocked social reward in RES females (EYFP, F (1, 12) = 2.362, $P = 0.1502$, $n = 7$, ChR2, F (1, 14) = 0.5543, $P = 0.4689$, $n = 8$) **g**, Inhibition of NT$^{LS}$→NAc with NpHR rescued social reward deficits in SUS females (EYFP, F (1, 14) = 0.5105, $P = 0.4867$, NpHR, F (1, 14) = 4.183, $P = 0.0601$, $n = 8$ per group).

**h**, Activation of NT$^{LS}$→AHN with ChR2 blocked social reward in RES females (EYFP, F (1, 12) = 4.289, $P = 0.0606$, $n = 7$, ChR2, F (1, 14) = 0.0001296, $P = 0.9911$, $n = 8$). **i**, Inhibition of NT$^{LS}$→AHN with NpHR rescued social reward deficits in SUS females (EYFP, F (1, 14) = 0.1068, $P = 0.7487$, NpHR, F (1, 14) = 4.575, $P = 0.0505$, $n = 8$ per group). **j**, Activation of NT$^{LS}$→NAc with ChR2 blocked social reward in RES males (EYFP, F (1, 12) = 5.703, $P = 0.0343$, $n = 7$, ChR2, F (1, 14) = 0.2484, $P = 0.6259$, $n = 8$). **k**, Inhibition of NT$^{LS}$→NAc with NpHR rescued social reward deficits in SUS males (EYFP, F (1, 14) = 4.053, $P = 0.0637$, NpHR, F (1, 14) = 4.140, $P = 0.0613$, $n = 8$ per group). **l**, Activation of NT$^{LS}$→AHN with ChR2 blocked social reward in RES males (EYFP, F (1, 12) = 6.329, $P = 0.0271$, $n = 7$, ChR2, F (1, 14) = 0.1567, $P = 0.6981$, $n = 8$). **m**, Inhibition of NT$^{LS}$→AHN with NpHR rescued social reward deficits in SUS males (EYFP, F (1, 12) = 0.6881, $P = 0.4230$, NpHR, F (1, 14) = 10.02, $P = 0.0069$, $n = 8$ per group). Two-way repeated measures ANOVA were performed for any comparison in this figure: ns, not significant. $^{*}P < 0.05$, $^{**}P < 0.01$. All data are expressed as mean ± s.e.m. BioRender was used to generate schematic figures in **a**.

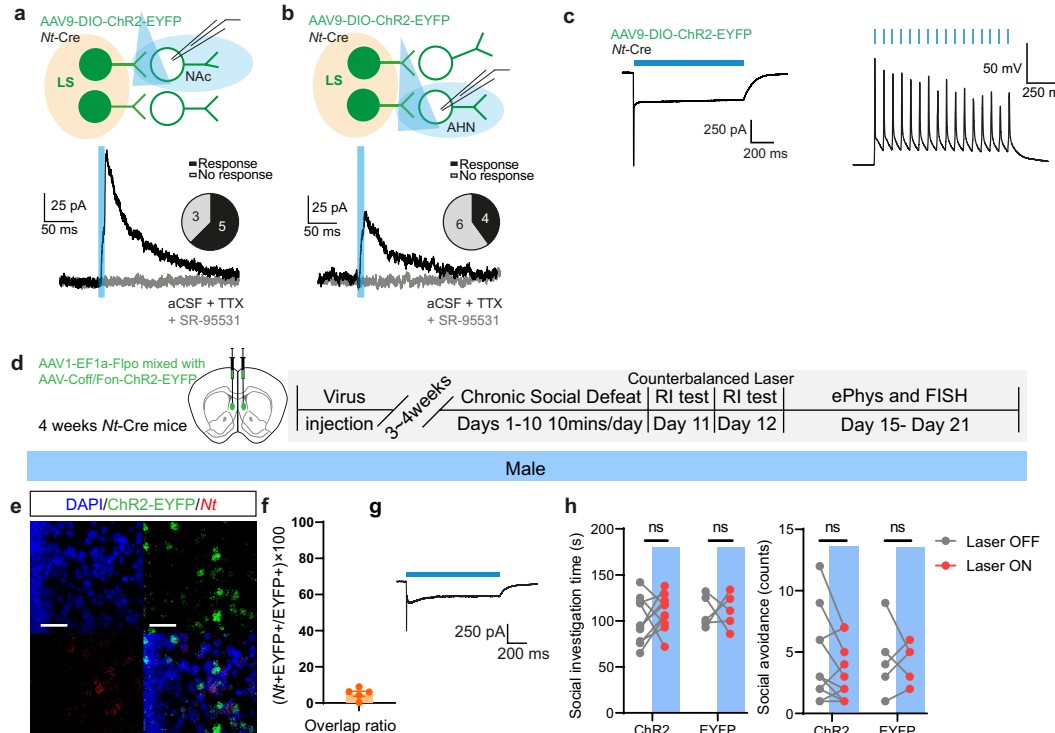

**Extended Data Fig. 10 | Optogenetic validation of NT^LS monosynaptic projections and effects of optogenetic stimulation of non-NT^LS neurons on social behaviour. a,b,** ChR2-assisted circuit mapping of NT^LS→NAc and NT^LS→AHN pathways showing monosynaptic (with TTX), inhibitory (Cs-based internal, clamped at 0 mV) and GABAa-dependent (SR-95531, Gabazine) connections in 5/8 NAc/NDB neurons and 4/10 AHN neurons. **c,** Light-evoked inward current in a ChR2-EYFP^+ Cre^+ neuron in voltage-clamp mode (clamped at −70 mV) upon 1 s illumination with 470 nm blue light (left) and 15 Hz pulse-induced spikes (right). **d,** Schematic diagram of non-NT^LS neuron infection and manipulation during social behaviour tests. **e,f,** Multiplex ISH for *Nt* and CreOff-ChR2-EYFP. **c,** Overlap of Nt^+ EYFP^+ neurons among EYFP^+ neurons (2-3 slices per mouse, *n* = 5 mice). **g,** Light-evoked inward current in ChR2-EYFP^+ non-Cre neurons in voltage-clamp mode (clamped at −70 mV) upon 1 s illumination with blue light (470 nm). **h,** Effect of ChR2 stimulation of non-*Nt* neurons on social investigation and social avoidance during RI test (Mixed-effects analysis two-way ANOVA, left, $F_{(1,13)} = 0.2137$, $P = 0.6516$, right, $F_{(1,13)} = 0.5672$, $P = 0.4648$, n = ChR2 (10), EYFP (5)). ns, not significant. All data are expressed as mean ± s.e.m. Scale bar, 50 μm.

**Extended Data Table 1 | Regions that show significant difference between SUS and RES females (iDISCO+ analysis)**

| name | z score | p value | q value |
|---|---|---|---|
| Claustrum | 3.94954061 | 7.83E-05 | 0.00565596 |
| Subparafascicular nucleus | 3.88740614 | 0.00010132 | 0.00565596 |
| Interanteromedial nucleus of the thalamus | 3.91670706 | 8.98E-05 | 0.00565596 |
| Mediodorsal nucleus of thalamus | 3.85373284 | 0.00011633 | 0.00565596 |
| Parataenial nucleus | 3.9769067 | 6.98E-05 | 0.00565596 |
| Taenia tecta | 3.55268855 | 0.00038132 | 0.01544982 |
| Anterior group of the dorsal thalamus | 3.40087327 | 0.00067171 | 0.02332753 |
| Dorsal peduncular area | 3.27726682 | 0.00104817 | 0.02413187 |
| Lateral septal complex | 3.28483241 | 0.00102043 | 0.02413187 |
| Interanterodorsal nucleus of the thalamus | 3.26570038 | 0.00109194 | 0.02413187 |
| Central medial nucleus of the thalamus | 3.26908486 | 0.00107896 | 0.02413187 |
| Anterior olfactory nucleus | 3.1043635 | 0.00190689 | 0.03863041 |
| Intermediodorsal nucleus of the thalamus | 2.93775546 | 0.00330598 | 0.04840822 |
| Posterior hypothalamic nucleus | 2.95543271 | 0.00312231 | 0.04840822 |
| Pedunculopontine nucleus | 2.94672372 | 0.0032116 | 0.04840822 |
| Parabrachial nucleus | 3.00543928 | 0.00265198 | 0.04840822 |
| Sublaterodorsal nucleus | 2.93040617 | 0.00338519 | 0.04840822 |
| Interpeduncular nucleus | -2.9101501 | 0.00361255 | 0.04878949 |
| Agranular insular area | 2.86009616 | 0.00423513 | 0.04978821 |
| Paraventricular nucleus of the thalamus | 2.86580056 | 0.00415956 | 0.04978821 |
| Dorsal cochlear nucleus | -2.8495233 | 0.00437848 | 0.04978821 |
| Paragigantocellular reticular nucleus | 2.84039861 | 0.00450572 | 0.04978821 |

**Extended Data Table 2 | Regions that show significant difference between SUS and CTRL females (iDISCO+ analysis)**

| name | z score | p value | q value |
|---|---|---|---|
| Parafascicular nucleus | 7.42105509 | 1.16E-13 | 8.2012E-12 |
| Ventral posterolateral nucleus of the thalamus | 6.39662807 | 1.59E-10 | 8.431E-09 |
| Central lateral nucleus of the thalamus | 5.69125366 | 1.26E-08 | 5.3449E-07 |
| Perihypoglossal nuclei | 5.28600805 | 1.25E-07 | 0.00000442 |
| Cuneiform nucleus | 5.23366285 | 1.66E-07 | 0.00000503 |
| Intermediodorsal nucleus of the thalamus | 5.19238992 | 2.08E-07 | 0.00000551 |
| Anterior olfactory nucleus | 5.16280271 | 2.43E-07 | 0.00000573 |
| Dorsal premammillary nucleus | 4.75561176 | 1.98E-06 | 0.000042 |
| Taenia tecta | 4.67501109 | 2.94E-06 | 0.00005669 |
| Ventral premammillary nucleus | 4.56632344 | 4.96E-06 | 0.00008767 |
| Nucleus of reunions | -4.5268604 | 5.99E-06 | 0.00009773 |
| Field CA3 | 4.35660877 | 1.32E-05 | 0.00019998 |
| Suprageniculate nucleus | 4.16412819 | 3.13E-05 | 0.00041492 |
| Precommissural nucleus | 4.16587548 | 3.10E-05 | 0.00041492 |
| Supramammillary nucleus | 4.1405031 | 3.47E-05 | 0.00043293 |
| Posterior hypothalamic nucleus | 4.07556728 | 4.59E-05 | 0.00054086 |
| Medial preoptic area | 3.9833926 | 6.79E-05 | 0.00075798 |
| Ventral anterior-lateral complex of the thalamus | 3.82699384 | 0.0001297 | 0.00137568 |
| Dorsomedial nucleus of the hypothalamus | 3.76381873 | 0.0001673 | 0.00169013 |
| Nucleus ambiguus | 3.69776315 | 0.0002175 | 0.00209699 |
| Diagonal band nucleus | 3.67228799 | 0.0002403 | 0.00221681 |
| Lateral septal complex | 3.63419581 | 0.0002788 | 0.00246434 |
| Gigantocellular reticular nucleus | -3.5972594 | 0.0003215 | 0.00272837 |
| Anterior hypothalamic nucleus | 3.5425464 | 0.0003962 | 0.00323273 |
| Rhomboid nucleus | 3.53120464 | 0.0004136 | 0.00324961 |
| Primary somatosensory area | -3.4415002 | 0.0005785 | 0.00438214 |
| Parastrial nucleus | 3.39977377 | 0.0006744 | 0.00493257 |
| Anteroventral preoptic nucleus | 3.34261297 | 0.0008299 | 0.00586768 |
| Dorsal part of the lateral geniculate complex | 3.29545625 | 0.0009826 | 0.00672302 |
| Retrochiasmatic area | 3.27420523 | 0.0010596 | 0.00702316 |
| Dorsal cochlear | -3.2048784 | 0.0013512 | 0.00868453 |
| Paraventricular hypothalamic nucleus | 3.17340511 | 0.0015066 | 0.00913012 |
| Pedunculopontine nucleus | 3.17565592 | 0.0014949 | 0.00913012 |
| Fasciola cinerea | -3.1466895 | 0.0016513 | 0.00972891 |
| Periaqueductal gray | 3.11353426 | 0.0018486 | 0.01059703 |
| Subiculum | 3.06294968 | 0.0021916 | 0.01223298 |
| Central medial nucleus of the thalamus | -3.0255988 | 0.0024814 | 0.01349505 |
| Sublaterodorsal nucleus | 3.01472366 | 0.0025721 | 0.01363872 |
| Anteroventral periventricular nucleus | 2.9524726 | 0.0031524 | 0.0163079 |
| Mammillary body | 2.93155802 | 0.00337266 | 0.01703193 |
| Rostral linear nucleus raphe | -2.9204887 | 0.0034948 | 0.01723845 |
| Entorhinal area | 2.90364559 | 0.0036884 | 0.01778001 |
| Paracentral nucleus | 2.88682983 | 0.0038914 | 0.0183417 |
| Peripeduncular nucleus | 2.82950556 | 0.004662 | 0.02149587 |
| Lateral preoptic area | 2.79957929 | 0.0051169 | 0.02309146 |
| Tuberomammillary nucleus | 2.77386502 | 0.0055394 | 0.02397799 |
| Parabrachial nucleus | 2.77762365 | 0.0054758 | 0.02397799 |
| Ventral part of the lateral geniculate complex | 2.74955794 | 0.0059675 | 0.02531443 |
| Infracerebellar nucleus | 2.74005674 | 0.0061428 | 0.02554707 |
| Median preoptic nucleus | 2.71344308 | 0.0066588 | 0.02716022 |
| Claustrum | 2.65281184 | 0.00798244 | 0.03135325 |
| Parasubthalamic nucleus | 2.65775917 | 0.00786621 | 0.03135325 |
| Nucleus of the lateral lemniscus | 2.63884012 | 0.00831902 | 0.03208117 |
| Primary somatosensory area | -2.6051921 | 0.00918228 | 0.03477789 |
| Interanteromedial nucleus of the thalamus | 2.58003327 | 0.00987908 | 0.03676058 |
| Posterodorsal preoptic nucleus | 2.53860308 | 0.0111296 | 0.04004074 |
| Posterior pretectal nucleus | 2.53833401 | 0.01113816 | 0.04004074 |
| Posterolateral visual area | 2.49354328 | 0.01264752 | 0.04470898 |
| Primary motor area | -2.4823439 | 0.01305212 | 0.04538286 |
| Medial septal nucleus | 2.46128015 | 0.01384422 | 0.04736063 |

**Extended Data Table 3 | Regions that show significant difference between RES and CTRL females (iDISCO+ analysis)**

| name | z score | p value | q value |
|---|---|---|---|
| Nucleus of the solitary tract | 5.28801676 | 1.24E-07 | 0.00000476 |
| Ventral cochlear nucleus | 4.9116476 | 9.03E-07 | 0.00002971 |
| Gigantocellular reticular nucleus | 4.68681401 | 2.77E-06 | 0.00007973 |
| Intermediate reticular nucleus | 4.52158931 | 6.14E-06 | 0.0001571 |
| Perihypoglossal nuclei | 4.39535501 | 1.11E-05 | 0.00025561 |
| Precommissural nucleus | 3.84378291 | 0.00012115 | 0.00253622 |
| Inferior olivary complex | -3.7799914 | 0.00015683 | 0.00300957 |
| Parasubthalamic nucleus | 3.71578723 | 0.00020257 | 0.00358829 |
| Dorsal motor nucleus of the vagus nerve | -3.4654636 | 0.00052932 | 0.00870656 |
| Primary somatosensory area | -3.3605125 | 0.00077798 | 0.01160539 |
| Ventral premammillary nucleus | 3.3506073 | 0.00080635 | 0.01160539 |
| Primary somatosensory area | -3.3127961 | 0.00092368 | 0.01181695 |
| Spinal nucleus of the trigeminal | 3.32659247 | 0.00087915 | 0.01181695 |
| Orbital area | -3.2855351 | 0.00101789 | 0.01233683 |
| Parastrial nucleus | 3.20911945 | 0.00133142 | 0.01532997 |
| Mammillary body | 3.1911409 | 0.00141712 | 0.01553973 |
| Magnocellular nucleus | 3.16931751 | 0.00152797 | 0.01599368 |
| Primary somatosensory area_bfd | 3.70988163 | 0.00158365 | 0.0185646 |
| Anterior hypothalamic nucleus | 3.10296516 | 0.00191592 | 0.01918252 |
| Thalamus | 3.06199861 | 0.00219864 | 0.02058482 |
| Anterior pretectal nucleus | 3.05711802 | 0.00223476 | 0.02058482 |
| Dorsomedial nucleus of the hypothalamus | 3.00238689 | 0.00267872 | 0.02284651 |
| Medial preoptic area | 3.00514153 | 0.00265457 | 0.02284651 |
| Subiculum | 2.98357499 | 0.00284902 | 0.02343115 |
| Lateral hypothalamic area | 2.93308509 | 0.00335612 | 0.02664991 |
| Anterior group of the dorsal thalamus | -2.9172723 | 0.00353107 | 0.02710449 |
| Posterior auditory area | 2.88169639 | 0.00395541 | 0.0291333 |
| Cuneiform nucleus | 2.87436587 | 0.0040484 | 0.0291333 |
| Dorsal premammillary nucleus | 2.8581801 | 0.00426078 | 0.0297325 |
| Cuneate nucleus | 2.83699726 | 0.004554 | 0.03084397 |
| Agranular insular area | -2.7423234 | 0.00610063 | 0.03927668 |
| Nucleus of the lateral lemniscus | 2.74020028 | 0.00614018 | 0.03927668 |
| Parasubiculum | 2.70757742 | 0.00677763 | 0.0421825 |
| Parafascicular nucleus | 2.6781982 | 0.00740194 | 0.04485576 |
| Suprachiasmatic nucleus | 2.65433383 | 0.00794652 | 0.04692114 |
| Posterior limiting nucleus of the thalamus | 2.62509199 | 0.00866256 | 0.04987036 |

**Extended Data Table 4 | Regions that show significant difference between SUS and RES males (iDISCO+ analysis)**

| name | z score | p value | q value |
|---|---|---|---|
| Ectorhinal area | 4.47972509 | 7.47E-06 | 0.00085632 |
| Locus ceruleus | 4.26157298 | 2.03E-05 | 0.00155139 |
| Olfactory tubercle | 4.14432931 | 3.41E-05 | 0.00195453 |
| Parafascicular nucleus | 4.02339327 | 5.74E-05 | 0.00263202 |
| Parasubiculum | 3.86123637 | 0.00011281 | 0.00400738 |
| Lateral amygdalar nucleus | 3.80856035 | 0.00013978 | 0.00400738 |
| Mediodorsal nucleus of thalamus | 3.80778588 | 0.00014022 | 0.00400738 |
| Nucleus of the brachium of the inferior colliculus | 3.77923655 | 0.00015731 | 0.00400738 |
| Anteroventral nucleus of thalamus | 3.58670311 | 0.00033489 | 0.00767802 |
| Septofimbrial nucleus | 3.52502324 | 0.00042345 | 0.00882585 |
| Perirhinal area | 3.49542271 | 0.00047331 | 0.00904298 |
| Medial preoptic area | 3.45481622 | 0.00055067 | 0.0097117 |
| Endopiriform nucleus | 3.42689613 | 0.00061052 | 0.00999814 |
| Nucleus sagulum | 3.34181909 | 0.00083231 | 0.01272158 |
| Nucleus of the trapezoid body | 3.31306979 | 0.00092278 | 0.01322286 |
| Secondary motor area | 3.29503648 | 0.00098409 | 0.01323847 |
| Subiculum | 3.2567128 | 0.0011271 | 0.01323847 |
| Interanterodorsal nucleus of the thalamus | 3.23494244 | 0.00121667 | 0.01323847 |
| Midbrain reticular nucleus | 3.23681636 | 0.00120871 | 0.01323847 |
| Pedunculopontine nucleus | 3.24403066 | 0.00117851 | 0.01323847 |
| Nucleus of the lateral lemniscus | 3.22260093 | 0.00127032 | 0.01323847 |
| Field CA3 | 3.15598786 | 0.00159956 | 0.01470309 |
| Globus pallidus | 3.15531402 | 0.00160325 | 0.01470309 |
| Retrochiasmatic area | 3.17419671 | 0.00150252 | 0.01470309 |
| Posterior limiting nucleus of the thalamus | 3.11110696 | 0.00186387 | 0.01643575 |
| Dentate gyrus | 3.08837296 | 0.00201256 | 0.01708962 |
| Entorhinal area | 3.07037674 | 0.00213789 | 0.0175055 |
| Primary motor area | 3.05124694 | 0.00227893 | 0.01801691 |
| Mammillary body | 2.99520191 | 0.00274263 | 0.02096009 |
| Field CA2 | 2.9377807 | 0.00330571 | 0.02368438 |
| Nucleus of the posterior commissure | 2.94024246 | 0.00327956 | 0.02368438 |
| Lateral septal complex | 2.90917625 | 0.00362383 | 0.02517683 |
| Medial geniculate complex | 2.88289109 | 0.00394044 | 0.02657131 |
| Anterior cingulate area | 2.85198264 | 0.00434475 | 0.0284606 |
| Nucleus of Darkschewitsch | 2.78161581 | 0.0054089 | 0.03444718 |
| Prelimbic area | 2.77231818 | 0.00556586 | 0.03448878 |
| Temporal association areas | 2.75754842 | 0.00582366 | 0.03513659 |
| Nucleus of the optic tract | 2.74772955 | 0.00600095 | 0.03527789 |
| Orbital area | 2.69602442 | 0.00701725 | 0.03998046 |
| Lateral posterior nucleus of the thalamus | 2.65857582 | 0.00784717 | 0.03998046 |
| Vascular organ of the lamina terminalis | 2.66786192 | 0.00763356 | 0.03998046 |
| Anterior hypothalamic nucleus | 2.66348829 | 0.00773351 | 0.03998046 |
| Precommissural nucleus | 2.67487517 | 0.00747571 | 0.03998046 |
| Ventral tegmental nucleus | 2.68030692 | 0.00735547 | 0.03998046 |
| Anterodorsal nucleus | 2.62030028 | 0.00878524 | 0.04378678 |
| Posterior amygdalar nucleus | 2.60926754 | 0.00907363 | 0.04426194 |
| Diagonal band nucleus | 2.56669745 | 0.01026722 | 0.04904095 |

**Extended Data Table 5 | Regions that show significant difference between RES and CTRL males (iDISCO+ analysis)**

| name | z score | p value | q value |
|---|---|---|---|
| Nucleus of the solitary tract | 3.90889137 | 9.27E-05 | 0.01123524 |
| Lateral hypothalamic area | 3.79705385 | 0.00014643 | 0.01183154 |
| Arcuate hypothalamic nucleus | 3.71868549 | 0.00020026 | 0.01213576 |
| Supramammillary nucleus | 3.6307392 | 0.00028261 | 0.01370093 |
| Subparafascicular nucleus | 3.47475817 | 0.00051131 | 0.02065692 |
| Pontine reticular nucleus | 3.32250912 | 0.00089212 | 0.03089284 |
| Superior colliculus | 3.22981759 | 0.00123869 | 0.03563819 |
| Anterodorsal preoptic nucleus | 3.21089902 | 0.0013232 | 0.03563819 |

# Reporting Summary

## Statistics

For all statistical analyses, confirm that the following items are present in the figure legend, table legend, main text, or Methods section.

| n/a | Confirmed | |
|---|---|---|
| ☐ | ☒ | The exact sample size (*n*) for each experimental group/condition, given as a discrete number and unit of measurement |
| ☐ | ☒ | A statement on whether measurements were taken from distinct samples or whether the same sample was measured repeatedly |
| ☐ | ☒ | The statistical test(s) used AND whether they are one- or two-sided<br>*Only common tests should be described solely by name; describe more complex techniques in the Methods section.* |
| ☒ | ☐ | A description of all covariates tested |
| ☐ | ☒ | A description of any assumptions or corrections, such as tests of normality and adjustment for multiple comparisons |
| ☐ | ☒ | A full description of the statistical parameters including central tendency (e.g. means) or other basic estimates (e.g. regression coefficient) AND variation (e.g. standard deviation) or associated estimates of uncertainty (e.g. confidence intervals) |
| ☐ | ☒ | For null hypothesis testing, the test statistic (e.g. *F*, *t*, *r*) with confidence intervals, effect sizes, degrees of freedom and *P* value noted<br>*Give P values as exact values whenever suitable.* |
| ☒ | ☐ | For Bayesian analysis, information on the choice of priors and Markov chain Monte Carlo settings |
| ☒ | ☐ | For hierarchical and complex designs, identification of the appropriate level for tests and full reporting of outcomes |
| ☐ | ☒ | Estimates of effect sizes (e.g. Cohen's *d*, Pearson's *r*), indicating how they were calculated |

*Our web collection on statistics for biologists contains articles on many of the points above.*

## Software and code

Policy information about availability of computer code

| Data collection | Social CPP data was acquired using CPP box(Med Associates). Other behaviors were assayed using Ethovision XT 11/12 (Noldus). Electrophysiological data was performed with pClamp 10.0 (Molecular Devices). Fiber photometry data was acquired with open source Bonsai software 2.4.0 and custom Neurophotometrics (Neurophotometrics, Ltd) hardware. Cleared brain imaging was done on lightsheet microscope (LaVision). |
|---|---|
| Data analysis | Analysis of electrophysiological data was performed with Clampfit 10.0 (Molecular Devices). Analysis of fiber photometry data was performed with Matlab 2019/2020/2021 (Mathworks, Inc.). Histological data was analyzed using ImageJ1.52b(Fiji). Cleared brain data was analyzed by ClearMap (python2.7). Statistics are done by using GraphPad Prism8 and 9. All MATLAB code for calcium imaging process, Python code for iDISCO+ analysis can be obtained from github (https://github.com/nyclong/2021-07-11642-Nature.git) |

For manuscripts utilizing custom algorithms or software that are central to the research but not yet described in published literature, software must be made available to editors and reviewers. We strongly encourage code deposition in a community repository (e.g. GitHub). See the Nature Portfolio guidelines for submitting code & software for further information.

## Data

Policy information about availability of data

All manuscripts must include a data availability statement. This statement should provide the following information, where applicable:
- Accession codes, unique identifiers, or web links for publicly available datasets
- A description of any restrictions on data availability
- For clinical datasets or third party data, please ensure that the statement adheres to our policy

All raw data for animal behaviors, ISH and IHC statistics are available as source data files. Allen Brain Atlas ISH database is used to search for possible molecular markers in lateral septum.

# Field-specific reporting

Please select the one below that is the best fit for your research. If you are not sure, read the appropriate sections before making your selection.

☒ Life sciences    ☐ Behavioural & social sciences    ☐ Ecological, evolutionary & environmental sciences

For a reference copy of the document with all sections, see nature.com/documents/nr-reporting-summary-flat.pdf

# Life sciences study design

All studies must disclose on these points even when the disclosure is negative.

| | |
|---|---|
| Sample size | Sample sizes were chosen according to previous experiments (Chaudhury et al. 2013, Friedman et al. 2014, Christoffel et al. 2015) |
| Data exclusions | Grubb's test were used to exclude significant outliers. This test was only performed once per data set. In experiments requiring viral infection of a specific brain region, mice were excluded from behavioral analysis if the virus was found to be mis-targeted or not expressed according to pre-determined anatomical criteria. No data was excluded for other reasons. |
| Replication | Data was collected using biological replicates (e.g. multiple brain slices per animal analyzed for in-situ hybridization and Immunohistochemistry). All attempts at replication were successful. Fig. 2c and Extended Data Fig. 3i were repeated in 3 separate cohorts per sex, with all showing similar results. Fig. 5a, b (right panel) was repeated in 3 seperate male cohorts (n = 6) and 1 female cohort (n = 2) with all showing similar results. Extended Data Fig. 3j was repeated twice in both sexes, with both showing similar results. Extended Data Fig. 6a was repeated in 4 separate cohorts in both sexes with all showing similar results. Extended Data Fig. 8a, d (right panel) and 10e were repeated twice in males only, with both cohorts showing similar results. Extended Data Fig. 8b was repeated three times, with all showing similar results. |
| Randomization | Animals and samples were assigned randomly to control and experimental groups, except in cases where social behaviors were compared between groups. In these cases, SI ratio was assessed and groups were counter-balanced for equal levels of SI ratio before manipulations were performed. |
| Blinding | Experimenters were blind to group allocation except for social CPP data collection since we need to pre-test social CPP and assign boxes for different groups of animals (CTRL or RES or SUS) separately, we need to know which ones belong to which groups so that we can balance the conditioned chambers to avoid bias, analyses were performed blind to experimental conditions (Behavioral scoring from videos, fiber photometry analysis, quantification of in-situ hybridization results). |

# Reporting for specific materials, systems and methods

We require information from authors about some types of materials, experimental systems and methods used in many studies. Here, indicate whether each material, system or method listed is relevant to your study. If you are not sure if a list item applies to your research, read the appropriate section before selecting a response.

## Materials & experimental systems

| n/a | Involved in the study |
|---|---|
| ☐ | ☒ Antibodies |
| ☒ | ☐ Eukaryotic cell lines |
| ☒ | ☐ Palaeontology and archaeology |
| ☐ | ☒ Animals and other organisms |
| ☒ | ☐ Human research participants |
| ☒ | ☐ Clinical data |
| ☒ | ☐ Dual use research of concern |

## Methods

| n/a | Involved in the study |
|---|---|
| ☒ | ☐ ChIP-seq |
| ☒ | ☐ Flow cytometry |
| ☒ | ☐ MRI-based neuroimaging |

# Antibodies

| | |
|---|---|
| Antibodies used | Mouse monoclonal IgG anti-c-Fos (Santa Cruz Biotechnology, C-10, Cat. No.: sc-271243, 1:1000)<br>Rabbit Polyclonal anti-cfos antibody (synaptic systems, Cat. No.: 226 003, 1:1000)<br>Cy™2 AffiniPure Donkey Anti-Rabbit IgG (H+L), Jackson ImmunoResearch Laboratories, Inc.Code Number: 711-225-152, Lot Number: 78325, Clonality: Polyclonal, RRID: AB_2340612, 1:1000;<br>Cy™3 AffiniPure Donkey Anti-Rabbit IgG (H+L), Jackson ImmunoResearch Laboratories, Inc.Code Number: 711-165-152, Lot Number: 88067, Clonality: Polyclonal, RRID: AB_2307443, 1:1000;<br>Cy™5 AffiniPure Donkey Anti-Rabbit IgG (H+L), Jackson ImmunoResearch Laboratories, Inc.Code Number: 711-175-152, Lot Number: 84963, Clonality: Polyclonal, RRID: AB_2340607, 1:1000.<br>Donkey anti-Rabbit IgG (H+L) Highly Cross-Adsorbed Secondary Antibody, Alexa Fluor™ 647, Thermo Fisher Scientific, cat: A-31573, RRID, AB_2536183, 1:1000 |
| Validation | According to manufacturers, mouse polyclonal IgG anti-cFos was validated in mouse and human tissue for immunofluorescence. This antibody has been utilized in previous publications to detect cfos in mice using immunofluorescence (Liu, J. et al.,2017).<br>According to manufacturers, Rabbit Polyclonal anti-cfos antibody IgG was validated in transgenic mice expressing GFP using immunofluorescence. This antibody has been used in previous publications to detect cfos in mice using immunofluorescence (Li, X, et al., 2021).<br>According to manufacturers, Cy™2/3/5 AffiniPure Donkey Anti-Rabbit IgG (H+L) was validated in mice. This antibody has been used in previous publications in mice using immunofluorescence (HE, X., et al., 2018).<br>According to manufacturers, Donkey anti-Rabbit IgG (H+L) Highly Cross-Adsorbed Secondary Antibody, Alexa Fluor™ 647 was validated in mice. This antibody has been used in previous publications in mice using immunofluorescence (Kyprianou, C., et al., 2020). |

# Animals and other organisms

Policy information about studies involving animals; ARRIVE guidelines recommended for reporting animal research

| | |
|---|---|
| Laboratory animals | C57BL6/J: Obtained from Jackson Laboratory (male and female), 4-10 weeks of age, cat. 000664).<br>NT-Cre (017525-B6;129-Nts <tm1(cre) Mgmj>/J; Jackson Laboratory, Stock No: 017525), (male and female) 6-10 weeks of age. Heterozygous mice were used for all experiments.<br>CD1: Obtained from Charles River labs (male, 4-6 months of age, cat. CRL22). Male mice were used as aggressors and not studied here.<br>ERα-Cre: B6N.129S6(Cg)-Esr1tm1.1(cre)And/J, Obtained from Jackson Laboratory (male, 10 weeks of age, Stock No: 017911), were crossed with CD1 wild-type mice for one generation to generate ERα-Cre/CD1 F1 hybrids. Male F1 hybrids 4-6 months of age were used for experiments. Female F1 hybrids were co-housed with F1 males prior to testing. These mice were used as aggressors and not studied here. |
| Wild animals | No wild animals were used in the study. |
| Field-collected samples | No field collected samples were used in the study. |
| Ethics oversight | Procedures were performed in accordance with the National Institutes of Health Guide for Care and approved by the Use of Laboratory Animals and the Icahn School of Medicine at Mount Sinai Institutional Animal Care and Use Committee. Additional information about mice used in this study can be found in the Life Sciences Reporting Summary. |

Note that full information on the approval of the study protocol must also be provided in the manuscript.

