## [Peer Review File · Nature]

Manuscript Title: Social trauma engages lateral septum circuitry to occlude social reward

Reviewer Comments & Author Rebuttals

Reviewer Reports on the Initial Version:

Referees' comments:

Referee #1 (Remarks to the Author):

Li et al. present data to suggest that a genetically defined group of cells within the lateral septum serve as an inhibitory brake upon the intrinsic reward associated with social interactions, perhaps switching the valence such that a normally rewarding stimuli becomes aversive or perceived as something to avoid. While the data are interesting, I have major concerns regarding the specificity of observations and manipulations which were done. It may be peripheral to the current manuscript, but I found the sex differences in figure 2D intriguing and certainly worthy of future examination.

Major comments:

-Something I struggled with while reading this manuscript was that the interpretation seems to have led the experimental design and therefore the data. For example, from the data presented in figure 2D it seems feasible that nearly any brain region could have been selected for in depth study. If I am interpreting it correctly, figure 2d is intriguing and looks to be one of the strongest differential effects of the paper (between male and female). There is no brain region in the male mouse that shows a decrease in c-fos expression and perhaps 30% of the brain regions in females are above 2 stdev (guessing because there is no N listed for the Y-axis and the heatmap makes it difficult to estimate). Indeed, the LS in male mice actually appears to be in the lowest 1/5th of all regions, suggesting it is actually less affected when compared with nearly anywhere else in the brain.

-Something similar is done when selecting for NTS+ cells within lateral septum. What was the % overlap for SST+ and cFOS? While the manipulations of neurotensin-expressing cells are novel, it remains unclear whether these effects are due to the number of cells manipulated or their specific genetic cell type. Would a pan-neuronal manipulation produce similar behavioral effects?

-I am not convinced that the data support anything specific to social interactions. It has been previously proposed that LS circuitry is relevant for generalization of context and affect (Brady & Nauta 1953; Leutgeb & Mizumori 2002) and the data are consistent with a broader deficit in context representation and over-generalization. I am also a bit unclear whether the authors are actually trying to say something specific about social interactions or using it as a method to assess a more general computation.

“Together, these data suggest that previously rewarding social targets are possibly perceived as social threats in SUS mice, resulting from hyperactive NTLs neurons that occlude social reward processing”

“On the basis of these data, we suggest that after CSDS, SUS mice may overgeneralize social threat cues and perceive juveniles as social threats, similar to what is observed when being attacked by a highly aggressive CD-1 mouse.”

I would ask the authors to more explicitly state their view on this throughout the text. Is the LSnts population of neurons blocking specifically social reward or are they part of a circuit to generalize

conditioned experiences to similar stimuli? However, I do feel that further experiments would be required to make a definitive statement on this. Specifically, in a non-social task would the same manipulations induce similar over-generalization? Do mice that experience CVS (specifically restraint) show avoidance to all falcon tubes? If CVS was paired with conspecific odors/vocalizations do mice then show avoidance to juvenile mice? Any further experiments in this space would be invaluable in determining which line of reasoning, and interpretation, may be applied.

-In line with the above comment, I am highly skeptical of the specificity of figure 3. It is well established that LS/limbic activity is important for locomotion (Bender et al., 2015, Wirtshafter et al., 2019) and effort/motivation (Jarrard 1973, 1984; Tracy et al., 2001), and animals are undoubtedly struggling to move when pinched, attacked, or investigating a conspecific. These data are consistent with both prior interpretations, neither of which require positive or negative valence or specificity for social encounters. In fact, the authors' own data (extended data figure 6) would seem to suggest that LSnts neurons are directly involved in explicitly non-social behaviors used to measure affective state. I also do not think that the activity patterns shown in figure 3 are surprising or specific to the LS. Can the authors show that any other brain region (or cell type in LS) does not show a similar fluorescence change during tail pinch?

-At no point is the Social Interaction Ratio task actually described in the text in enough detail to understand what is happening. This includes the paper Li et al. cites for this task (Christoffel et al., 2015). In fact, Christoffel et al., 2015 cites two others (Bubser, M. & Deutch, A.Y. *Synapse* 32, 13–22 (1999) and Chen, B.T. et al. *Nature* 496, 359–362 (2013)), neither of which appears to actually have conducted any form of social interaction task. What are the dimensions of the arena? Where is the 'interaction' zone? Is the target mouse freely moving or constrained by something?

Minor Comments

-Why switch between neuron and mouse comparisons for figures 2k-m? What does figure 2m look like when all cells are shown? Or Figures 2k and l when averaged to the mouse level?

-line 77 "ie." Vs 'i.e.'

-Were there susceptible mice to the CVS conditioning? Could SUS/RES status with CSDS be used to predict CVS response? Or vice versa? With a milder CVS that more closely matches CSDS (10-minutes rather than 1-hour; 10 days instead of 28), are there SUS/RES groupings with CVS?

-From the example images in figure 5A it seems that NAC and PAG are two of the least heavily innervated regions, rather than "some of the most densely connected regions". Could the authors describe why these regions were selected? There is a rich history of topographic inputs/outputs with the LS (Swanson & Cowan 1979; Risold & Swanson 1997), I'm wondering if some of the projection differences here may be explained by topographic differences in connectivity?

-As neurotensin is highly expressing in neighboring brain regions (LPO, BNST, nucleus accumbens), please show entire slice histology for DREADD experiments in figure 4 and extended data figure 5. Can variability in transfection accuracy/efficacy be correlated with effect size?

-I fully understand the space limitations of the journal make this difficult, but it may be worth discussing the following papers: McHenry et al. 2017, Sweeney and Yang 2016, Bredewold et al. 2015.

Referee #2 (Remarks to the Author):

In this study, the authors ask if chronic social defeat stress can alter social reward (it can) and what are the mechanisms involved. Using a host of approaches, the authors identify neurotensin neurons in the lateral septum as a key driver of social stress reduced social reward. This is a really exciting set of data, scoring high in originality, rigor, robustness. However I think that there are a few issues that when addressed would strengthen the paper.

1) iDISCO analysis is fairly cursory, and it is unclear how every group is different in males. The authors could apply so more sophisticated approaches to understand brain networks in this process. While I realize they have chosen to focus on LS, this is a huge data set that can provide great insight.

2) With relevance to iDISCO, it is unclear if they are correcting for multiple comparisons in any way. This is crucial as they are comparing over a hundred regions (I assume, because the authors do not provide any raw data here)

3) The electrophysiology data was intriguing, but again, not rigorously explored. For example, there is a clear difference between RMP (lower) and spiking (higher). The authors could explore mechanisms of this based on the spike shape in their existing data set. This could be highly informative, especially since this may explain the in vivo data for photometry.

4) While it was helpful to look at cocaine as a proxy for reward, a more ethologically relevant reward such as food, high fat, or chow, would be much more informative with respect to the engagement of this circuit in vivo. I do not consider IP injection of cocaine to be a good proxy for reward, and I think the field collectively needs to stop doing this.

5) The authors attempt to use optogenetics to demonstrate the pathways that drive the lack of reward, and come down to LS to NAc and LS to AHn using Chr2. The authors should demonstrate that their stimulation paradigms alter function of the downstream regions in the fashion they suspect either in vivo or ex vivo. The only data presented is a single pulse showing a putative IPSC that is not pharmacologically isolated. For example, it is possible that repeated high frequency stimulation could drive depolarization block of terminals and have the opposite effect on target firing. I think doing this in slice would be sufficient, and is critical to interpretation of the findings. In this same vein, it would be extremely helpful for the authors assertion that this pathway (s) are engaged to drive this behavior to silence in vivo, perhaps using some of the new tools developed to silence terminals. I realize these are new tools, and don't yet know how well they work in other labs hands, but it seems a great place to try.

Minor point: It is unclear why ER-alpha-cre mice were given CNO (line 672). This mouse line is also mentioned in line 659. Either this is a mistake of inclusion, or the authors have to explain why these mice are being used as there is no mention of them in the results. These ER-alpha mice are also mentioned in line 496.

Referee #3 (Remarks to the Author):

Summary.

The authors investigate the neural circuitry underlying the effects of social stress to alter subsequent social interaction and preference using chronic social defeat stress (CSDS) as a model of social trauma. Male and female mice whom demonstrated reduced preference for social interaction following CSDS were classified as “susceptible” (SUS) and compared to resilient and control mice. Using a large variety of techniques including whole-brain cFos mapping (iDisco+), in vivo Ca²⁺ imaging, multiplex ISH, whole cell recordings, chemogenetics, optogenetics, and circuit mapping, the authors identify a population of neurotensin+ neurons in the lateral septum (NTLS) in the control of CSDS-impacted social behavior in SUS mice.

Concerns.

- 1) The authors use a fixed order for testing the behavioral effects of CSDS: 1) SI test, 2) RI test, 3) CPP test, rather than using a counterbalanced test order design. One potential concern is the contamination effects that one assay might carry over onto subsequent assays. For example, stressful aggressive encounters during the RI may alter subsequent learning or preference in the CPP assay, or interaction in the SI assay may influence behavior in the RI assay. The authors should provide a control experiment where similar behavioral effects are observed in each assay when that assay is tested first.
- 2) Does CSDS specifically alter preference on the social CPP assay or simply block learning overall? Understanding whether social stress alters learning and memory processing in general as opposed to socially-specific learning would contribute to the overall scope and specificity of the paper.
- 3) One of the major indices of social behavior used by the authors is “avoidance” time during the RI assay, however, this is not clearly defined. In the methods, the authors define “social avoidance” as the “avoidance or escape behavior of the experimental mouse when approached”. This lacks specificity regarding how this behavior was rigorously scored. What constitutes “escape” behavior? What about “avoidance behavior”? No specifics or references to scoring rubric or parameters are provided.
- 4) Was there aggression during the RI assay in any of the groups? This is not mentioned in the text and could potentially influence subsequent behavior.
- 5) The manuscript emphasizes the role of NTLS neurons in social reward. While data presented in figure 4 uses a chemogenetic approach to nicely demonstrate that NTLS neurons are required for the effects of CSDS to block social CPP, the majority of the experiments presented examine the role of NTLS neurons in social interaction (NOT social reward) using the SI or RI assay (e.g. figures 2, 3 and 5). Indeed, critical characterization of these neurons (Fig. 2 and 3) is entirely based off of their role in social behavior. Thus, it would seem that perhaps the large focus on social reward in the text as well as in the title should be shifted more to focus on the role of these neurons in mediating social interaction or social behavior more generally.

6) In their multiplex ISH experiments, the authors report that Nt shares only 5% overlap with Drd3+. This finding is crucial, as a recent study (Shin et al., 2018, Neuron) found that Drd3+ neurons in the LS mediate normal social interaction and show reduced activation when stress is delivered prior to testing mice in a 3-chamber social interaction assay or a homecage social assay – two of the very same assays used in this manuscript. This poses the interesting possibility that NTLN neurons and Drd3+ neurons exert opposing functions to control social behavior following stress. Given the extreme relevance of the Shin et al. study, the authors should incorporate some kind of discussion regarding how this prior study impacts their findings and how the data presented in each paper can be reconciled.

7) In addition to identifying a role for NTLN neurons in social interaction, the authors clearly demonstrate a role for NTLN neurons in anxiety-like behavior (Ext. Data Figs. 1 and 6). This leads to the question of whether these neurons do indeed specifically encode social reward following CSDS in SUS mice, or whether they simply encode a state of anxiety, which in turn effects subsequent social behavior, as anxiety is well known to inhibit adaptive social behavior. Moreover, the role of these neurons in anxiety further begs the question of what role these neurons are playing that Crhr2 LS neurons have not already been shown to play (e.g. Anthony et al., 2014 Cell). This later issue is especially critical given that the authors themselves find that Nt and Crhr2 mRNA are “largely colocalized” in LS (Ext. data Fig. 3). The role of NTLN neurons in anxiety and how this interacts with the role of these neurons in social behavior should be discussed.

8) In Fig. 4b and 4i, the authors use a paired two-tailed t-test to analyze their results, but it seems that given that these mice were run together and are being displayed on the same graph, the appropriate statistic to use would be a 3X2 mixed ANOVA, with drug condition as the repeated measure and virus condition as the between-subjects measure, then follow this up with post-hoc tests to look at differences between CNO and vehicle within each viral condition. Same argument applies elsewhere in the figure.

Referee #4 (Remarks to the Author):

In this study, Li. et. al. identified the LS Nt cells as an important population for chronic social-defeat induced social avoidance and social reward learning. They found differences in LS Nt cell responses in SUS and RES animals and demonstrated that LS Nt activity can bidirectionally modulate social investigation and social reward learning in defeated animals. The authors further identified the downstream targets of LS Nt cells and demonstrated that LS-AHN and LS-NAc pathway manipulation can phenocopy the LS Nt manipulation. This study is interesting and the results are generally robust. The reviewer particularly appreciates the inclusion of females in the study. However, although the study attempts to reach a conclusion regarding LS Nt role in social reward learning, the data mainly supports a role of LS Nt in generalized social avoidance and general anxiety.

Here are some specific comments:

1. Are the SUS and RES animals representing animals within one continuous distribution or two qualitatively different categories of animals? Please plot avoidance and each behavior parameter from all animals in one plot and determine whether the nature of the distribution (e.g. bi-modal or Gaussian).

3. The social CPP paradigm appears to be not very robust. As stated in the method, “For female sCPP, during conditioning, the juvenile mice were confined in a wire mesh cup, which we found was necessary for females to form CPP, whereas males only formed a preference when they were able to freely interact with the juvenile outside the cup.” Why females only form sCPP with a confined juvenile? If the social CPP is due to the positive valence of social interaction, shouldn't free interaction provides more opportunities for interaction?

2. This study analyzed two behavioral phenotypes induced by chronic defeat: social avoidance (SI test) and social reward learning (social CPP test). SUS animals are defined as animals that show social avoidance towards a non-threatening conspecific. The SUS animals also show impaired social reward-dependent contextual learning. However, social avoidance and social reward learning appears to be only weakly correlated especially in males (although it did reach significance). The correlation appears to be driven mainly by animals in RES and CTRL animals as in 1f and 1k, animals with SI ration below 1 (SUS animals) showed highly variable CPP scores, suggesting that SUS animals can either show social CPP or not. These results suggest that these two behavioral phenotypes are separable. SUS animals do not necessarily show social CPP deficits. If the authors would like to study the role of LS NST cells in social reward learning, should not all the comparisons be made between animals that show social reward learning deficits and those not? As of now, all comparisons are made between animals that show social avoidance vs. those not. In some experiments, animals' performance during social reward learning is not measured (Figure 2 and Figure 3). Thus, no conclusion can be reached regarding how LS NST cells differ between animals show defeat-induced deficits in social reward learning and those not.

4. It is interesting that LS NTS cells showed clear difference in SUS and RES animals. Is this difference limited to LS NTS cells? Do GFP-negative cells also show such a difference?

5. An important question is whether the differences in cell responses between SUS and RES animals are due to pre-defeat differences. This question is hard to address for terminal experiments (e.g. slice recording) but can be addresses with in vivo recording that allows for repeated sampling. Do SUS and RES animals differ in their responses to different social targets and non-social stress before defeat?

6. As the non-social stress strongly activates LS NT cells and LS NT manipulation also influences performance in OFT and EPM, it appears that the cells signal general stress and anxiety level. The changes in cell excitability suggests a non-input specific change in responses. How specific is the LS NTS cell in vivo response to juvenile after chronic defeat? Do the cells also respond more to other non-social stimulus (e.g. novel object), especially those with negative valence (e.g. air puff)? If it is not specific to social cues, how do the NT cells differ from other LS populations, e.g. Crhr2, Oxtr, Drd3, SST, that have been indicated in stress, fear and anxiety?

7. It is interesting that the chemogenetic induced changes in social investigation is defeat-experience dependent. Please discuss the LS role in the known social avoidance circuit (e.g. with NAc, VTA, BLA). Why increase the LS activity alone is insufficient in driving social avoidance behavior?

8. Does chemogenetic manipulation of LS Nt cells influence social CPP in stress-naïve animals?

9. The conclusion regarding the extent of overlapping of LS-AHN, LS-PAG, LH-NAc projectors in Extended Figure 7 requires further investigation. The number of retrogradely labeled cells in the LS is generally low, especially those labeled with tdTomota. With such a low number of cells being labeled, the chance overlap will be close to 0. Thus, a negative conclusion (no overlap) is hard to draw. The fact that 7/17 LS cells retrogradely labeled from AHN also retrogradely labeled from NAc suggest that the extent of bifurcation to NAc and AHN is likely to be very high. The authors need to

either increase the labeling efficiency (e.g. CTB could work much better) or use a different method, e.g. examine the whole brain projection pattern of LS cells that are retrogradely labeled from one specific region.

Author Rebuttals to Initial Comments:

Response to Referees

Referees' comments:

Referee #1 (Remarks to the Author):

Li et al. present data to suggest that a genetically defined group of cells within the lateral septum serve as an inhibitory brake upon the intrinsic reward associated with social interactions, perhaps switching the valence such that a normally rewarding stimuli becomes aversive or perceived as something to avoid. While the data are interesting, I have major concerns regarding the specificity of observations and manipulations which were done. It may be peripheral to the current manuscript, but I found the sex differences in figure 2D intriguing and certainly worthy of future examination.

Major comments:

-Something I struggled with while reading this manuscript was that the interpretation seems to have led the experimental design and therefore the data. For example, from the data presented in figure 2D it seems feasible that nearly any brain region could have been selected for in depth study. If I am interpreting it correctly, figure 2d is intriguing and looks to be one of the strongest differential effects of the paper (between male and female). There is no brain region in the male mouse that shows a decrease in c-fos expression and perhaps 30% of the brain regions in females are above 2 stdev (guessing because there is no N listed for the Y-axis and the heatmap makes it difficult to estimate). Indeed, the LS in male mice actually appears to be in the lowest 1/5th of all regions, suggesting it is actually less affected when compared with nearly anywhere else in the brain.

We thank the reviewer for pointing this out and apologize for not better describing the data. Please also refer to reviewer #2 question 1) and 2). In figure 2d we show only the most differentially regulated brain regions according to data from males and the corresponding regions in females. In our revised version, we reanalyzed the data and added a table, which contains all regions detected as significantly different between groups by multiple comparison. With regards to the rank of the lateral septum, the reviewer's point is well taken. Indeed, lateral septum isn't the most differentially regulated region in either sex, however, in our revised version, after a new analysis requested by other reviewers, we show that the LS ranks #2 among all regions similarly regulated in both sexes during juvenile RI (please also refer to Extended Data tables 1-6). Please see figure below and also Fig. 2d.

-Something similar is done when selecting for NTS+ cells within lateral septum. What was the % overlap for SST+ and cFOS? While the manipulations of neurotensin-expressing cells are novel, it remains unclear whether these effects are due to the number of cells manipulated or their specific genetic cell type. Would a pan-neuronal manipulation produce similar behavioral effects?

We agree with the reviewer and have now examined SST-cfos co-expression using RNAscope in all three groups (CTRL/RES/SUS) in both sexes. Please see figure below and Extended data Fig.4 j,k. While we do see a significant increase of Sst+ cFos+ neurons between CTRL and SUS mice, SUS mice do not differ from RES mice.

The reviewer questioned whether the effects are due to genetic cell type or the number of cells manipulated and asked “Would a pan-neuronal manipulation produce similar behavioral effects?”. If we understand this correctly, the reviewer is asking us to test whether non-NT neurons regulate social behavior. According to the literature, the cell types and microcircuit organization of the septum is quite complex with cell types defined by different genetic markers producing different effects on behaviors. For example, among all neuronal subtypes in LS, the *Drd3* cell type has been shown to produce an opposite effect in terms of stress behavioral phenotypes compared to what we observed with NT neurons. For this reason, we don’t feel that a pan-neuronal manipulation in LS would help us to address the reviewer’s important question. Rather, we chose to inject CreOff-ChR2 into LS of *Nt-cre* mice to specifically label non-NT neurons with ChR2 and regulate their activity during social behavior. Indeed, by doing so we found optogenetic activation of non-NT neurons in LS does not affect social behavior (please see figure below and Extended data Fig. 12).

-I am not convinced that the data support anything specific to social interactions. It has been previously proposed that LS circuitry is relevant for generalization of context and affect (Brady & Nauta 1953; Leutgeb & Mizumori 2002) and the data are consistent with a broader deficit in context representation and over-generalization. I am also a bit unclear whether the authors are actually trying to say something specific about social interactions or using it as a method to assess a more general computation.

“Together, these data suggest that previously rewarding social targets are possibly perceived as social threats in SUS mice, resulting from hyperactive NTLN neurons that occlude social reward processing”

“On the basis of these data, we suggest that after CSDS, SUS mice may overgeneralize social threat cues and perceive juveniles as social threats, similar to what is observed when being attacked by a highly aggressive CD-1 mouse.”

I would ask the authors to more explicitly state their view on this throughout the text. Is the LSnts population of neurons blocking specifically social reward or are they part of a circuit to generalize conditioned experiences to similar stimuli? However, I do feel that further experiments would be required to make a definitive statement on this. Specifically, in a non-social task would the same manipulations induce similar over-generalization? Do mice that experience CVS (specifically restraint) show avoidance to all falcon tubes? If CVS was paired with conspecific odors/vocalizations do mice then show avoidance to juvenile mice? Any further experiments in this space would be invaluable in determining which line of reasoning, and interpretation, may be applied.

This is a very important point, and we appreciate the reviewer’s thoughts and suggestions. To clarify our position, we do not believe these cells specifically control social behaviors, but rather they may be involved in more general computations that use past information from stressful or threatening situations to guide future behaviors towards cues associated with those threatening or stressful situations. Our data is consistent with an older literature (Brady & Nauta 1953; Leutgeb & Mizumori 2002) whereby the authors used gross lesions to link the LS to emotional behaviors (ie. startle response) or *in vivo* physiology to show that LS neurons exhibit context-specific spatial representations. In the case of social threat/stress, we hypothesize that past experience can guide the over-generalizing threat responses toward other social cues (in this case, a juvenile mouse), but we also believe that past non-social threatening context can cause avoidance behavior to non-social cues associated with or similar to the past non-social threat experience. To test this hypothesis, we performed chronic restraint stress (CRS) and then tested whether mice would avoid a restrainer tube, similar to those used for stressful restraint. The restrained mice show robust avoidance toward the tubes and much less investigation. We also found that by inhibiting NT neurons with AAV-DREADDs, we could partially rescue the tube avoidance. Lastly, we paired a separate group of mice with same sex juvenile C57BL/6J bedding during CRS and tested whether those mice exhibit avoidance of juvenile mice. We found no effect of this type of olfactory-based conditioning on juvenile avoidance in CRS mice. Please see figure below and Extended data Fig.7

-In line with the above comment, I am highly skeptical of the specificity of figure 3. It is well established that LS/limbic activity is important for locomotion (Bender et al., 2015, Wirtshafter et al., 2019) and effort/motivation (Jarrard 1973, 1984; Tracy et al., 2001), and animals are undoubtedly struggling to move when pinched, attacked, or investigating a conspecific. These data are consistent with both prior interpretations, neither of which require positive or negative valence or specificity for social encounters. In fact, the authors' own data (extended data figure 6) would seem to suggest that LSnts neurons are directly involved in explicitly non-social behaviors used to measure affective state. I also do not think that the activity patterns shown in figure 3 are surprising or specific to the LS. Can the authors show that any other brain region (or cell type in LS) does not show a similar fluorescence change during tail pinch?

We appreciate the reviewers' concerns; this is an important point that needs further clarification. As mentioned above (and Extended data Fig 12), we are not suggesting that NT^{LS} neurons only respond to social cues, in fact our new data shows that NT^{LS} neurons encode information about stressful experience more generally. To disentangle issues related to signal specificity and motor activity, we have provided new data described below:

1) We analyzed motor behaviors during fiber photometry analysis of NT^{LS} activity in the RI test to compare motor behavior-related calcium events with those elicited by other behaviors such as social interaction, attack or aggressor approach. We find NT^{LS} neurons do respond to motion (initiation of movement, rearing, and turning head), but the Ca²⁺ responses are far lower than those elicited in response to stressful or threatening events. The average DF/F for a motion response is significantly lower (~2-5%) compared to social encounters or stressful experience where DF/F ranges from ~10%-35%. Together, these data confirm that NT^{LS} activity captured by fiber photometry is not just a non-specific motion signal, but a detectable signal that accompanies stressful/threatening events (e.g. tail suspension and aggressive attack). Please see figure below:

2) We have tested the impact of DREADD manipulations of NT^{LS} neurons on motor activity in the open field test (OFT) across all groups in both sexes (Extended Data Fig. 6d). We show that regulation of NT^{LS} neurons does alter motor behavior in the OFT, which might otherwise have confounded interpretation of results from our behavioral assays.

3) As the reviewer suggests, we have also examined Ca²⁺ activity in a region outside the LS that we know to be involved in social behaviors, [REDACTED], and found no changes in Ca²⁺ activity in response to a tail pinch.

[REDACTED]

-At no point is the Social Interaction Ratio task actually described in the text in enough detail to understand what is happening. This includes the paper Li et al. cites for this task (Christoffel et al., 2015). In fact, Christoffel et al., 2015 cites two others (Bubser, M. & Deutch, A.Y. Synapse 32, 1322 (1999) and Chen, B.T. et al. Nature 496, 359362 (2013)), neither of which appears to actually have conducted any form of social interaction task. What are the dimensions of the arena? Where is the interaction zone, Is the target mouse freely moving or constrained by something?

We apologize for the confusion as the Christoffel et al., (2015) paper was cited in error. In our revision, we now cite the proper social defeat protocol paper [Sam Golden, et al, 2011, nature protocol] and include a more detailed description of the social interaction test in the methods section.

Minor Comments

-Why switch between neuron and mouse comparisons for figures 2k-m? What does figure 2m look like when all cells are shown? Or Figures 2k and l when averaged to the mouse level?

For consistency, we've plotted the data according to the reviewer's suggestion in our revised version. Below is figure 2l where data is analyzed per animal rather than per cell. We also include other electrophysiological parameters in Extended data Fig. 5e.

-line 77 ie. Vs i.e.

We've edited the manuscript to make sure this is consistent.

-Were there susceptible mice to the CVS conditioning?

It's difficult to define susceptible mice in this way because CVS does not induce social avoidance. We don't have validated behavioral outputs that would allow us to accurately stratify based on susceptibility to CVS.

-Could SUS/RES status with CSDS be used to predict CVS response? Or vice versa? With a milder CVS that more closely matches CSDS (10-minutes rather than 1-hour; 10 days instead of 28), are there SUS/RES groupings with CVS?

We appreciate the reviewer's suggestion, but we've tested CVS protocols starting from 3 days to 28 days and found that for male mice only 21-28 days of 1-hour daily stressors are robust enough to elicit any lasting behavioral deficits. Therefore, if we limit the model to only 10 mins for 10 days there will be no significant stress effect.

-From the example images in figure 5A it seems that NAc and PAG are two of the least heavily innervated regions, rather than some of the most densely connected regions. Could the authors describe why these regions were selected?

We apologize for our language. We modified the text on Page 9 to better describe our rationale for choosing these regions, which was based largely on historical data. There is a well-established literature showing that the NAc is critical for regulating reward, including social reward, and the PAG is strongly related to control of defensive behaviors. For this reason, we hypothesized that these pathways downstream of NT^{LS} neurons were integral in mediating the circuit specific nature of social behavior.

-There is a rich history of topographic inputs/outputs with the LS (Swanson & Cowan 1979; Risold & Swanson 1997), I'm wondering if some of the projection differences here may be explained by topographic differences in connectivity?

The inputs and outputs of LS neurons are very fascinating indeed. Here we used cell-specific tracing methods, which we assume will not perfectly overlap with Dr. Swanson's data. We now cite these papers, and discuss discrepancies with our data that suggest this may be due to genetic subtype and/or topographic differences in connectivity (see also Antoine Besnard, et al, 2022, Molecular Psychiatry).

-As neurotensin is highly expressing in neighboring brain regions (LPO, BNST, nucleus accumbens), please show entire slice histology for DREADD experiments in figure 4 and extended data figure 5. Can variability in transfection accuracy/efficacy be correlated with effect size?

We always check for virus expression and ferrule position and exclude any animals with expression outside the LS or with very low virus expression. We replaced these images with those containing a broader view to confirm the specificity of our injections, please see below.

-I fully understand the space limitations of the journal make this difficult, but it may be worth discussing the following papers: McHenry et al. 2017, Sweeney and Yang 2016, Bredewold et al. 2015.

We now cite and discuss the papers above in our revised draft.

Referee #2 (Remarks to the Author):

In this study, the authors ask if chronic social defeat stress can alter social reward (it can) and what are the mechanisms involved. Using a host of approaches, the authors identify neurotensin neurons in the lateral septum as a key driver of social stress reduced social reward. This is a really exciting set of data, scoring high in originality, rigor, robustness. However, I think that there are a few issues that when addressed would strengthen the paper.

1) iDISCO analysis is fairly cursory, and it is unclear how every group is different in males. The authors could apply so more sophisticated approaches to understand brain networks in this process. While I Realize they have chosen to focus on LS, this is a huge data set that can provide great insight.

We appreciate the reviewer's perspective. We did a more comprehensive analysis, including additional comparison of all three groups CTRL/RES/SUS across both sexes. We include these additional analyses in Extended Data Table 1-6, which includes all regions significantly different between groups. Unfortunately, our sample size was too small to accurately model networks using weighted correlation analysis, which requires 20-30 per group. We feel this may be better for future follow up studies.

2) With relevance to iDISCO, it is unclear if they are correcting for multiple comparisons in any way. This is crucial as they are comparing over a hundred regions (I assume, because the authors do not provide any raw data here)

We apologize for the confusion. We applied multiple comparisons for the data analysis and now clarify this detail in the methods section.

3) The electrophysiology data was intriguing, but again, not rigorously explored. For example, there is a clear difference between RMP (lower) and spiking (higher). The authors could explore mechanisms of this based on the spike shape in their existing data set. This could be highly informative, especially since this may explain the in vivo data for photometry.

This is an excellent suggestion. We have reanalyzed additional physiological parameters (AP threshold, amplitude, half-width duration, fast hyperpolarization) and these findings are now included in the revised draft (Extended data Fig 5a). Interestingly, we found no differences in these parameters in SUS vs RES mice.

4) While it was helpful to look at cocaine as a proxy for reward, a more ethologically relevant reward such as food, high fat, or chow, would be much more informative with respect to the engagement of this circuit in vivo. I do not consider IP injection of cocaine to be a good proxy for reward, and I think the field collectively needs to stop doing this.

The reviewer's point is well taken. To address this concern, we measured Ca^{2+} activity in NT^{LS} neurons during consumption of a highly palatable peanut butter cup. We find no response of NT^{LS} neurons during any phase of peanut butter cup approach or consumption. We've now added this to Extended figure 5c.

5) The authors attempt to use optogenetics to demonstrate the pathways that drive the lack of reward, and come down to LS to NAc and LS to AHn using Chr2. The authors should demonstrate that their stimulation paradigms alter function of the downstream regions in the fashion they suspect either in vivo or ex vivo. The only data presented is a single pulse showing a putative IPSC that is not pharmacologically isolated. For example, it is possible that repeated high frequency stimulation could drive depolarization block of terminals and have the opposite effect on target firing. I think doing this in slice would be sufficient, and is critical to interpretation of the findings. In this same vein, it would be extremely helpful for the authors assertion that this pathway (s) are engaged to drive this behavior to silence in vivo, perhaps using some of the new tools developed to silence terminals. I realize these are new tools, and don't yet know how well they work in other labs hands, but it seems a great place to try.

We thank the reviewer for this thoughtful suggestion. We performed whole cell patch clamp electrophysiology in the presence of TTX to prevent action potentials along with GABazine to block

GABAA-mediated signaling. We now confirm that NT^{LS} sends monosynaptic inhibitory GABAergic inputs to the AHN and NAc. Please see below and Extended Data Fig. 11.

We also tested inhibitory opsins (eNpHR3.0) to silence NT^{LS} terminals in NAc or AHN to confirm necessity and sufficiency of NT^{LS} neuron activation in social interaction and social reward deficits after CSDS in male and female mice (see figure below and Extended Data Fig. 10).

Minor point: It is unclear why ER-alpha-cre mice were given CNO (line 672). This mouse line is also mentioned in line 659. Either this is a mistake of inclusion, or the authors have to explain why these mice are being used as there is no mention of them in the results. These ER-alpha mice are also mentioned in line 496.

Sorry about the confusion, we use ER-alpha-cre mice as our female CSDS aggressors. Stimulation of VMH neurons with DREADDs in the ER-alpha-cre mice is necessary to initiate aggression in male mice

towards females in the CSDS model. We have now clarified this point in the methods section and cited the female CSDS paper (Ref.12: Takahashi, A. et al., Sci Rep, 2017).

Referee #3 (Remarks to the Author):

Summary.

The authors investigate the neural circuitry underlying the effects of social stress to alter subsequent social interaction and preference using chronic social defeat stress (CSDS) as a model of social trauma. Male and female mice whom demonstrated reduced preference for social interaction following CSDS were classified as susceptible (SUS) and compared to resilient and control mice. Using a large variety of techniques including whole-brain cFos mapping (iDisco+), in vivo Ca²⁺ imaging, multiplex ISH, whole cell recordings, chemogenetics, optogenetics, and circuit mapping, the authors identify a population of neurotensin⁺ neurons in the lateral septum (NT^{LS}) in the control of CSDS-impacted social behavior in SUS mice.

Concerns.

1) The authors use a fixed order for testing the behavioral effects of CSDS: 1) SI test, 2) RI test, 3) CPP test, rather than using a counterbalanced test order design. One potential concern is the contamination effects that one assay might carry over onto subsequent assays. For example, stressful aggressive encounters during the RI may alter subsequent learning or preference in the CPP assay, or interaction in the SI assay may influence behavior in the RI assay. The authors should provide a control experiment where similar behavioral effects are observed in each assay when that assay is tested first.

We agree with the reviewer and performed the tests in reverse order (CPP-RI-SI) in WT mice after CSDS to rule out any order effects. Please see figure below and Extended data Fig. 2d-e.

2) Does CSDS specifically alter preference on the social CPP assay or simply block learning overall? Understanding whether social stress alters learning and memory processing in general as opposed to socially-specific learning would contribute to the overall scope and specificity of the paper.

We appreciate the reviewer's perspective. To assess general learning/memory, we performed novel object recognition and novel object placement in WT CTRL/SUS/RES animals following CSDS. We saw no difference in the time spent with the novel object or location in any group. See Fig below and Extended Data Fig 2a-c.

3) One of the major indices of social behavior used by the authors is avoidance time during the RI assay, however, this is not clearly defined. In the methods, the authors define social avoidance as the avoidance or escape behavior of the experimental mouse when approached. This lacks specificity regarding how this behavior was rigorously scored. What constitutes escape behavior? What about avoidance behavior No specifics or references to scoring rubric or parameters are provided.

We apologize for the omission. See also response to reviewer 1. We provide a more detailed description of the methods for social interaction and RI test and cite the correct protocol paper that describe SI metrics (Golden et al., Nat Protocol, 2011).

4) Was there aggression during the RI assay in any of the groups? This is not mentioned in the text and could potentially influence subsequent behavior.

In around 1000 mice used for these studies, only ~1% showed any aggression towards a juvenile, and these animals were excluded from analysis. We now state this explicitly in the methods section.

5) The manuscript emphasizes the role of NTLS neurons in social reward. While data presented in figure 4 uses a chemogenetic approach to nicely demonstrate that NTLS neurons are required for the effects of CSDS to block social CPP, the majority of the experiments presented examine the role of NTLS neurons in social interaction (NOT social reward) using the SI or RI assay (e.g. figures 2, 3 and 5). Indeed, critical characterization of these neurons (Fig. 2 and 3) is entirely based off of their role in social behavior. Thus, it would seem that perhaps the large focus on social reward in the text as well as in the title should be shifted more to focus on the role of these neurons in mediating social interaction or social behavior more generally.

We appreciate the reviewer's perspective. We performed fiber photometry and optogenetics (please see response to reviewer #2 question 5) during CPP and found that NT^{LS} neurons elicit higher activity during the social-paired conditioning session in SUS mice. We also show that optogenetic activation and inhibition of NT^{LS}-AHN and NT^{LS}-NAc terminals bidirectionally regulates sCPP. Please see figure below and also new Extended data Fig. 5d and 10b-m.

6) In their multiplex ISH experiments, the authors report that Nt shares only 5% overlap with *Drd3+*. This finding is crucial, as a recent study (Shin et al., 2018, *Neuron*) found that *Drd3+* neurons in the LS mediate normal social interaction and show reduced activation when stress is delivered prior to testing mice in a 3-chamber social interaction assay or a home-cage social assay two of the very same assays used in this manuscript. This poses the interesting possibility that NT^{LS} neurons and *Drd3+* neurons exert opposing functions to control social behavior following stress. Given the extreme relevance of the Shin et al. study, the authors should incorporate some kind of discussion regarding how this prior study impacts their findings and how the data presented in each paper can be reconciled.

We appreciate the reviewer's comments. Please also refer to our response to reviewer #1. We find it intriguing that *Drd3* and NT exert largely opposing effects on behavior. It is known that *Drd3* and NT neurons are topographically distinct in the LS and it's quite likely that they have very different input/output projection patterns as well. We now discuss this possibility in the discussion section on page 11.

7) In addition to identifying a role for NT^{LS} neurons in social interaction, the authors clearly demonstrate a role for NT^{LS} neurons in anxiety-like behavior (Ext. Data Figs. 1 and 6). This leads to the question of whether these neurons do indeed specifically encode social reward following CSDS in SUS mice, or whether they simply encode a state of anxiety, which in turn effects subsequent social behavior, as anxiety is well known to inhibit adaptive social behavior. Moreover, the role of these neurons in anxiety further begs the question of what role these neurons are playing that *Crhr2* LS neurons have not already been shown to play (e.g. Anthony et al., 2014 *Cell*). This later issue is especially critical given that the authors themselves find that Nt and *Crhr2* mRNA are largely colocalized in LS (Ext. data Fig. 3). The role of NT^{LS} neurons in anxiety and how this interacts with the role of these neurons in social behavior should be discussed.

The reviewer's point is well taken. We now discuss new evidence in our revised manuscript, which shows that the generalized anxiety behavior, measured by OFT/EPM, is separable from social behavior deficits. 1) When we stimulate NT^{LS} neurons in stress naïve mice, we are able to produce a generalized exploratory deficit on EPM/OFT (Extended data Fig. 8), however, such stimulation does not induce social avoidance (Extended data Fig. 6 a-d). 2) Both RES and SUS mice in the CSDS model exhibit anxiety-like behaviors in the OFT and EPM, yet only SUS mice exhibit social avoidance and reduced social reward. 3) CVS produces an increase in generalized anxiety-like behavior measured in OFT/EPM, yet it has no effect on social interaction or social reward (Extended data Fig. 6 e, f).

The second question about NT/*Crhr2* and anxiety/social behavior is indeed a great question. In Anthony et al., *Cell*, 2014, the authors tested the role of *crhr2* neurons in anxiety-like behavior. Here we confirm that NT^{LS} neurons largely overlap with *crhr2* and of course they do regulate anxiety, but we also show that NT^{LS} neurons play a role in context-dependent social behavior that can be dissociated from generalized anxiety (see also response to reviewer 1). We've now added this to the discussion.

8) In Fig. 4b and 4i, the authors use a paired two-tailed t-test to analyze their results, but it seems that given that these mice were run together and are being displayed on the same graph, the appropriate statistic to use would be a 3X2 mixed ANOVA, with drug condition as the repeated measure and virus condition as the between-subjects measure, then follow this up with post-hoc tests to look at differences between CNO and vehicle within each viral condition. Same argument applies elsewhere in the figure.

We performed a 3X2 mixed ANOVA as the reviewer advised and replaced the analysis in Fig. 4.

Referee #4 (Remarks to the Author):

In this study, Li. et. al. identified the LS Nt cells as an important population for chronic social-defeat induced social avoidance and social reward learning. They found differences in LS Nt cell responses in SUS and RES animals and demonstrated that LS Nt activity can bidirectionally modulate social investigation and social reward learning in defeated animals. The authors further identified the downstream targets of LS Nt cells and demonstrated that LS-AHN and LS-NAc pathway manipulation can phenocopy the LS Nt manipulation. This study is interesting and the results are generally robust. The reviewer particularly appreciates the inclusion of females in the study.

However, although the study attempts to reach a conclusion regarding LS Nt role in social reward learning, the data mainly supports a role of LS Nt in generalized social avoidance and general anxiety.

This is an important concern, which we have addressed with both new data and a more sophisticated discussion. First, to better clarify our position, we believe that the NT^{LS} neurons encode contextual information about past stressful/traumatic experiences to guide future responses. Thus, the reason why CVS does not cause social avoidance behavior like CSDS is that the past stressful experience related to CVS had no social context. In fact, when we modified the design and subsequently exposed chronic restraint stress (CRS) mice to non-social stimuli (ie. a restrainer tube) associated with the stress, their NT^{LS} neurons do indeed become activated and they avoid those stimuli.

In addition, while our data suggests that NT^{LS} neurons are capable of promoting generalized anxiety, we show that such behavior is dissociable from the context dependent effects of stress/trauma avoidance for the following reasons: 1) following CSDS both SUS and RES mice exhibit generalized anxiety

responses on the EPM and OFT, yet only SUS mice avoid juvenile social targets. 2) CVS induces generalized anxiety but not generalized social avoidance. 3) While stimulation of NT^{LS} neurons in stress naïve mice promotes generalized anxiety in the EPM and OFT, it is not sufficient to promote social avoidance.

In all, we believe our data collectively shows that NT^{LS} neurons can both function to signal generalized anxiety in the naïve state, but also can encode context specific information based on past stressful experience to guide avoidance behavior (both social and non-social).

Here are some specific comments:

1. Are the SUS and RES animals representing animals within one continuous distribution or two qualitatively different categories of animals? Please plot avoidance and each behavior parameter from all animals in one plot and determine whether the nature of the distribution (e.g. bi-modal or Gaussian).

We added this to our revised manuscript as Extended Data Fig. 1u. The data shows SI/ CPP/social investigation data is normally distributed. Social avoidance, however, is not normally distributed because avoidance in most CTRL and RES mice is close to or at 0.

2. The social CPP paradigm appears to be not very robust. As stated in the method, for female sCPP, during conditioning, the juvenile mice were confined in a wire mesh cup, which we found was necessary for females to form CPP, whereas males only formed a preference when they were able to freely interact with the juvenile outside the cup. Why females only form sCPP with a confined juvenile? If the social CPP is due to the positive valence of social interaction, shouldn't free interaction provide more opportunities for interaction?

We thank the reviewer for pointing this out. Social CPP is similar in magnitude to CPP induced by other natural rewards such as palatable food, however, it is typically lower in magnitude than what is observed in addictive drug CPP models. Our results are consistent in magnitude to many other social CPP papers published (Zernig, Geralda & Pinheiro, Barbara S. Behavioural Pharmacology, 2015; Courtney Cann, et al, Behav Neurosci. 2020; Gul Dolen, et al, Nature, 2013).

With regard to the design for female CPP, we spent 1 year optimizing the conditions by testing social targets of all ages, sexes, familiarity, littermates and sexual status, as well as the sexual or maternal status of the experimental mice. We now include these results (please see below) in Extended Data Fig. 1t, which we believe could be a useful resource for the field.

After consultation with Dr. Jill Becker, an expert in sex differences and female behavioral physiology, she suggested we confine the social targets in a cup during the CPP conditioning. The rationale being that female rodents may only find social interactions to be rewarding when they are able to control the social interaction. Indeed, her studies have shown that only when females are able to control social interaction during a paced mating bout do you observe dopamine release in the NAc. At any rate, of all the designs we've tested, we found this design to be the most effective in consistently reproducing female social CPP.

3. This study analyzed two behavioral phenotypes induced by chronic defeat: social avoidance (SI test) and social reward learning (social CPP test). SUS animals are defined as animals that show social avoidance towards a non-threatening conspecific. The SUS animals also show impaired social reward-dependent contextual learning. However, social avoidance and social reward learning appears to be only weakly correlated especially in males (although it did reach significance). The correlation appears to be driven mainly by animals in RES and CTRL animals as in 1f and 1k, animals with SI ratio below 1 (SUS animals) showed highly variable CPP scores, suggesting that SUS animals can either show social CPP or not. These results suggest that these two behavioral phenotypes are separable. SUS animals do not necessarily show social CPP deficits. If the authors would like to study the role of LS NST cells in social reward learning, should not all the comparisons be made between animals that show social reward learning deficits and those not? As of now, all comparisons are made between animals that show social avoidance vs. those not.

The reviewer's point is well taken. We used the SI test because it's the best characterized way to define individual differences in a population of inbred mice in response to social defeat stress. We believe that given this historical precedent, it's important to represent the data using the historical definition of susceptibility and resilience.

However, we do agree that the use of susceptibility and resilience based on SI alone may miss important features of the individual differences in social behavior. We performed an additional analysis where we sorted mice according to the reviewer's suggestion from Fig. 1, which is now included as an extended data figure. When split by CPP score, both sexes show significant differences in the resident-intruder test index and a similar trend in SI ratio. Please see figure below and Extended data Fig. 2f-g.

In some experiments, animals' performance during social reward learning is not measured (Figure 2 and Figure 3). Thus, no conclusion can be reached regarding how LS NST cells differ between animals show defeat-induced deficits in social reward learning and those not.

We appreciate the reviewer's perspective. We performed fiber photometry (please see reviewer #3 comment 5), and optogenetics in the CPP test (please see reviewer #2 comment 5) and found that NT^{LS} neurons from SUS mice exhibit significantly higher activity during the social-paired conditioning session compared to the unpaired session. Further, optogenetic activation and inhibition of LS-AHN and LS-NAc during conditioning bi-directionally regulated sCPP.

4. It is interesting that LS NTS cells showed clear difference in SUS and RES animals. Is this difference limited to LS NTS cells? Do GFP-negative cells also show such a difference?

We appreciate the reviewer's suggestion. If we understand correctly, the reviewer is asking us to show c-Fos in NT-negative cells (not GFP-negative cells)? In our original submission, we included data showing that most cfos+ cells in SUS mice were also NT+ (~95-100%). In fact, cfos+ expression in non-NT+ cells was actually quite rare. To address this, we plotted the NT-negative data in SUS and RES mice and show it in Extended data Fig. 4I.

5. An important question is whether the differences in cell responses between SUS and RES animals are due to pre-defeat differences. This question is hard to address for terminal experiments (e.g. slice recording) but can be addressed with in vivo recording that allows for repeated sampling. Do SUS and RES animals differ in their responses to different social targets and non-social stress before defeat?

We thank the reviewer for this thoughtful suggestion and agree this is an interesting point. However, we believe that our data rules out the possibility that differences in NT^{LS} activity are preexisting. If such differences were reflective of a pre-existing difference, the stress naive CTRL animals would exhibit a bimodal distribution in their responses to a juvenile social target. Based on our data below, CTRL mice

shown in figure3d-l do not exhibit a bimodal distribution in their NT^{LS} responses during RI. Rather, our data suggests that the difference emerges only in SUS mice following CSDS. We are happy to include the individual traces below, but we believe that redoing the experiments using repeated sampling will not elicit any novel findings based upon our existing data.

6. As the non-social stress strongly activates LS NT cells and LS NT manipulation also influences performance in OFT and EPM, it appears that the cells signal general stress and anxiety level. The changes in cell excitability suggests a non-input specific change in responses. How specific is the LS NTS cell in vivo response to juvenile after chronic defeat? Do the cells also respond more to other non-social stimulus (e.g. novel object), especially those with negative valence (e.g. air puff)? If it is not specific to social cues, how do the NT cells differ from other LS populations, e.g. *Crhr2*, *Oxtr*, *Drd3*, *SST*, that have been indicated in stress, fear and anxiety?

Please also see response #1 Rev 1 and #7 Rev 3. To clarify our position, we do not believe these cells specifically control social behaviors, but rather they seem to be involved in more general computations that use past information from stressful or threatening situations to guide future behaviors towards cues associated with those threatening or stressful situations. In addition, while they clearly regulate anxiety, we provide evidence showing that the generalized anxiety behavior, measured through OFT/EPM, is separable from social behavior deficits. 1) When we stimulate NT^{LS} neurons in stress naïve mice, we are able to produce a generalized exploratory deficit in EPM/OFT (Extended data Fig. 8), however, such stimulation does not induce social avoidance to a social target (Extended data Fig. 6 a-d). 2) Both RES and SUS mice in the CSDS model exhibit anxiety-like behaviors in the OFT and EPM, yet only SUS mice exhibit social avoidance and reduced social reward. 3) CVS produces an increase in generalized anxiety-like behavior, yet it has no effect on social interaction or social reward (Extended data Fig. 6 e, f). Together, this supports that NT^{LS} neurons play a role in context-dependent negative social behavior that can be dissociated from generalized anxiety (see also response to your point #7 below and to reviewer 1). We've now added this to the discussion.

7. It is interesting that the chemogenetic induced changes in social investigation is defeat-experience dependent. Please discuss the LS role in the known social avoidance circuit (e.g. with NAc, VTA, BLA). Why increase the LS activity alone is insufficient in driving social avoidance behavior?

We appreciate the reviewer's advice. We now discuss the role of LS circuits in mediating social avoidance (see page 7). We believe that the reason NT^{LS} stimulation is insufficient to induce social avoidance in naïve mice is related to its role in computing contextual information. Please see response to reviewer 1 for additional discussion of our hypothesis. In this revision, we tested whether animals can

encode past aversive experience in order to generalize responses to related stimuli in the future. In the case of our CSDS model, the past social trauma is social defeat, thus the animals require that past traumatic social experience to drive social avoidance. In the case of chronic restraint stress, the animals encode information about cues related to restraint and therefore avoid restrainer tubes, but not social targets. Thus, in the absence of this past negative social experience in naïve mice, stimulation of NT^{LS} neurons has no effect on social behavior.

8. Does chemogenetic manipulation of LS Nt cells influence social CPP in stress-naïve animals?

This is really a good point and we apologize for omitting this important data. In fact, we have done the experiment and found no effect of NT^{LS} manipulations on social CPP, now we included it as Extended data Fig. 5e.

9. The conclusion regarding the extent of overlapping of LS-AHN, LS-PAG, LH-NAc projectors in Extended Figure 7 requires further investigation. The number of retrogradely labeled cells in the LS is generally low, especially those labeled with tdTomato. With such a low number of cells being labeled, the chance overlap will be close to 0. Thus, a negative conclusion (no overlap) is hard to draw. The fact that 7/17 LS cells retrogradely labeled from AHN also retrogradely labeled from NAc suggest that the extent of bifurcation to NAc and AHN is likely to be very high. The authors need to either increase the labeling efficiency (e.g. CTB could work much better) or use a different method, e.g. examine the whole brain projection pattern of LS cells that are retrogradely labeled from one specific region.

We thank the reviewer for their thoughtful comments. The efficiency of retrograde-AAV-DIO is in fact very low and we agree that more cells are required to make conclusions about connectivity. We adjusted the virus titer/volume, injected a new group of mice and measured from a much larger pool of NT^{LS} neurons for better quantifications. Please see below.

We also used CTB as the reviewer suggested, and while we got a much better yield, we were not successful at combining this approach with histological detection of NT via FISH. Because CTB is a peptide/protein, we believe that the protease step may have interfered with the FISH protocol to detect NT. That said, our results largely support data obtained with cell-type specific rabies tracing that LS inputs to NAc, AHN and PAG are largely non overlapping. See below and Extended Data Fig. 12.

Reviewer Reports on the First Revision:

Referees' comments:

Referee #1 (Remarks to the Author):

I thank the authors for addressing my many questions and presenting relevant new data. Given the current state of the manuscript I believe it is nearly ready for publication and is a significant and novel advance.

A few minor notes:

- I couldn't find the definition for sCCP acronym anywhere
- For all plots with error bars, please show the raw data (figure 1E, J; figure 4e-h, i-o)

In response to this: "We find NTL5 neurons do respond to motion (initiation of movement, rearing, and turning head), but the Ca²⁺ responses are far lower than those elicited in response to stressful or threatening events."

This seems to be a difference in magnitude rather than quality. Tail suspension and attack are accompanied by vigorous muscle activation, whereas 'motion' as it is measured in this paradigm is quite mild. Therefore, it cannot—from the present experiments alone—be excluded that all of the fluorescence activity observed is stimulus-independent muscle activation.

Referee #2 (Remarks to the Author):

The authors were responsive to my criticism in general. I have a lingering technical concern. The authors perform opto-inhibition studies by illuminating at the terminals. This is notoriously problematic. Further, it appears that many of the p values are hovering around 0.05 to 0.07 in these experiments. It would be very useful to see individual data points, and in addition, it would be useful to show that this approach actually inhibits inputs. Further, it is unclear why some of these controls in these experiments do not show the social CPP behavior, nor is it clear what kind of comparisons are being made to claim that 'bidirectional modulation alters behavior'. Are the authors conducting 2 way ANOVA for each condition, or just comparing t tests?

Referee #3 (Remarks to the Author):

The authors adequately addressed all of my concerns with their revisions to the text and new data.

Referee #4 (Remarks to the Author):

The authors have addressed all the reviewer's comments thoughtfully.

Author Rebuttals to First Revision:

Response to the Reviewers

Referee #1 (Remarks to the Author):

I thank the authors for addressing my many questions and presenting relevant new data. Given the current state of the manuscript I believe it is nearly ready for publication and is a significant and novel advance.

A few minor notes:

- I couldn't find the definition for sCCP acronym anywhere

We have introduced the acronym sCPP in the introduction when it is first discussed (please see line 60).

- For all plots with error bars, please show the raw data (figure 1E, J; figure 4e-h, i-o)

We have now plotted individual points for all graphs (please see below and also see response to reviewer 2)

Fig.1

Fig.4

Extended data Fig.6

In response to this: 'We find NTL5 neurons do respond to motion (initiation of movement, rearing, and turning head), but the Ca²⁺ responses are far lower than those elicited in response to stressful or threatening events.'

This seems to be a difference in magnitude rather than quality. Tail suspension and attack are accompanied by vigorous muscle activation, whereas 'motion' as it is measured in this paradigm is quite mild. Therefore, it cannot 'from the present experiments alone' be excluded that all of the fluorescence activity observed is stimulus-independent muscle activation.

We agree with the reviewer that this cannot be fully ruled out. However, we do see greater changes in fluorescence when a susceptible mouse is engaged in passive social interaction. The change in fluorescence is captured prior to initiation of movement by the experimental mouse. Thus, it seems likely that this is not due solely to muscle activation.

Referee #2 (Remarks to the Author):

The authors were responsive to my criticism in general. I have a lingering technical concern. The authors perform opto-inhibition studies by illuminating at the terminals. This is notoriously problematic. Further, it appears that many of the p values are hovering around 0.05 to 0.07 in these experiments. It would be very useful to see individual data points, and in addition, it would be useful to show that this approach actually inhibits inputs. Further, it is unclear why some of these controls in this experiment do not show the social CPP behavior, nor is it clear what kind of comparisons are being made to claim that 'bidirectional modulation alters behavior'. Are the authors conducting 2-way anova for each condition, or just comparing t tests?

We appreciate the reviewers' concerns. The experiment required ~100 mice in order to obtain 8-9 per condition in each sex. Unfortunately, the design was slightly underpowered to split males and females for analysis. Given that males and females showed the same trend, we combined them and performed a 2-way ANOVA to compare time in the paired vs. unpaired side during the pretest vs. the posttest. We show that EYFP resilient mice exhibit a significant social preference that is blocked by ChR2 terminal stimulation (panel f and h). We also show that EYFP susceptible mice do not show a significant social preference whereas NpHR terminal inhibition in susceptible mice significantly enhanced social preference (panel g and i). While we've only included this combined analysis and accompanying figure within the rebuttal letter, we're happy to replace Extended data figure 10 of the manuscript with it if the reviewer and editor feel it is appropriate.

Sexes combined

We have also plotted the individual points (please see Extended data Fig.10 and also below) from the original graphs to show the trend for each data point for each sex separately. And we used two-way repeated measures ANOVA for all the analysis in this figure.

It's difficult to provide evidence for NpHR inhibition since neurons in both downstream regions (ie. AHN and NAc) are not spontaneously active in slice. In our previous studies of lateral habenula, an area in which neurons are spontaneously active in slice, we have shown that NpHR can inhibit synaptic transmission both in vivo and ex vivo (Golden S, et al, 2016). While we agree that NpHR has its limitations, we were encouraged to see that inhibition of terminals with it produced opposite behavioral effects compared to ChR2 stimulation, suggesting that it is indeed working as expected. That said, we are very excited about the new optogenetic tools (e.g. eOPN and PPO) and look forward to using them in our future studies. We now discuss these new tools and their potential usage in future studies on line 275.

Reviewer Reports on the Second Revision:

Referees' comments:

Referee #2 (Remarks to the Author):

the authors were responsive to my comments.